# Surprising Behaviour of the Wageningen B-Screw Series Polynomials

Stephan Helma 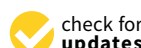

Stone Marine Propulsion (SMP) Ltd., SMM Business Park, Dock Road CH41 1DT, UK; sh@smpropulsion.com

**Abstract:** Undoubtedly, the Wageningen B-screw Series is the most widely used systematic propeller series. It is very popular to preselect propeller dimensions during the preliminary design stage before performing a more thorough optimisation, but in the smaller end of the market it is often used to merely select the final propeller. Over time, the originally measured data sets were faired and scaled to a uniform Reynolds number of $2 \cdot 10^6$ to increase the reliability of the series. With the advent of the computer, polynomials for the thrust and torque values were calculated based on the available data sets. The measured data are typically presented in the well-known open-water curves of thrust and torque coefficients $K_T$ and $K_Q$ versus the advance coefficient $J$. Changing the presentation from these diagrams to efficiency maps reveals some unsuspected and surprising behaviours, such as multiple extrema when optimising for efficiency or even no optimum at all for certain conditions, where an optimum could be expected. These artefacts get more pronounced at higher pitch to diameter ratios and low blade numbers. The present work builds upon the paper presented by the author at the AMT'17 and smp'19 conferences and now includes the extended efficiency maps, as suggested by Danckwardt, for all propellers of the Wageningen B-screw Series.

**Keywords:** propeller; Wageningen B-screw Series; open-water characteristics; propeller efficiency map; Danckwardt diagram; optimum propeller

---

## 1. Introduction

### 1.1. The Wageningen B-Screw Series and Its Polynomial Representation

The Wageningen B-screw Series dates back to 1936 [1], when the first results were published. In the following years, the series was systematically expanded to include more than 120 single propellers. The measured data were presented inter alia in open-water diagrams showing the dimensionless thrust and torque coefficients, $K_T$ and $K_Q$, and the open-water efficiency $\eta_o$ as functions of the also dimensionless advance coefficient $J$:

$$J = \frac{v_a}{nD}, \tag{1}$$

$$K_T = \frac{T}{\rho n^2 D^4}, \tag{2}$$

$$K_Q = \frac{Q}{\rho n^2 D^5} = \frac{P_P}{2\pi \rho n^3 D^5}, \text{ and} \tag{3}$$

$$\eta_o = \frac{J}{2\pi} \cdot \frac{K_T}{K_Q}, \tag{4}$$

where $T$ = measured thrust; $Q$ = measured torque; $P_P$ = propeller power; $\rho$ = water density; $n$ = shaft speed (in $\text{s}^{-1}$); $D$ = propeller diameter; and $v_a$ = speed of advance. If not stated otherwise, all values are in SI base units.

All tested propeller models had a diameter of 240 mm and, hence, different section chord lengths due to the variable blade area ratio and number of propeller blades. The propellers were tested in varying model basins of MARIN using a diverse rate of revolutions resulting in considerably different Reynolds numbers for each propeller in the whole series. Oosterveld and van Oossanen engaged in the formidable tasks of scaling all available open-water data sets to a uniform Reynolds number of $2 \cdot 10^6$ (based on chord length and section advance speed) and calculating polynomials for the thrust and torque coefficients by multiple regressions analysis [2]. With the help of these polynomials, it is possible to calculate these coefficients as functions of the advance coefficient $J$, the pitch to diameter ratio $P/D$, the expanded blade area ratio $A_e/A_0$, and the number of blades $Z$:

$$K_T = \sum C_{s,t,u,v} \cdot J^s \cdot (P/D)^t \cdot (A_e/A_0)^u \cdot Z^v \text{ and} \tag{5}$$

$$K_Q = \sum C_{s,t,u,v} \cdot J^s \cdot (P/D)^t \cdot (A_e/A_0)^u \cdot Z^v, \tag{6}$$

where $C_{s,t,u,v}$ = coefficient; and $s$, $t$, $u$, and $v$ = whole-number exponents.

These polynomials are nowadays widely used in either selecting the optimum propeller or as a basis for further refinements. It is therefore of utmost significance that these polynomials are consistent and accurate.

### 1.2. Time Line of Experimental Results

As shown by Helma for the B4-70 propeller, the open-water characteristics have changed substantially over time [3], see Figure 1. The effect on the widely used $B_p$–$\delta$ diagram was also outlined in the same work, see Figure 2.

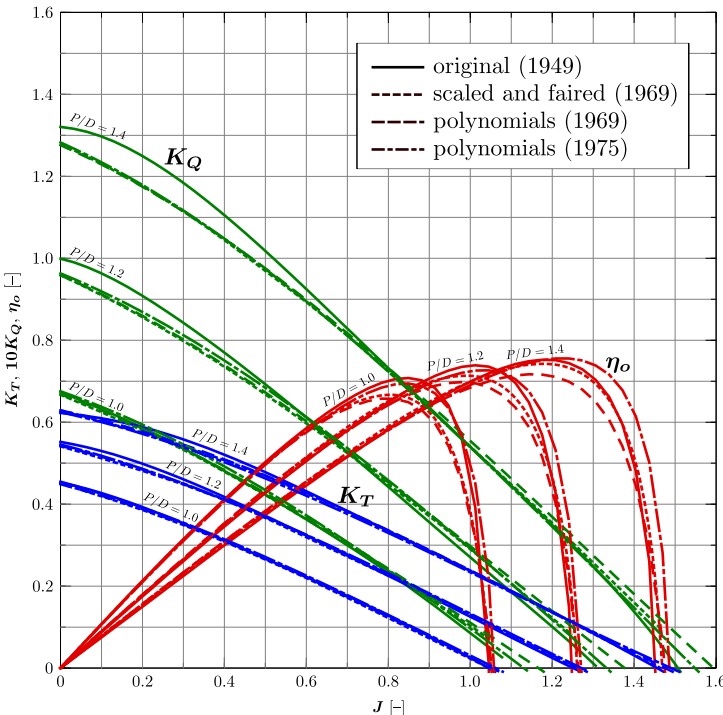

**Figure 1.** Open-water diagrams originally published by van Lammeren and van Aken in 1949 [4], scaled and faired, the fitted polynomials (both by van Lammeren et al. in 1969 [5]), and the most recent polynomials by Oosterveld and van Oossanen (1975) [2]) of propeller B4-70. The Reynolds number is $2.72 \cdot 10^5$ for the original data, otherwise $2 \cdot 10^6$. Reproduced from [3] with permission from AMT'17, 2017.

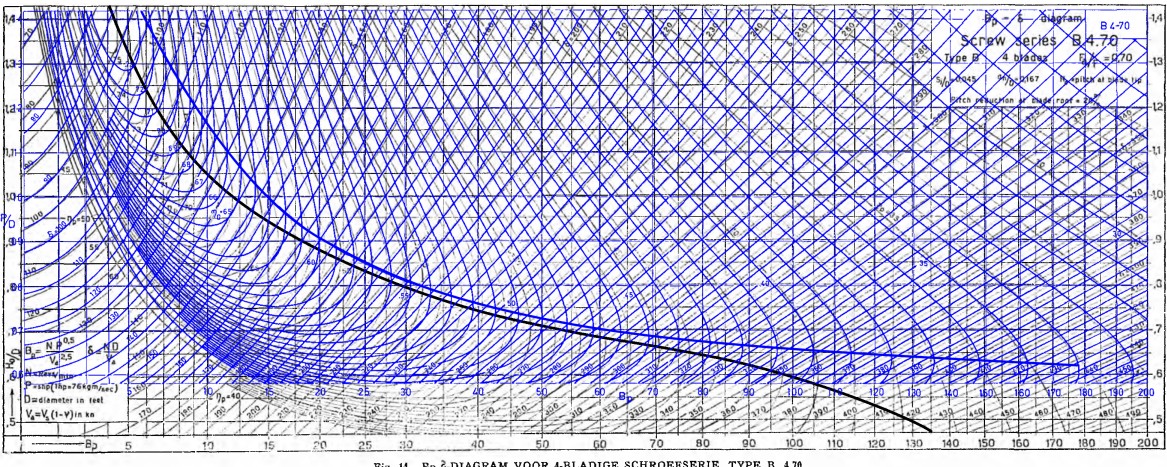

**Figure 2.** $B_p$–$\delta$ diagram of propeller B4-70 based on the originally published values by van Lammeren and van Aken in 1949 ([original black print, Reynolds number = $2.72 \cdot 10^5$) and on the scaled and faired open-water characteristics by van Lammeren et al. in 1969 (blue, Reynolds number = $2 \cdot 10^6$). Note the different lines for $\eta_{o,max}$ (in bold). $B_p = N \cdot \sqrt{P_P}/v_a^{2.5}$ and $\delta = N \cdot D/v_a$, where $N$ = shaft speed (in $\min^{-1}$); $P_P$ = delivered power (in hp); $v_a$ = advance speed (in kn); and $D$ = propeller diameter (in feet). Reproduced from [4,5], with permissions from SWZ | Maritime and Society of Naval Architects and Marine Engineers, 1949 and 1969.

## 2. Efficiency Maps

### 2.1. Basic Efficiency Maps

It is assumed, that the presentation in the form of the standard open-water diagrams for propellers are known to the reader. To recap, these diagrams show three families of curves with the thrust and torque coefficients and the efficiency as functions of the advance coefficient—$K_T(J)$, $K_Q(J)$, and $\eta_o(J)$—for a set of constant $P/D$-values for each propeller of the Wageningen B-screw Series.

Many authors suggested a different way of representing the nondimensional open-water data (see Appendix A.2); we will call them efficiency maps, as proposed by many of these authors. There are two efficiency maps: the $K_Q$–$J$ and the $K_T$–$J$ map. On the $K_Q$–$J$ efficiency map, we draw the family of $K_Q(J)$ curves for our set of constant $P/D$-values as for the conventional open-water diagram (thin red lines, please refer to Figure 3a). Instead of adding the family of efficiency curves for constant $P/D$-values, we add contour lines for the efficiency: $\eta_o\left(J, K_Q(J)\right)$ = const for a set of selected $\eta_o$-values (thin black lines). We can also draw the contour lines for the thrust coefficient: $K_T\left(J, K_Q(J)\right)$ = const (dashed red lines). So far, this gives us a diagram with exactly the same information content as for the conventional open-water diagram, just displayed in a different way.

We can now draw two lines of maximum propeller efficiency into this basis efficiency map: "$\eta_{o,max}$ for $J$ = const" (bold black line) and "$\eta_{o,max}$ for $P/D$ = const" (bold red line). (Appendix A.1 describes their derivation.) In the open-water diagram, the curves corresponding to the "$\eta_{o,max}$ for $J$ = const" and "$\eta_{o,max}$ for $P/D$ = const" lines would be the envelope to and the line connecting the maxima of all $\eta_o(J)$ curves, respectively.

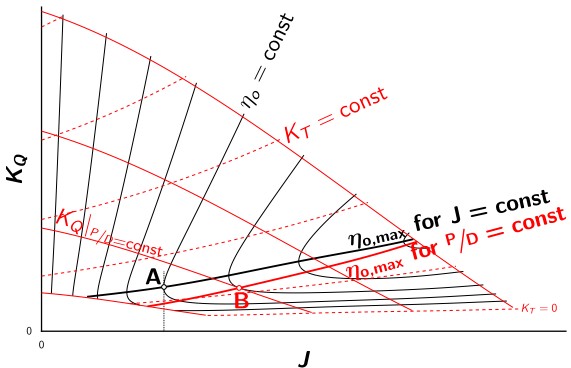

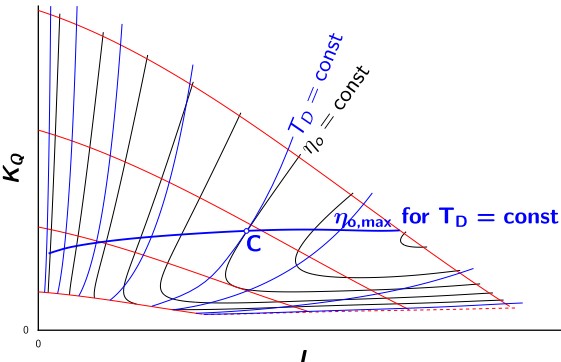

(**a**) Basic efficiency map showing $K_Q$ (thin red lines), $K_T = $ const (dotted red lines), $\eta_o = $ const (thin black lines), and the construction of "$\eta_{o,max}$ for $J = $ const" (dotted black line and point A, bold black line) and "$\eta_{o,max}$ for $P/D = $ const" (point B, bold red line).

(**b**) Extended efficiency map additionally showing $T_D = $ const (thin blue lines) and the construction of "$\eta_{o,max}$ for $T_D = $ const" (point C, bold blue line).

**Figure 3.** Composition of the $K_Q$–$J$ efficiency map. For its construction, see Appendix A.1.

## 2.2. Enhanced Efficiency Maps

To enhance the basic efficiency map, these families of curves can be added:

$$T_D = \frac{K_T}{J^2} = \text{const}, \tag{7}$$

$$P_D = \frac{K_Q}{J^3} = \text{const}, \tag{8}$$

$$T_n = \frac{K_T}{J^4} = \text{const, and} \tag{9}$$

$$P_n = \frac{K_Q}{J^5} = \text{const}. \tag{10}$$

These four curves have a very practical significance, because each of them eliminates one of the two unknowns in propeller optimisation: $D$ or $n$, the propeller diameter and the shaft speed, respectively. (Examples of how these diagrams can be used for propeller selection are presented in Appendix A.4).

Note that in the $K_Q$–$J$ efficiency map, the lines for $P_D$ and $P_n = $ const are ordinary curves to the single power of three and five, whereas the curves for $T_D$ and $T_n = $ const are truly parametric curves. Having established these families of curves, we can now draw the four lines of maximum propeller efficiency for each of them: "$\eta_{o,max}$ for $T_D = $ const", "$\eta_{o,max}$ for $P_D = $ const", "$\eta_{o,max}$ for $T_n = $ const", and "$\eta_{o,max}$ for $P_n = $ const", all connecting the points of maximum efficiency (see Appendix A.1.3 for how they are constructed). Figure 3b shows the family of curves for $T_D = $ const (thin blue lines) and for "$\eta_{o,max}$ for $T_D = $ const" (bold blue line).

As mentioned earlier, there exists a second diagram: the $K_T$–$J$ efficiency map with $K_T$ as the ordinate. It is composed of the families of $K_T(J)$ curves for the set of constant $P/D$-values and the contour lines $K_Q(J, K_T(J))$ and $\eta_o(J, K_T(J)) = $ const for the set of selected $K_Q$- and $\eta_o$-values. The lines for maximum efficiency are called "$\eta_{o,max}$ for $J = $ const" and "$\eta_{o,max}$ for $P/D = $ const" as before. This efficiency map can also be enhanced with the families of curves "$T_D$, $P_D$, $T_n$, and $P_n = $ const", where the curves for $T_D$ and $T_n = $ const are curves to the single power of three and five.

There are certain advantages of efficiency maps over open-water diagrams for finding the optimum propeller. These are described in Appendixes A.3 and A.4. For the purpose of this article, we will concentrate on the shape of the lines for maximum efficiencies. To recap, these lines are the solutions of the optimisation problems under different constraints. We will use the $K_Q$–$J$ efficiency map enhanced by the addition of the $T_D$ and $T_n = $ const curves and the $K_T$–$J$ map enhanced with

the $P_D$ and $P_n$ = const curves, as proposed by Danckwardt [6]. We will call them the T–J and P–J Danckwardt diagrams (see Appendix A.2 for a short historical overview).

*2.3. Remake*

The author presented efficiency maps for the Wageningen B-screw Series during the smp'19 conference [7]. For the purpose of this paper, the two Danckwardt diagrams for all propellers of the Wageningen B-screw Series, as outlined in Table 1, were recreated with the help of a purpose-made computer program. This program employs the polynomials (5) and (6), as described in Section 1 and published by Oosterveld and van Oossanen in 1975 [2]. For these newly generated diagrams, the symbols were updated to the ITTC nomenclature [8] (see also Table A1 for the differences to the nomenclature used by Danckwardt). In addition to the curves presented in the original diagrams, the line for "$\eta_{o,max}$ for $P/D$ = const" and the contour lines for $K_T(J, K_Q)$ or $K_Q(J, K_T)$ = const were added. All these recalculated Danckwardt diagrams are presented in Appendix B, and Table A4 shows an overview of the composition of these diagrams. Examples of how these diagrams are used are explained in Appendix A.4.

**Table 1.** Summary of the propeller models of the Wageningen B-screw Series.

| $Z$ | $A_e/A_0$ | | | | | | | | | | |
|---|---|---|---|---|---|---|---|---|---|---|---|
| 2 | 0.30 | 0.38 | | | | | | | | | |
| 3 | | 0.35 | | 0.50 | | 0.65 | | 0.80 | | | |
| 4 | | | 0.40 | | 0.55 | | 0.70 | | 0.85 | | 1.00 |
| 5 | | | | 0.45 | | 0.60 | | 0.75 | | 0.90 | | 1.05 |
| 6 | | | 0.50 | | 0.65 | | 0.80 | | 0.95 | |
| 7 | | | | 0.55 | | 0.70 | | 0.85 | | |

The computer program calculates the lines of maximum efficiency by finding all points, where the tangents to the $T_D$, $T_n$, $P_D$, and $P_n$ curves are tangential to the efficiency contour lines. As a free variable, the pitch to diameter ratio $P/D$ was used. This choice was found to be necessary to be able to remove possible multiple solutions for the $\eta_{o,max}$ lines. It must be mentioned that, as a possible consequence, the calculation of the $\eta_{o,max}$ lines for $T_D$, $T_n$, $P_D$, and $P_n$ = const sometimes did not succeed at one or the other boundary due to numerical difficulties. The artefacts on the left boundary at low $J$- and $P/D$-values were manually deleted and extrapolated by hand whenever possible. The right border, where the $J$- and $P/D$-values are high, posed a different numerical challenge. In all cases however, it was possible to reconstruct the valid line manually, but sometimes not right up to the maximum value of $P/D$. It should be mentioned that these difficulties never arose at high $P/D$-values when the $\eta_{o,max}$ line doubles back, as discussed in the following section.

## 3. Ambiguity of the $\eta_{o,max}$ Lines

*3.1. Introductory Example*

We have seen that the Danckwardt diagrams lend themselves to finding the optimum propeller under certain constraints. For the sake of argument, let us assume that we have already decided on the blade number ($Z = 5$) and blade area ratio ($A_e/A_0 = 0.9$), and we know the torque $Q$ (from the given available power $P_P$), the inflow velocity $v_a$, and propeller diameter $D$ but not the shaft speed $n$, which should be optimised together with the pitch to diameter ratio $P/D$. Using these known values, we can calculate $P_D$ from Equation (A4), which we assume to be 0.15. On the Danckwardt P–J diagram for the Wageningen B5-90 propeller (see Appendix B), we find the intersection of the curve for $P_D$ = 0.15 with the "$\eta_{o,max}$ for $P_D$ = const" line (blue lines). We can read off the values for $J$, $K_T$, $K_Q$, $P/D$, and $\eta_o$ (approximately 0.65, 0.24, 0.041, 1.04, and 0.60, respectively).

Let us now assume that we want to investigate a three-bladed propeller with the blade area ratio $A_e/A_0$ of 0.80 for the same operating condition. In the diagram for the Wageningen B3-80 propeller,

we find the intersection for our assumed value of $P_D = 0.15$ with the "$\eta_{o,max}$ for $P_D = $ const" line (blue lines) at $J$, $K_T$, $K_Q$, $^{P}\!/_{D}$, and $\eta_o$ equal to 0.62, 0.20, 0.035, 1.00, and 0.57, respectively. However, there also exists another intersection between the $P_D = 0.15$ and the "$\eta_{o,max}$ for $P_D = $ const" lines at a higher $^{P}\!/_{D}$-value: 0.74, 0.295, 0.062, 1.30, and 0.56. It looks to be that there are two optimum propellers for this condition, since the $\eta_{o,max}$ lines are solutions to the optimisation problem under the given constraints!

The reason for this somewhat puzzling behaviour is obviously the doubling back of the $\eta_{o,max}$ lines. Two solutions to the optimisation problem exist in the region of this overlap, whereas no solution to the optimisation problem exists right of this region of overlap.

### 3.2. Classification

To classify this overlap, the value of $T_D\big|_{^{P}\!/_{D}\big|_{max}}$, where the "$\eta_{o,max}$ for $T_D = $ const" line intersects the maximum $^{P}\!/_{D}\big|_{max}$ curve, was calculated for every propeller in the Wageningen B-screw Series (please refer to Figure 4) as proposed in [7]. The intersection of this $T_D = $ const curve with the "$\eta_{o,max}$ for $T_D = $ const" line was found. The pitch to diameter ratio at this intersection is denoted as $\widehat{^{P}\!/_{D}}\big|_{T_D}$. The minimum value of $T_D\big|_{min}$ is determined, where there is also no optimum propeller right of this curve! The difference between these two values for $T_D$ is denoted as $\Delta T_D$. For the $T_n$, $P_D$, and $P_n$ curves, these values are calculated accordingly. Table 2 shows an overview of all these values for all propellers in the Wageningen B-screw Series. The table clearly shows that this overlap does not just sporadically occur, but that it is a widespread phenomena.

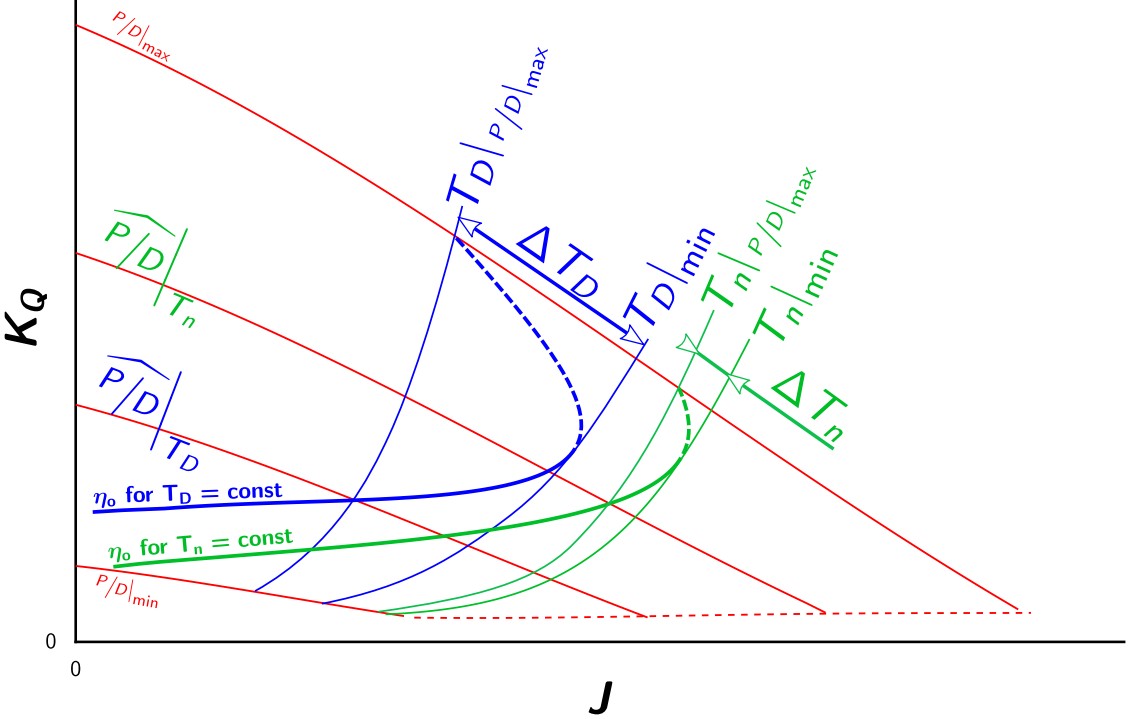

**Figure 4.** Symbols and definitions used to describe the overlap of the $\eta_{o,max}$ lines given in Table 2 using the example of the $T_D$ and $T_n$ curves in the Danckwardt T–J efficiency map.

**Table 2.** Main characteristics of all overlaps found in the recreated Danckwardt diagrams. (For symbols and definitions, see Section 3.2 and Figure 4).

| Propeller | $T_D$ | | | $T_n$ | | | $P_D$ | | | $P_n$ | | |
|---|---|---|---|---|---|---|---|---|---|---|---|---|
| | $\widehat{P/D}$ | $T_D\|_{min}$ | $\Delta T_D$ | $\widehat{P/D}$ | $T_n\|_{min}$ | $\Delta T_n$ | $\widehat{P/D}$ | $P_D\|_{min}$ | $\Delta P_D$ | $\widehat{P/D}$ | $P_n\|_{min}$ | $\Delta P_n$ |
| B2-30 | | ——— | | | ——— | | | ——— | | | ——— | |
| B2-38 | | ——— | | | ——— | | | ——— | | | ——— | |
| B3-35 | 0.94 | 0.155 | 0.182 | 1.14 | 0.102 | 0.043 | 0.93 | 0.033 | 0.049 | 1.14 | 0.021 | 0.010 |
| B3-50 | 0.89 | 0.167 | 0.225 | 1.13 | 0.114 | 0.042 | 0.88 | 0.037 | 0.065 | 1.12 | 0.025 | 0.010 |
| B3-65 | 0.86 | 0.228 | 0.385 | 1.12 | 0.162 | 0.056 | 0.86 | 0.054 | 0.127 | 1.11 | 0.037 | 0.014 |
| B3-80 | 0.86 | 0.384 | 0.932 | 1.11 | 0.266 | 0.094 | 0.85 | 0.101 | 0.389 | 1.11 | 0.063 | 0.025 |
| B4-40 | 1.02 | 0.272 | 0.134 | 1.18 | 0.177 | 0.042 | 1.01 | 0.063 | 0.039 | 1.17 | 0.039 | 0.010 |
| B4-55 | 1.04 | 0.229 | 0.100 | 1.23 | 0.144 | 0.018 | 1.04 | 0.052 | 0.028 | 1.22 | 0.031 | 0.004 |
| B4-70 | 1.10 | 0.229 | 0.064 | 1.31 | 0.138 | 0.004 | 1.10 | 0.053 | 0.018 | 1.31 | 0.030 | 0.001 |
| B4-85 | 1.21 | 0.254 | 0.027 | | ——— | | 1.21 | 0.060 | 0.008 | | ——— | |
| B4-100 | 1.38 | 0.286 | 0.000 | | ——— | | 1.38 | 0.070 | 0.000 | | ——— | |
| B5-45 | 1.21 | 0.301 | 0.028 | 1.30 | 0.191 | 0.008 | 1.21 | 0.071 | 0.008 | 1.30 | 0.043 | 0.002 |
| B5-60 | 1.25 | 0.231 | 0.015 | 1.35 | 0.141 | 0.001 | 1.25 | 0.052 | 0.004 | 1.35 | 0.031 | 0.000 |
| B5-75 | 1.32 | 0.195 | 0.003 | | ——— | | 1.32 | 0.043 | 0.001 | | ——— | |
| B5-90 | | ——— | | | ——— | | | ——— | | | ——— | |
| B5-105 | | ——— | | | ——— | | | ——— | | | ——— | |
| B6-50 | 1.37 | 0.246 | 0.001 | 1.40 | 0.165 | 0.000 | 1.37 | 0.057 | 0.000 | 1.40 | 0.037 | 0.000 |
| B6-65 | 1.37 | 0.210 | 0.001 | | ——— | | 1.37 | 0.047 | 0.000 | | ——— | |
| B6-80 | 1.40 | 0.197 | 0.000 | | ——— | | 1.40 | 0.044 | 0.000 | | ——— | |
| B6-95 | | ——— | | | ——— | | | ——— | | | ——— | |
| B7-55 | 1.39 | 0.204 | 0.000 | | ——— | | 1.39 | 0.047 | 0.000 | | ——— | |
| B7-70 | | ——— | | | ——— | | | ——— | | | ——— | |
| B7-85 | | ——— | | | ——— | | | ——— | | | ——— | |

An overlap can only be observed for the "$\eta_{o,max}$ for $T_D$, $T_n$, $P_D$, and $P_n$ = const" lines, but never for "$\eta_{o,max}$ for $J$ = const" or "$\eta_{o,max}$ for $P/D$ = const".

## 4. Discussion

### 4.1. Evaluation

For the following discussion, it should be kept in mind that the $\eta_{o,max}$ lines were calculated in a descriptive way by finding all points, where the tangents to the families of the $T_D$, $T_n$, $P_D$, and $P_n$ = const curves coincide with the tangents to the contour lines of $\eta_{o,max}$ = const. Mathematically speaking, this is equivalent to solving an extrema problem.

To investigate the solutions, we take the propeller used in the example in Section 3.2 and plot the efficiencies along the $P_D$ = 0.15 curve against the $P/D$-value, see the blue line in Figure 5. It can be seen, that the extremum at the lower $P/D$-value of about 1.0 is a maximum, whereas the extremum at the higher $P/D$-value of about 1.3 is a minimum. Additionally shown in this figure is the run of the efficiency curves for $P_D$ = 0.1 and 0.07 (green and orange lines), the first represents $P_D|_{min}$ and just touches the "$\eta_{o,max}$ for $P_D$ = const" curve in the apex; the second comes to lay right of the apex. For these two cases, it is apparent that the propeller with the highest possible efficiency is situated beyond the boundary of $P/D$ = 1.4.

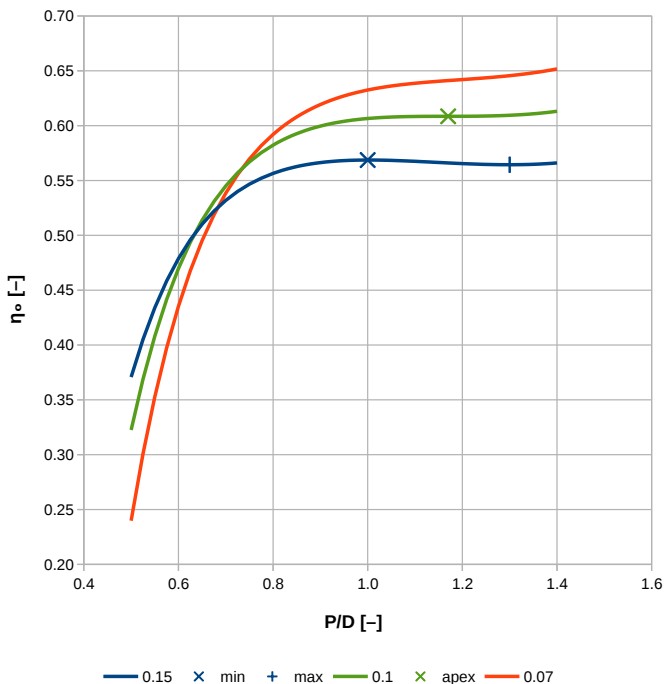

**Figure 5.** Run of the efficiencies versus pitch ratio $^P\!/_D$ along the lines $P_D = 0.15, 0.1,$ and $0.07$, corresponding to the region of the overlap $P_D|_{\max}$ and the region right of the overlap, respectively. Additionally shown are the extrema for $P_D = 0.15$ and the apex for $P_D|_{\max}$.

The author of this paper believes that these overlaps of the $\eta_{o,max}$ lines causing the ambiguities described are physically not explainable.

There are the obvious reasons: Firstly, the open-water efficiency drops and starts to climb again in the region of the overlap. Secondly, there is no optimum except at the boundary of the available data, right of the overlap.

Generally, we can argue that we can extend the propeller series to even higher $^P\!/_D$-values. Eventually we will arrive at a pitch setting, where the open-water efficiency will become zero, since such a propeller would have blades perpendicular to the section inflow and hence would not be able to accelerate water in the axial direction. Thus, it is not unreasonable to assume that a pitch to diameter ratio must exist, where the open-water efficiency is globally at its highest. Indeed, this can be seen in both Danckwardt diagrams for the Wageningen B2-30 and B2-38 propellers (see Appendix B): a peak of the open-water efficiency can be noticed at a *J*-value of about 1 and a $^P\!/_D$ ratio of about 1.1. It can be observed that all four lines for $\eta_{o,max}$ pass (and must pass) through this absolute maximum of $\eta_o$. Even if this point of the absolute maximum of $\eta_o$ comes to lie right of and above the set of the $K_T$ and $K_Q$ curves—and thus is not displayed in the diagram—the four lines for $\eta_{o,max}$ must still converge towards this single point of the global absolute maximum of $\eta_o$. Following this thought, it is evident that the lines of $\eta_{o,max}$ can not bend back, as can be seen with certain propellers, and this behaviour is deemed as physically inexplicable.

*4.2. Implications*

Admittedly, paper charts are seldom used nowadays in propeller design work, but depending on the computer algorithm used for automatically searching the propeller with the highest achievable efficiency, the following problems can be encountered: Firstly, in the region of the overlap, the computer program could pick the solution on the upper branch of the $\eta_{o,max}$ curve, if no appropriate checks are implemented. It could also jump between the two extrema. Equipped with the knowledge of the described behaviour, the algorithm can be tweaked to find the correct solution. Secondly, in the region right of the overlap, where there exists no optimum, the algorithm could calculate

the optimum propeller to have a pitch ratio of 1.4, which is right on the boundary of the available data. As can be clearly seen in Figure 5, the line for $P_D = 0.07$ still climbs in the vicinity of the boundary, indicating that there exist propellers with even higher efficiencies beyond the boundary of the tested propellers. It has to be emphasised that, based on the polynomials, there simply does not exist an optimum propeller in this region, where such an optimum propeller must exist. Thirdly, it can be argued that there exists an optimum propeller up to the point where the $\eta_{o,max}$ curve doubles back. Nevertheless, special care must be taken in the region of the apex, because the line of optimum efficiency already starts to swerve away.

### 4.3. Accuracy of the Polynomials

As the issue of the doubling back of the $\eta_{o,max}$ lines cast some doubts on the accuracy of the underlying polynomials, the question of the overall accuracy of the polynomials also arises. Whereas Figure 6a,b show a good agreement of the $\eta_{o,max}$ lines between the original and the recreated diagrams for the whole range but the area of doubling back, Figure 6c,d show a big discrepancy between the polynomials published in 1969 and 1975. These deviations can be of varying significance, depending on the subsequently employed and more detailed optimisation procedures. When the Wageningen B-screw Series is used to find the optimum propeller and the thus obtained dimensions will be kept fixed and only small corrections are applied to them—as is common practice at the smaller end of the market—it is of paramount importance that this optimisation routine based on the polynomials gives consistent and accurate results.

For large propellers, the outcome of the optimisation based on the Wageningen B-screw Series polynomials is used as the starting point for further and more detailed optimisation. An accurate starting point would speed up the subsequent full optimisation processes, but should not change the outcome. In reality, the optimised variable $D$ or $n$ found in the first step is very often kept fixed and will not be optimised further in the final optimisation. In this case, it is again of highest importance to get accurate results from the polynomials.

### 4.4. Provenance and Causes of Overlaps

It must be emphasised, that the overlaps observed are not a feature of the presentation in the form of efficiency maps, but of the underlying data, i.e., the polynomials (5) and (6) published by Oosterveld and van Oossanen in 1975 [2]. It should also be noted that at the time when Danckwardt published his diagrams—which show no overlaps at all, see Figure A1a,b—the and polynomials were not known yet. The design charts published by Yosifov et al. are already based on the polynomials [9]. On all their efficiency maps, the lines for $\eta_{o,max}$ stop before they reach the maximum $P/D$-value. Yosifov et al. do not mention or explain this behaviour. Those diagrams, where the lines stop far from the maximum $P/D$ ratio, are for the same propellers, where we have identified an overlap.

Nonetheless, it is not clear where these ambiguities were introduced during the process of manufacturing, measuring, fairing, scaling to uniform Reynolds number, and calculating the regression polynomials. Without further investigation into all of these steps, the source of this behaviour is not known, but some possibilities spring to mind: Between the testing of the first and the last propeller, a time span of more than 30 years passed. During this time span, it can be assumed that the manufacturing of the model propellers and the testing technology improved. The propellers were tested at different basins and also at different Reynolds numbers, and were only later corrected to a uniform Reynolds number of $2 \cdot 10^6$. Even the numerical regression used to calculate the polynomials could have introduced this behaviour. Helma shows in [3], for selected propellers, that the lines of maximum efficiency in the recreated Danckwardt diagrams follow the published lines by Danckwardt at low $P/D$-values (see Figure 6a,b). However, the regression curves given by van Lammeren et al. for propellers with four blades [5] already exhibit a troublesome behaviour at higher values of the pitch to diameter ratio (see Figure 6c,d).

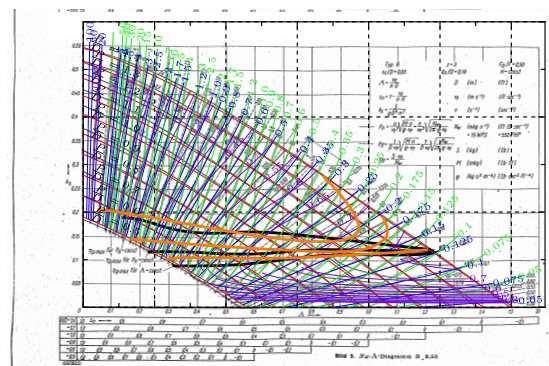

(**a**) P–J ($K_T$–$J$) diagram of propeller B3-50, 1956 (original black and white print) and 1975 (coloured).

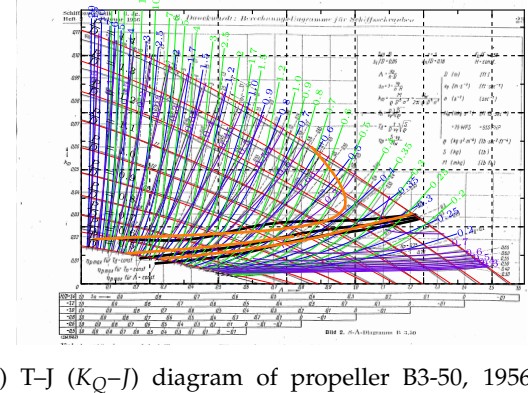

(**b**) T–J ($K_Q$–$J$) diagram of propeller B3-50, 1956 (original black and white print) and 1975 (coloured).

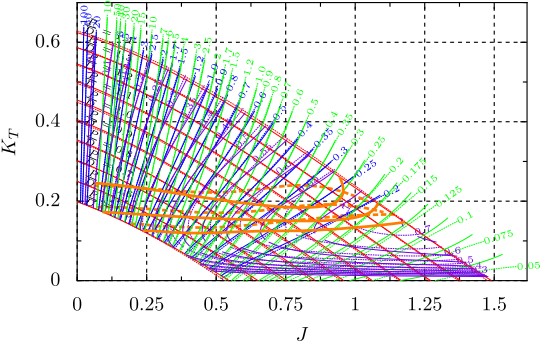

(**c**) T–J ($K_Q$–$J$) diagram of propeller B4-70, based on the polynomials from 1969 (broken lines) and 1975 (solid lines).

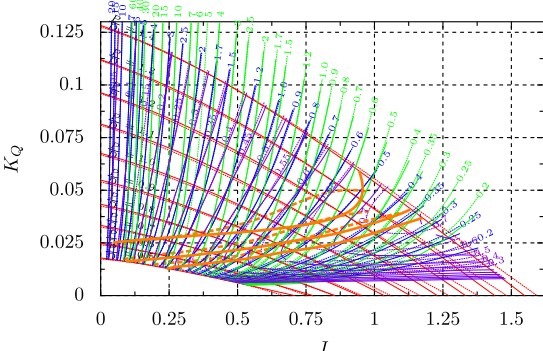

(**d**) P–J ($K_T$–$J$) diagram of propeller B4-70, based on the polynomials from 1969 (broken lines) and 1975 (solid lines).

**Figure 6.** Comparison between the original Danckwardt diagrams, 1956 [6] (probably based on the original data from 1949 [4]); diagrams calculated with polynomials for the scaled and faired open-water characteristics, 1969 [5]; and the most recent polynomials, 1975 [2]. Reproduced from [3], with permission from AMT'17, 2017.

## 5. Conclusions

With the help of the alternative presentation of open-water characteristics as efficiency maps, it was shown that the current set of polynomials for the Wageningen B-screw Series, as published by Oosterveld and van Oossanen in 1975 [2], shows some troublesome behaviours for higher pitch to diameter ratios for many propellers of the series.

Considering the widespread use of these polynomials, it is suggested to revisit the originally tested data and check all steps involved in the processing of the data sets for the deduction of the polynomials.

The propeller designers would be very well advised to take caution when designing propellers whenever any line of $\eta_{o,max}$ doubles back.

**Funding:** This research received no external funding.

**Conflicts of Interest:** The author declares no conflict of interest.

## Abbreviations

The following abbreviations are used in this manuscript:

| | |
|---|---|
| AMT | International Conference on Advanced Model Measurement Technology for the Maritime Industry |
| ITTC | International Towing Tank Conference |
| MARIN | Maritime Research Institute Netherlands |
| smp | International Symposium on Marine Propulsors |

## Appendix A. Efficiency Maps

*Appendix A.1. Lines of Maximum Efficiency*

Section 2 introduces the efficiency map and describes the elements found therein. It also introduces the lines "$\eta_{o,max}$ for $J$, $P/D$, $T_D$, $P_D$, $T_n$, and $P_n$ = const", where the efficiency is maximum under certain constraints. Efficiency maps lend themselves to construct these lines of maximum efficiency graphically, but they can also be calculated with the help of any optimisation procedures solving for maximum efficiency with the appropriate constraints. For the following discussion on how these lines are derived and their significance, please consult Figure 3.

Appendix A.1.1. Line "$\eta_{o,max}$ for $J$ = const"

If the $J$-value is known, the propeller with the maximum efficiency can be found by drawing a vertical line at this given value of $J$ (dotted black line). The propeller with the maximum efficiency can be found at the point where this vertical line just touches the efficiency contour line (point A). The $K_Q$-value can be read off the ordinate and the $K_T$- and $P/D$-values can be interpolated between the curves for $K_Q|_{P/D=const}$ and $K_T$ = const (thin red and dotted red lines). Connecting all points of these maximum efficiencies for every $J$-value gives us the line "$\eta_{o,max}$ for $J$ = const", which can be drawn into the efficiency map (bold black line). Note that the equivalent to this line on the conventional open-water diagram is the envelope to all $\eta_o(J)$ lines.

Appendix A.1.2. Line "$\eta_{o,max}$ for $P/D$ = const"

Another line of maximum efficiencies is the line called "$\eta_{o,max}$ for $P/D$ = const" (bold red line). This line can be used to find the propeller with the maximum efficiency, if the $P/D$ ratio is known. It connects all points where the tangents of the $K_Q$ curve (thin red line) and the $\eta_o$ contour line (thin black line) coincide (point B). On the conventional open-water diagram, this line corresponds to the line connecting the maxima of all $\eta_o(J)$ (which is situated below the envelope to the efficiency curves, resulting in a lower efficiency for the same advance coefficient).

Appendix A.1.3. Line "$\eta_{o,max}$ for $T_D$ = const"

If the propeller diameter, the speed of advance, and the (required) thrust are known, we can plot the curve

$$T_D = \frac{1}{D^2 v_a^2} \frac{T}{\rho} = const, \tag{A1}$$

which is equal to

$$T_D(J) = \frac{K_T(J)}{J^2} = const, \tag{A2}$$

into the efficiency map (thin blue lines). This formulation uses the old trick of eliminating the unknown shaft speed $n$, which now becomes part of the solution, when optimising for the highest possible efficiency.

This curve can be drawn either into the $K_T$–$J$ or the $K_Q$–$J$ efficiency map. In the $K_T$–$J$ diagram, the curve is a simple quadratic curve in the form of $cJ^2$, where $c$ = suitable constant; whereas in the $K_Q$–$J$ diagram, the line becomes the truly parametric curve with J as parameter

$$T_D\left(J, K_Q(J)\right) = \frac{K_T\left(J, K_Q(J)\right)}{J^2}. \tag{A3}$$

In case of a given set of constant $T_D$-values, a family of curves results.

Once again, a line for propellers with the highest possible efficiency can be constructed by connecting all points, where the tangent to the "$T_D$ = const" curve coincides with the tangent to the "$\eta_o$ = const" contour line (point C); this is called the "$\eta_{o,max}$ for $T_D$ = const" line (thick blue line).

Please refer to Appendix A.4 to see how this can be used to find the optimum propeller quickly if the thrust $T$, the propeller diameter $D$, and the speed of advance $v_a$ is known.

Appendix A.1.4. Lines "$\eta_{o,max}$ for $P_D$, $T_n$, and $P_n = const$"

Finally, other families of curves can be drawn:

$$P_D = \frac{K_Q}{J^3} = \frac{1}{D^2 v_a^2} \frac{Q n}{\rho v_a} = \frac{1}{D^2 v_a^2} \frac{P_P}{2\pi \rho v_a}, \tag{A4}$$

$$T_n = \frac{K_T}{J^4} = \frac{n^2}{v_a^4} \frac{T}{\rho}, \text{ and} \tag{A5}$$

$$P_n = \frac{K_Q}{J^5} = \frac{n^2}{v_a^4} \frac{Q n}{\rho v_a} = \frac{n}{v_a^4} \frac{P_P}{2\pi \rho v_a}. \tag{A6}$$

These formulations eliminate the shaft speed $n$, Equation (A4), and the propeller diameter $D$, Equations (A5) and (A6). In Appendix A.4, it is shown how these formulations help in solving each of the six possible optimisation problems a propeller designer can encounter.

The corresponding lines of maximum efficiency are called "$\eta_{o,max}$ for $P_D = const$", "$\eta_{o,max}$ for $T_n = const$", and "$\eta_{o,max}$ for $P_n = const$", respectively.

*Appendix A.2. Origins*

The first diagrams using the presentation discussed were published in 1917 by Bendemann and Madelung [10] and in 1923 by von der Steinen (Von der Steinen argues in his paper that he had finished it earlier, but could not publish it for 6 years because of a exceptionally high work load, thus claiming the intellectual of these diagrams.) [11]. Bendeman and Madelung based their idea and how the data should be presented on the polar diagram of aerofoils. In 1936, Papmel published design charts [12] using the same setup. Schoenherr included all four lines of maximum efficiency in 1949 [13]. All authors mentioned so far suggested to plot the $T_D$ and $T_n$ curves into the $K_T$–$J$ and $P_D$ and to plot $P_n$ into the $K_Q$–$J$ efficiency map. Bendemann and Madelung pointed out that the usage of a double logarithmic scale results in the $T_D$ and $T_n$ curves becoming straight lines, making it easier for the designer to use these maps.

Danckwardt calculated design charts for the Wageningen B-screw Series in 1956 [6], but instead of drawing the $T_D$ and $T_n$ curves into the $K_T$–$J$ efficiency map, he plotted the $P_D$ and $P_n$ curves (and vice versa). This deliberate decision makes life easier for the propeller designer (but not for the draftsman plotting these efficiency maps!), since now only one single chart is required to get the missing torque or thrust coefficient, which are not available in the efficiency maps suggested by previous authors. To honour the inventor, these came to be known as Danckwardt diagrams. We will refer to them as P–J and T–J diagrams to distinguish them from the general $K_T$–$J$ and $K_Q$–$J$ efficiency maps. As a mnemonic, remember that you use the T–J diagram whenever the thrust $T$ is known and P–J whenever the power $P_P$ (or the torque $Q$) is known.

In 1983, Yosifov et al. calculated the design charts according to Papmel for the Wageningen B-screw Series with the aid of a computer using the polynomials [14], which previously became available in 1975 [2]. Finally, Yosifov et al. published polynomials for the $\eta_{o,max}$ lines in 1986 [9], removing the need to resort to paper and pencil.

All these diagrams might use different symbols or alternative definitions of the variables (mostly multiplied by constant values or using inverse values). They also show different degrees of details, but they all build on the same idea of the efficiency map. Examples for original Danckwardt diagrams and design charts calculated by Yosifov et al. are shown in Figures A1 and A2, respectively.

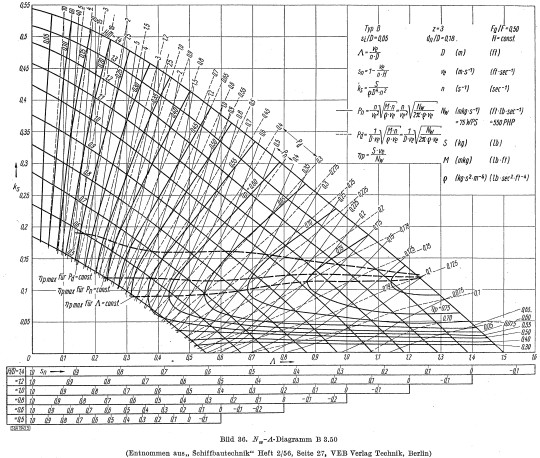

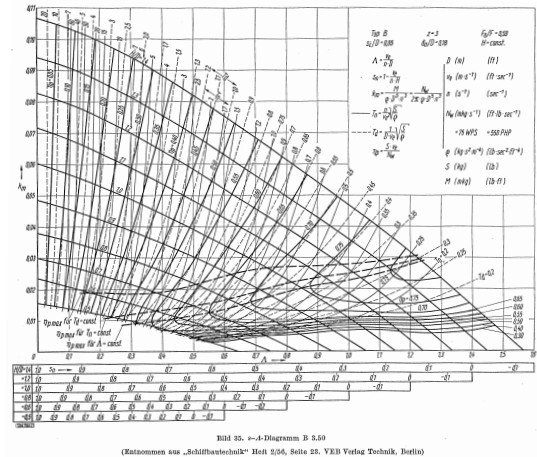

(**a**) P–J diagram, based on the $K_T$–$J$ efficiency map, called $N_w$–$\Lambda$ diagram by Danckwardt.

(**b**) T–J diagram, based on the $K_Q$–$J$ efficiency map, called $s$–$\Lambda$ diagram by Danckwardt.

**Figure A1.** Example of the original Danckwardt diagrams for the propeller B3-50. See Table A1 for the differences between Danckwardt's and ITTC's nomenclature. Reproduced from [6].

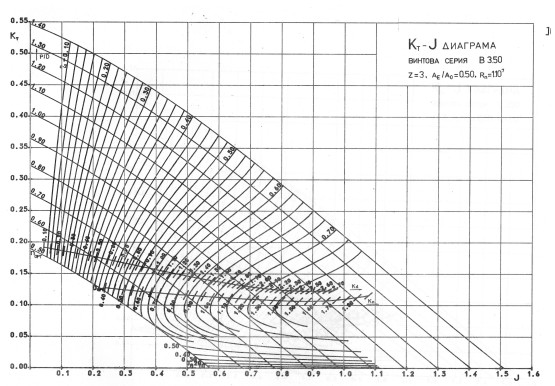

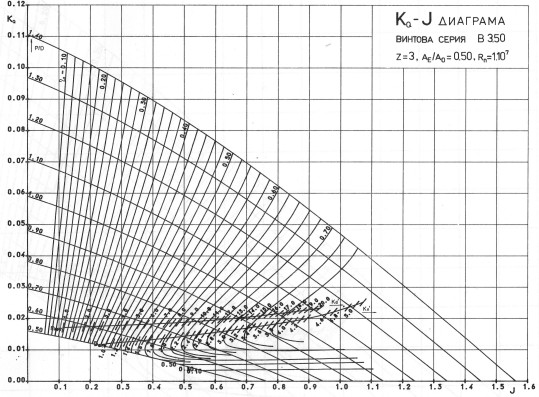

(**a**) Papmel $K_T$–$J$ design chart. Note the use of $K_d = 1/\sqrt{T_D}$ and $K_n = 1/\sqrt[4]{T_n}$.

(**b**) Papmel $K_Q$–$J$ design chart. Note the use of $K'_d = 3.455/\sqrt{P_D}$ and $K'_n = 3.455/\sqrt[4]{P_n}$.

**Figure A2.** Example of the original Papmel design charts for the propeller B3-50, as recreated by Yosifov et al. Reproduced from [14], with permission from Bulgarian Ship Hydrodynamics Centre, 1983.

**Table A1.** Differences between the nomenclature used by Danckwardt and ITTC.

| Name | Danckwardt | ITTC |
|---|---|---|
| Pitch | $H$ | $P$ |
| Blade area ratio | $F_a/F$ | $A_e/A_0$ |
| Speed of advance | $v_e$ | $v_a$ |
| Thrust | $S$ | $T$ |
| Torque | $M$ | $Q$ |
| Delivered power | $N_W$ | $P_D$ |
| Advance coefficient | $\Lambda$ | $J$ |
| Thrust coefficient | $k_s$ | $K_T$ |
| Torque coefficient | $k_m$ | $K_Q$ |
| Open-water efficiency | $\eta_p$ | $\eta_o$ |
| Slip | $s_n$ | $S_R$ |

*Appendix A.3. Advantages of Efficiency Maps*

It was certainly noticed by the reader that the efficiency maps introduced in Section 2 and above are comparable, but not identical, to the $B_p$–$\delta$ diagram. One of the benefits of the efficiency maps is that they include the bollard pull condition, whereas this condition disappears into infinity on the $B_p$–$\delta$ diagram, because $\delta$—as the inverse of the $J$-value— becomes infinity for $J = 0$.

The efficiency maps can also include all six solutions lines for the optimum propeller in one diagram, but such diagrams were never published because they would be even more confusing than the published diagrams with three or four solution lines. In comparison, the $B_p$–$\delta$ diagrams can only be used to solve one single problem. This is certainly another big advantage for the propeller designer.

Compared to the conventional open-water diagrams, the efficiency maps can contain the solutions for real-world optimisation problems, whereas the open-water diagrams can only show the "$\eta_{o,max}$ for $J = $ const" and "$\eta_{o,max}$ for $P/D = $ const" lines (but very seldom do).

*Appendix A.4. How Are Efficiency Maps Used to Optimise a Propeller for Given Conditions?*

The main purpose of using efficiency maps in the scope of this paper is to show the ambiguity of polynomials, as explained in Section 3. Nevertheless, we want to show in this section the practical significance of these diagrams for optimising a propeller.

The optimisation challenges a propeller designer can encounter can be categorised into six basic problems. These are tabulated in Table A2 together with the corresponding solution path using the Danckwardt diagrams. Efficiency maps can be used to solve any of these problems in a direct way without the need to resort to an iterative process.

**Table A2.** The six basic optimisation problems encountered by propeller designers and their solution paths. All unknown values can finally be calculated from the solution.

| | Problem Definition | | Solution Path | | | |
|---|---|---|---|---|---|---|
| N° | Known Values | Unknown Values | Calculate … | …Using Equation | To Find Optimum Use Line "$\eta_{o,max}$ for … = const" | Solution |
| 1 | $v_a, n, D$ | $P/D$ | $J$ | (1) | $J$ | $P/D, K_T, K_Q$ |
| 2 | $D, P$ | $v_a, n$ | $P/D$ | — | $P/D$ | $J, K_T, K_Q$ |
| 3 | $T, v_a, D$ | $n, P/D$ | $T_D$ | (A1) | $T_D$ | $J, P/D, K_Q$ |
| 4 | $Q, v_a, D$ | $n, P/D$ | $P_D$ | (A4) | $P_D$ | $J, P/D, K_T$ |
| 5 | $T, v_a, n$ | $D, P/D$ | $T_n$ | (A5) | $T_n$ | $J, P/D, K_Q$ |
| 6 | $Q, v_a, n$ | $D, P/D$ | $P_n$ | (A6) | $P_n$ | $J, P/D, K_T$ |

We will explain the solution path using the "Example 1: Optimum rotation rate for a given diameter" from Kuiper's book "The Wageningen Propeller Series" [15]. Kuiper presents the problem, where the propeller thrust $T$ (1393 kN), the propeller diameter $D$ (7 m), and the advance speed $v_a$ (16.8 kn) are given. The density of water $\rho$ is assumed to be 1025 kg m$^{-3}$. These values were typical for a container vessel of that time with a speed of 21 kn. He also assumes that a four-bladed propeller has been chosen. The task at hand is to find the optimum propeller, the required power $P_P$, and especially the shaft speed $n$. Kuiper calculates the required blade area ratio $A_E/A_0$ to be 0.48. For the optimisation process, he selects a blade area ratio of 0.55 to agree with the diagrams published.

Having established the basic conditions of the optimisation problem, he explains that "the thrust and the diameter are known, but the rotation rate is not. This means that the parameters $K_T$ and $J$ cannot be calculated yet. However, the parameter $K_T/J^2$ can be calculated because it does not contain the rotation rate (as we already have seen in Appendix A.1.3, where we called this parameter $T_D$). Using the figures above in Equation (A1), he gets a value of 0.3707. Kuiper now starts the program supplied with the book and searches the $P/D$ value for the fixed $K_T/J^2$ value of 0.3707, where the open-water

efficiency $\eta_o$ becomes a maximum. Stopping at an accuracy of $^5/_{100}$ for $^P/_D$, he obtains the values given at iteration 2 of Table A3.

**Table A3.** The solutions for "Example 1" in Kuiper's book [15], as obtained by Kuiper using the program supplied with the book by using Danckwardt diagrams and the exact solution.

|  | **Iteration** | $^P/_D$ | $J$ | $K_T$ | $K_Q$ | $\eta_o$ | $T_D$ |
|---|---|---|---|---|---|---|---|
|  | 1 | 0.95 | 0.674 | 0.168 | 0.0278 | 0.650 | 0.371 |
| Kuiper | 2 | 1.00 | 0.699 | 0.181 | 0.0310 | 0.651 | 0.371 |
|  | 3 | 1.05 | 0.723 | 0.194 | 0.0344 | 0.650 | 0.371 |
| Danckwardt diagram |  | 1.00 | 0.70 | 0.18 | 0.031 | 0.65 | 0.371 |
| Exact solution |  | 1.004 | 0.7007 | 0.1823 | 0.031 24 | 0.6509 | 0.371 310 |

Kuiper states that "the conclusion is that the optimum efficiency can be reached with a pitch ratio of 1.0". From the advance ratio $J$, the required shaft speed is derived as $1.767\,\mathrm{s^{-1}}$ (Equation (1)) and the required power from the torque coefficient $K_T$ as $18\,513\,\mathrm{kW}$ (Equation (3)).

Using the Danckwardt diagrams to solve this optimisation problem follows the same path, but instead of finding the optimum by manually searching for the maximum efficiency, we use the T–J diagram for the propeller B4-55 (see Appendix B). We pencil the line for $T_D = 0.371$ between the thin blue lines for $T_D = 0.25$ and $T_D = 0.5$ and find its intersection with the thick blue line "$\eta_{o,max}$ for $T_D = \mathrm{const}$". The values are given in line 4 of Table A3. The unknown values for $n$ and $P_P$ are calculated as before and we get the same figures.

For comparison, the exact solution is given in the last line of Table A3 with an accuracy of $^1/_{1000}$ for $^P/_D$.

The other design challenges from Table A2 are solved accordingly.

**Appendix B. Efficiency Maps for Wageningen B-Screw Series**

The following pages contain the recreated Danckwardt diagrams of all propellers of the Wageningen B-screw Series, according to Table 1. They are valid for a sectional Reynolds number of $2 \cdot 10^6$. The diagrams are based on the polynomials published in 1975 by Oosterveld and van Oossanen [2]. Table A4 explains the composition of the diagrams and the line colours and types used.

**Table A4.** Composition of the Danckwardt P–J ($K_T$–J) and T–J ($K_Q$–J) efficiency maps. Additionally shown is the significations of line colour and type. $K_T$ = thrust coefficient; $K_Q$ = torque coefficient; $\eta_o$ = open-water efficiency; $J$ = advance coefficient; $T$ = thrust; $Q$ = torque; $P_P$ = propeller power; $D$ = propeller diameter; $P$ = propeller pitch; $n$ = shaft speed (in $s^{-1}$); $v_a$ = speed of advance; and $\rho$ = water density. If not stated otherwise, all values are in SI base units.

| **Diagram:** | **P–J** | **T–J** |
|---|---|---|
| Abscissa: | $J = \dfrac{v_a}{nD}$ | $J = \dfrac{v_a}{nD}$ |
| Ordinate: | $K_T = \dfrac{T}{\rho n^2 D^4}$ | $K_Q = \dfrac{Q}{\rho n^2 D^5} = \dfrac{P_P}{2\pi\rho n^3 D^5}$ |
| One set of curves: | — $K_T(J)$ for $P/D = \text{const}$ | — $K_Q(J)$ for $P/D = \text{const}$ |
| Two sets of contour lines: | — $\eta_o = \dfrac{v_a}{nD} = \text{const}$  <br> ··· $K_Q = \dfrac{Q}{\rho n^2 D^5} = \dfrac{P_P}{2\pi\rho n^3 D^5} = \text{const}$ | — $\eta_o = \dfrac{v_a}{nD} = \text{const}$  <br> ··· $K_T = \dfrac{T}{\rho n^2 D^4} = \text{const}$ |
| Two families of (parametric) curves for sets of constant values: | — $P_D = \dfrac{K_Q}{J^3} = \dfrac{1}{D^2 v_a^2}\dfrac{Qn}{\rho v_a} = $ <br> $= \dfrac{1}{D^2 v_a^2}\dfrac{P_P}{2\pi\rho v_a} = \text{const}$  <br>  — $P_n = \dfrac{K_Q}{J^5} = \dfrac{n}{v_a^4}\dfrac{Qn}{\rho v_a} = $ <br> $= \dfrac{n}{v_a^4}\dfrac{P_P}{2\pi\rho v_a} = \text{const}$ | — $T_D = \dfrac{K_T}{J^2} = \dfrac{1}{D^2 v_a^2}\dfrac{T}{\rho} = \text{const}$  <br><br><br>  — $T_n = \dfrac{K_T}{J^4} = \dfrac{n}{v_a^4}\dfrac{T}{\rho} = \text{const}$ |
| Four lines of $\eta_{o,max}$: | — "for $P_D = \textbf{const}$" (for known $Q$, $v_a$, $D$) <br> — "for $P_n = \textbf{const}$" (for known $Q$, $v_a$, $n$) <br> — "for $J = \textbf{const}$" (for known $v_a$, $n$, $D$) <br> — "for $P/D = \textbf{const}$" (for known $P/D$) | — "for $T_D = \textbf{const}$" (for known $T$, $v_a$, $D$) <br> — "for $T_n = \textbf{const}$" (for known $T$, $v_a$, $n$) <br> — "for $J = \textbf{const}$" (for known $v_a$, $n$, $D$) <br> — "for $P/D = \textbf{const}$" (for known $P/D$) |

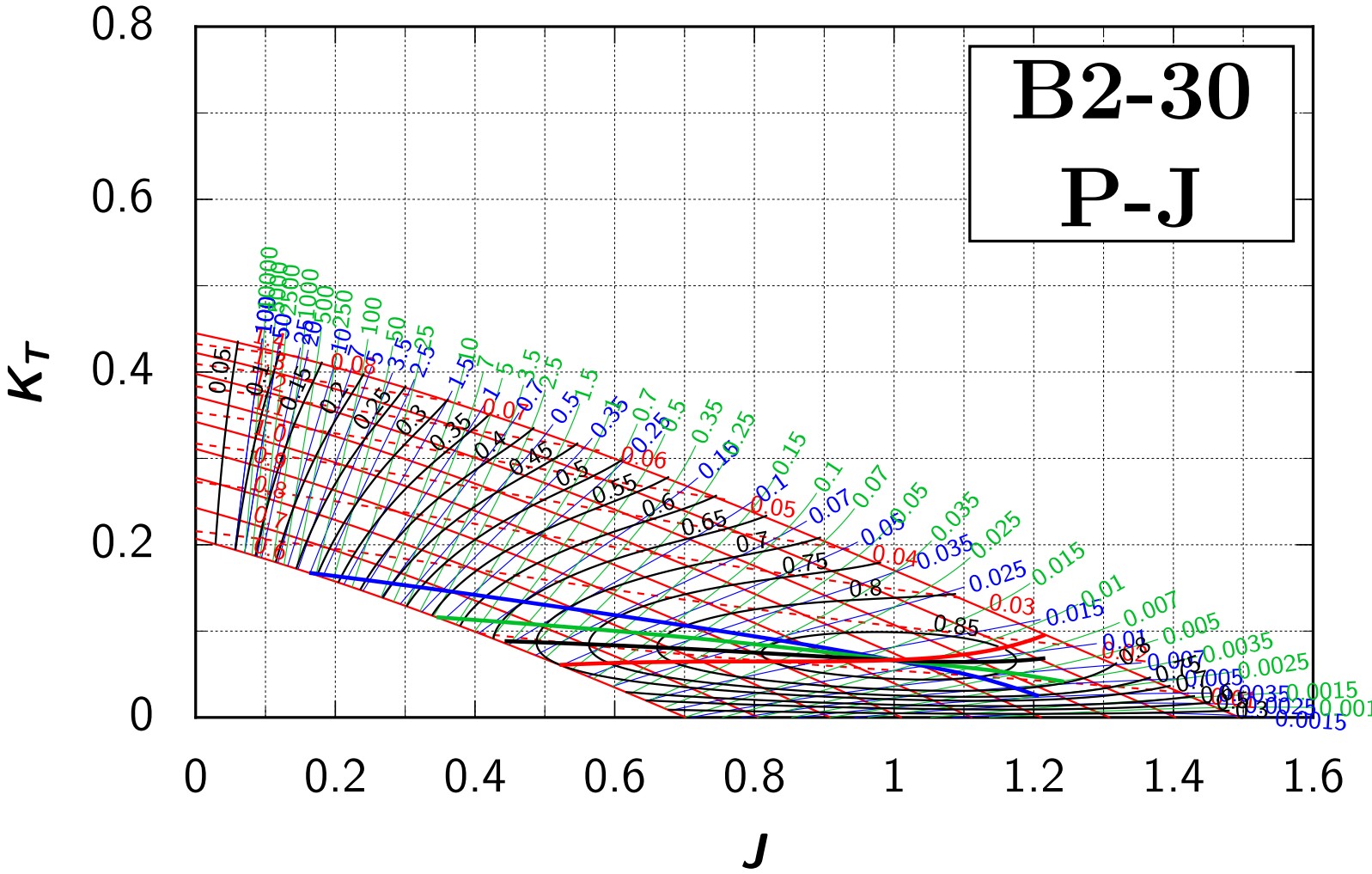

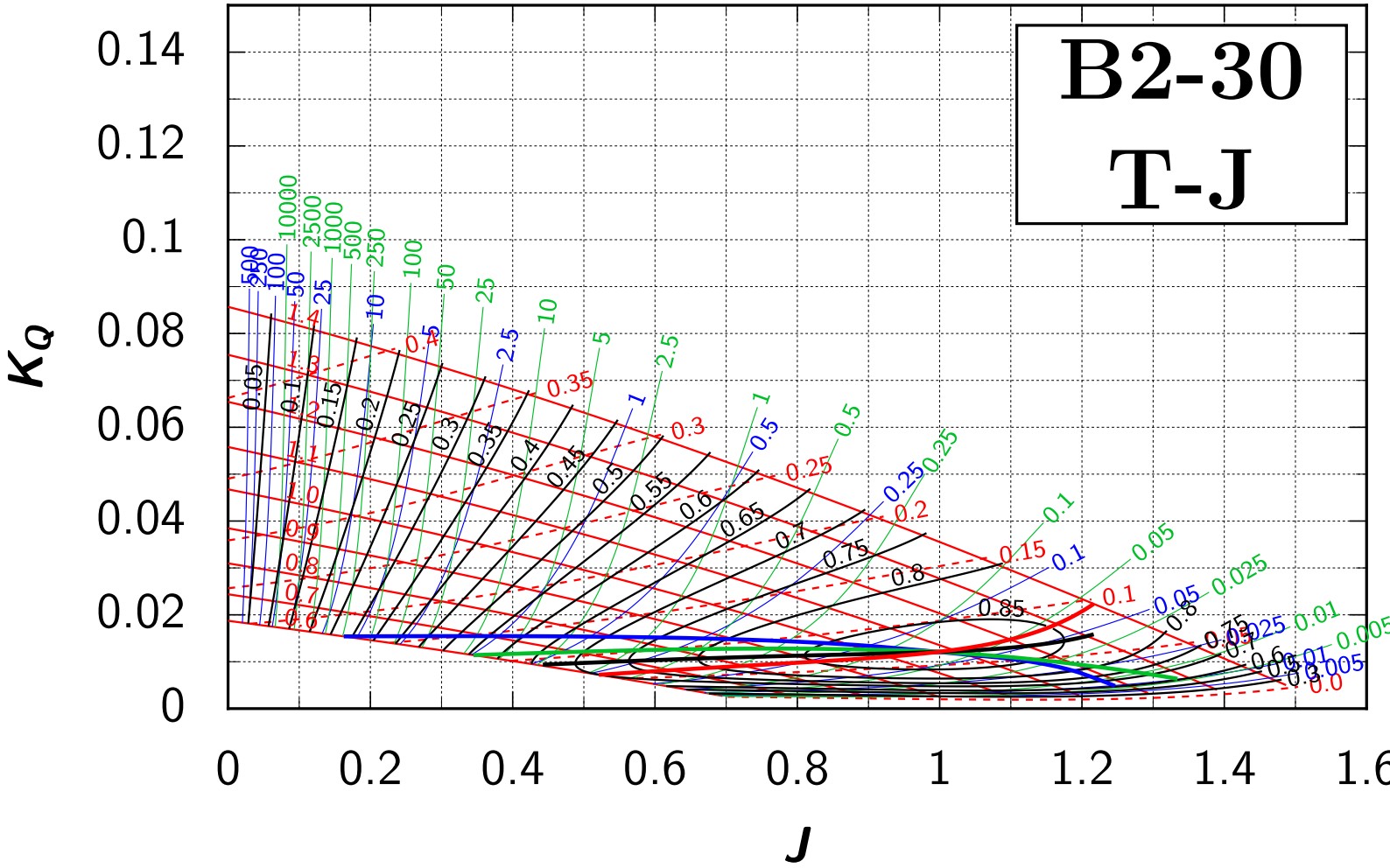

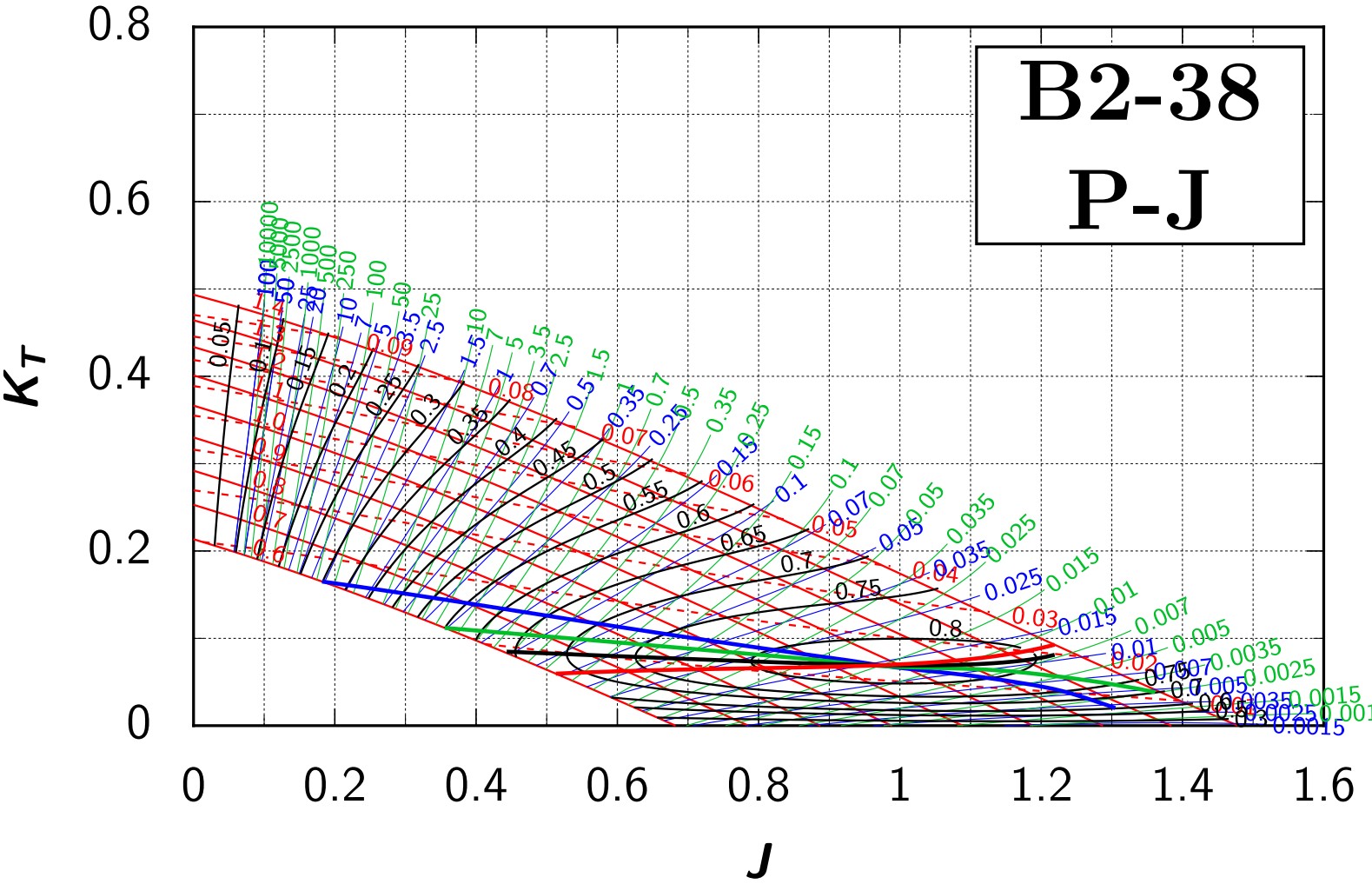

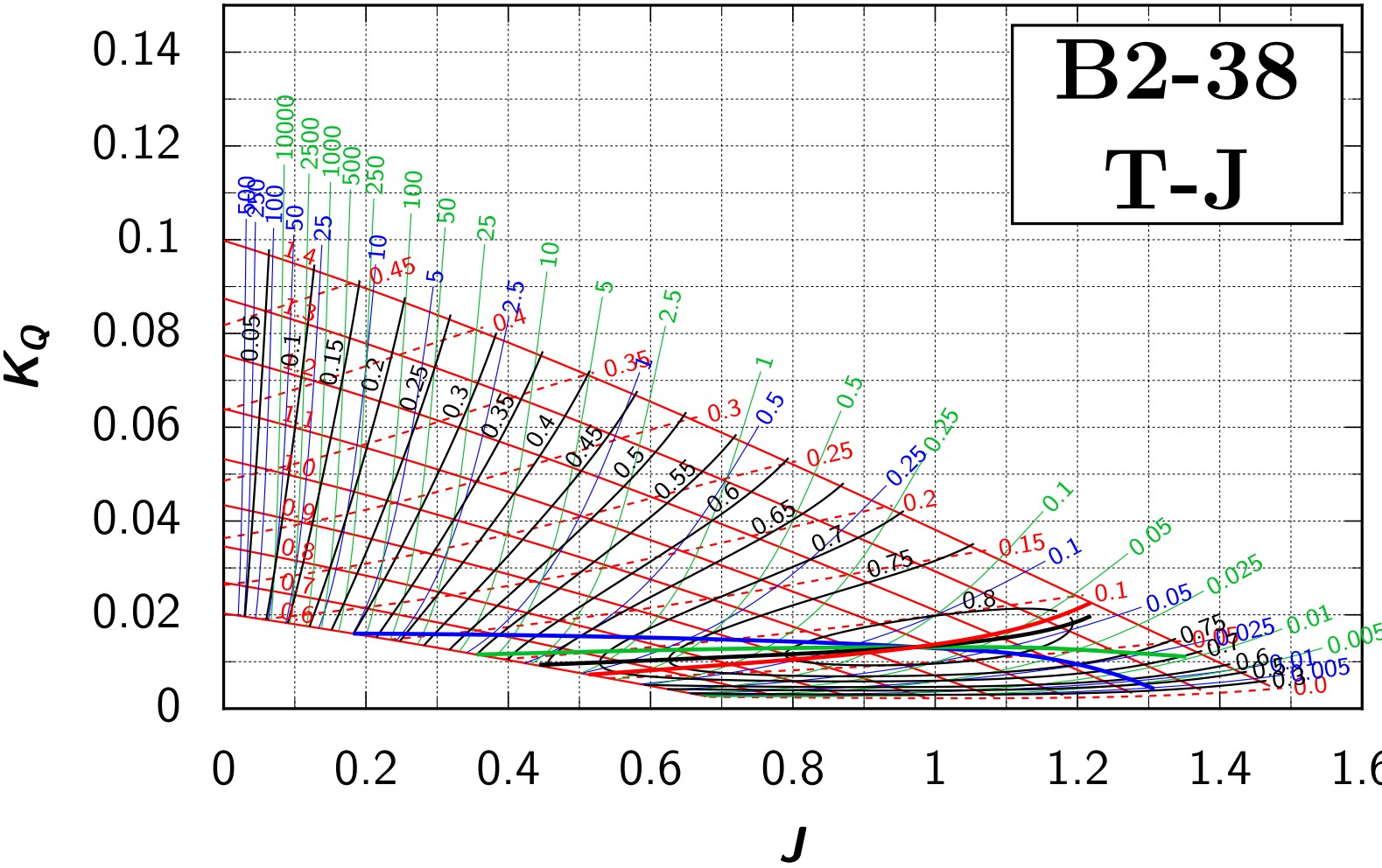

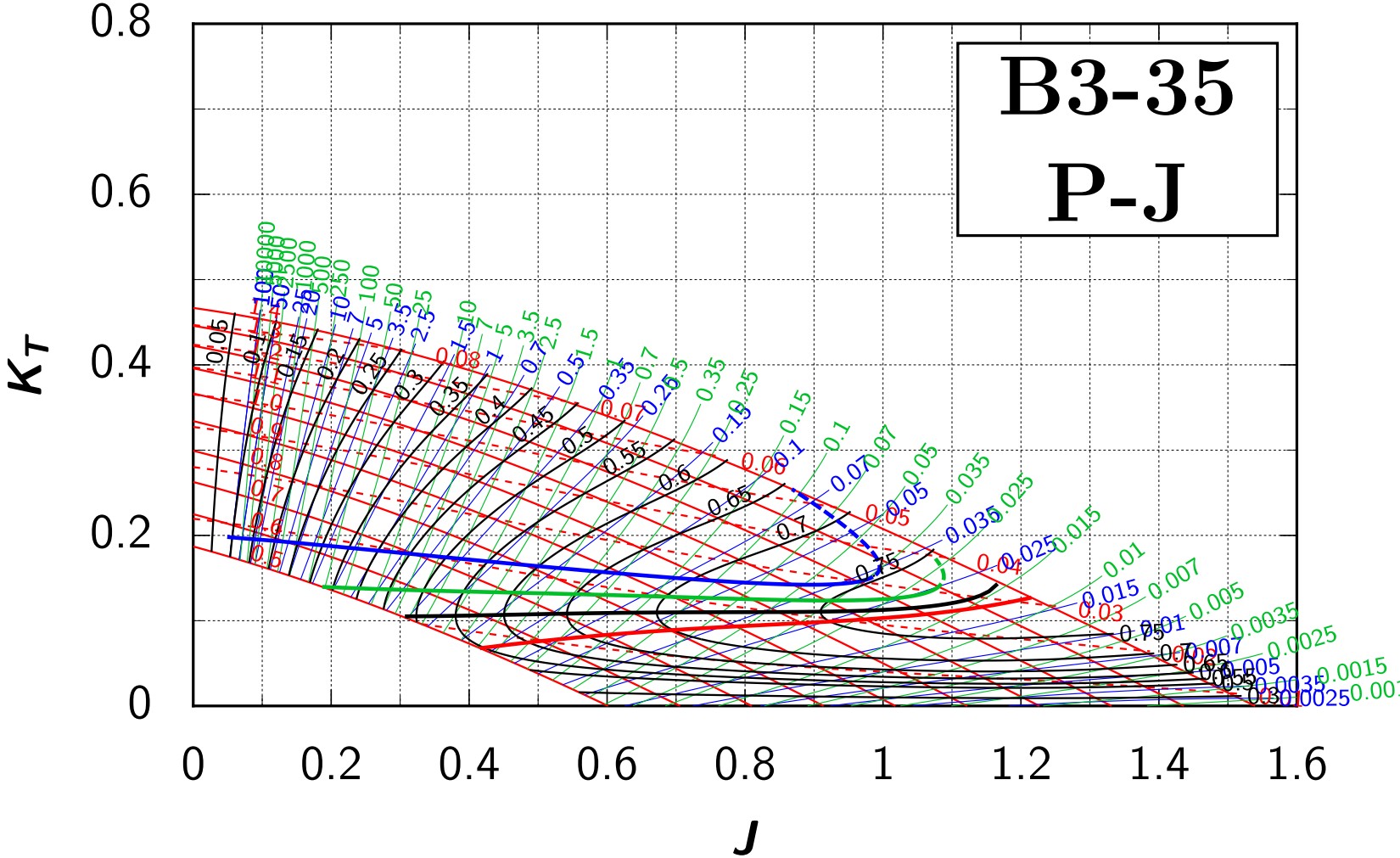

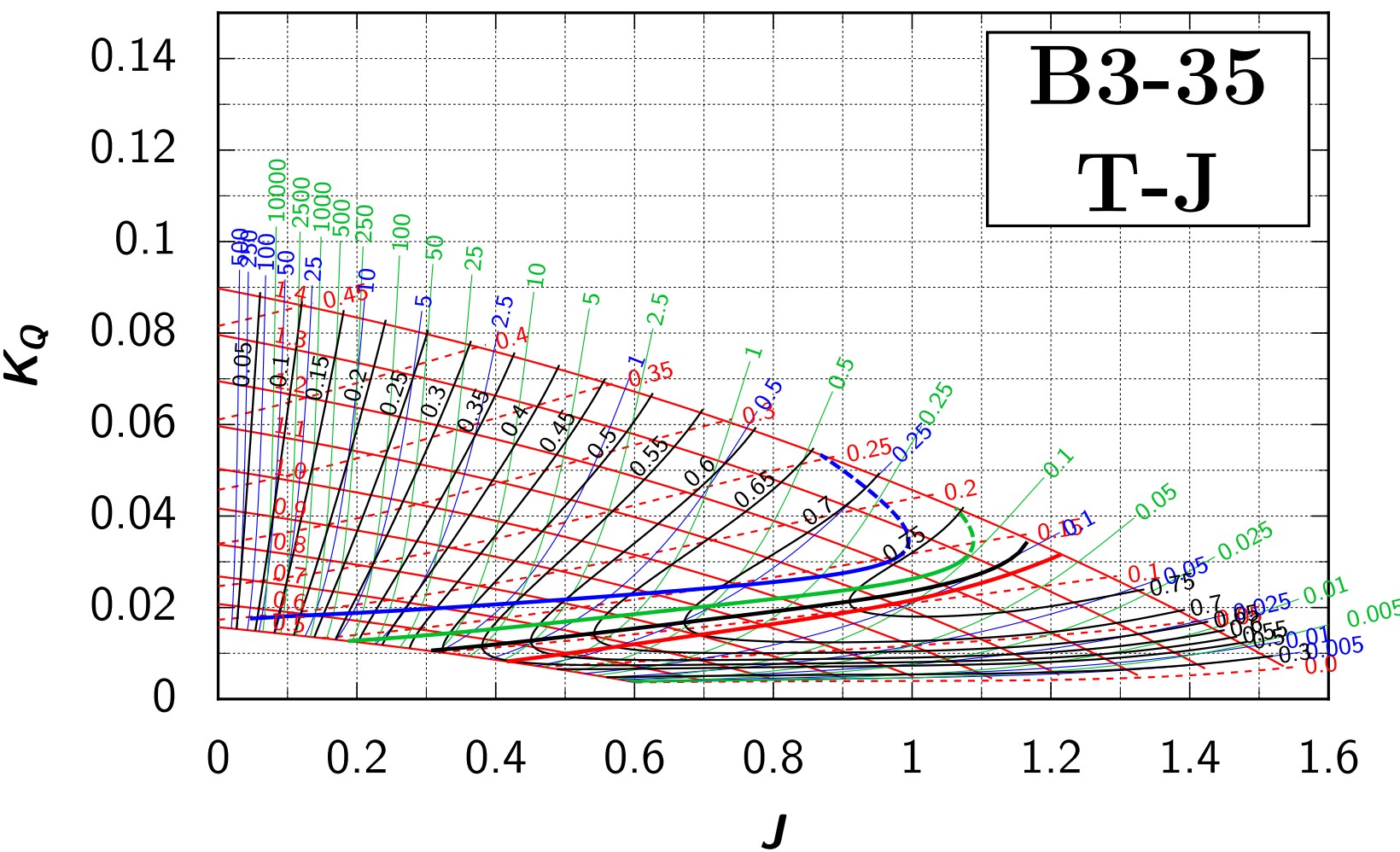

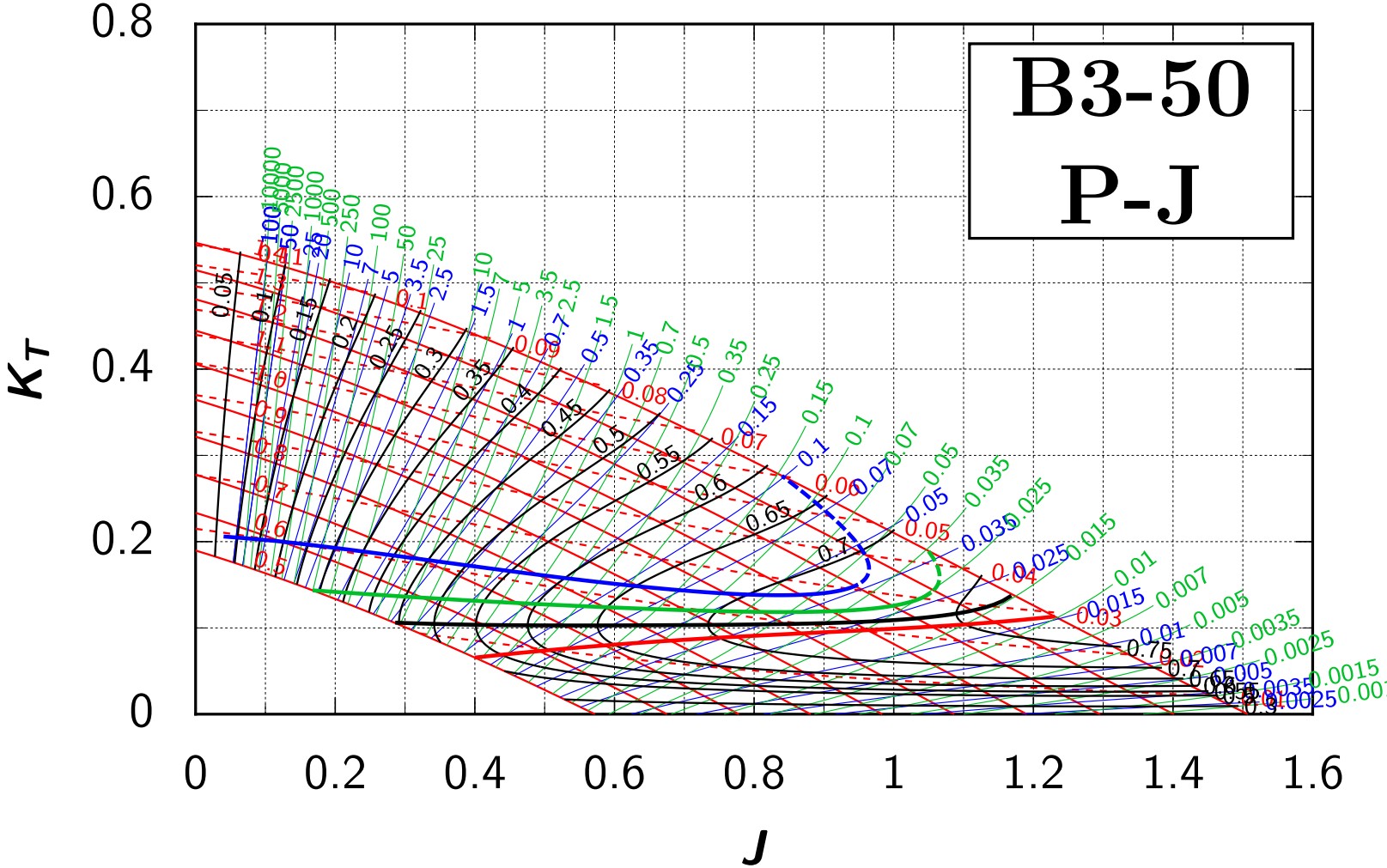

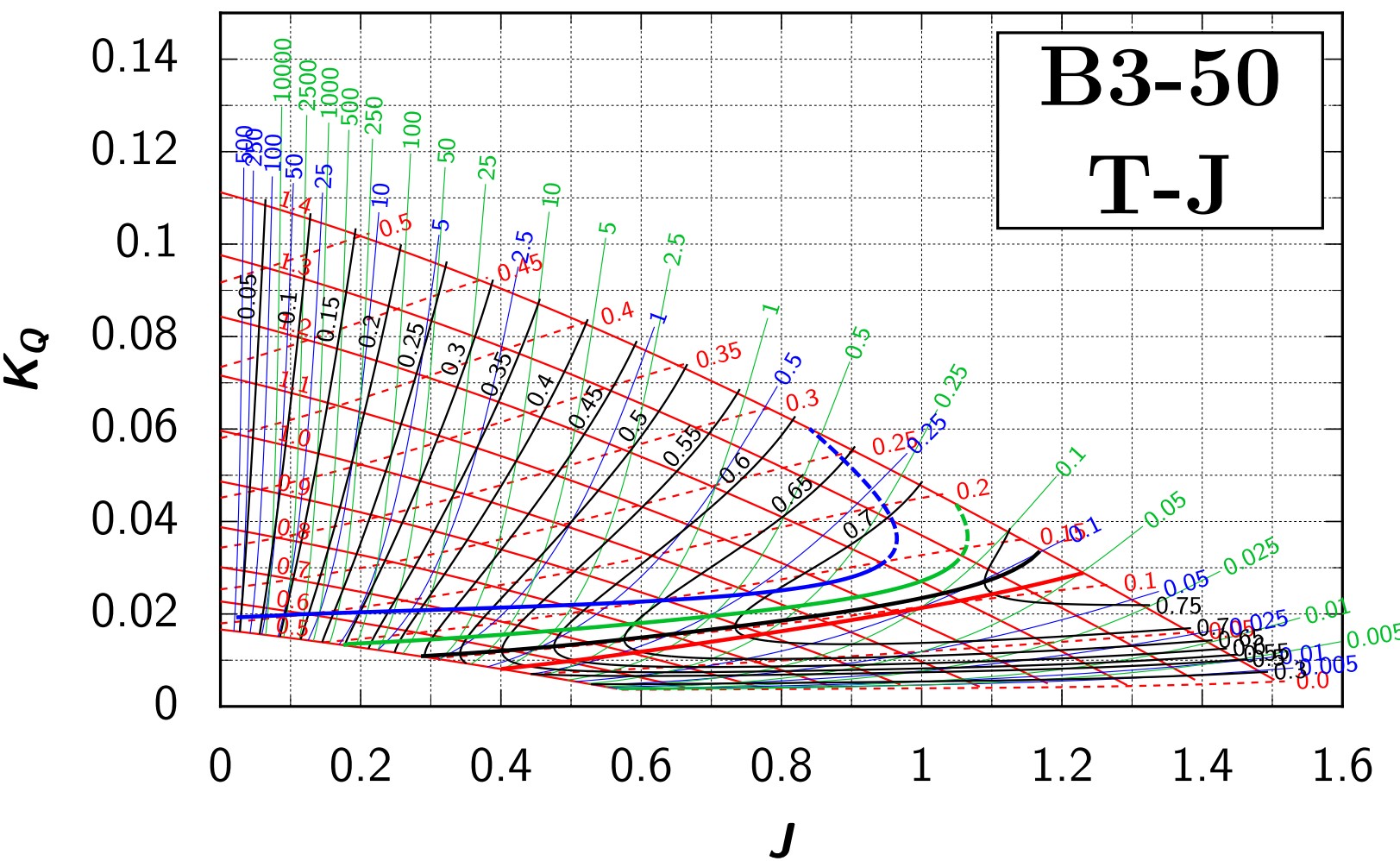

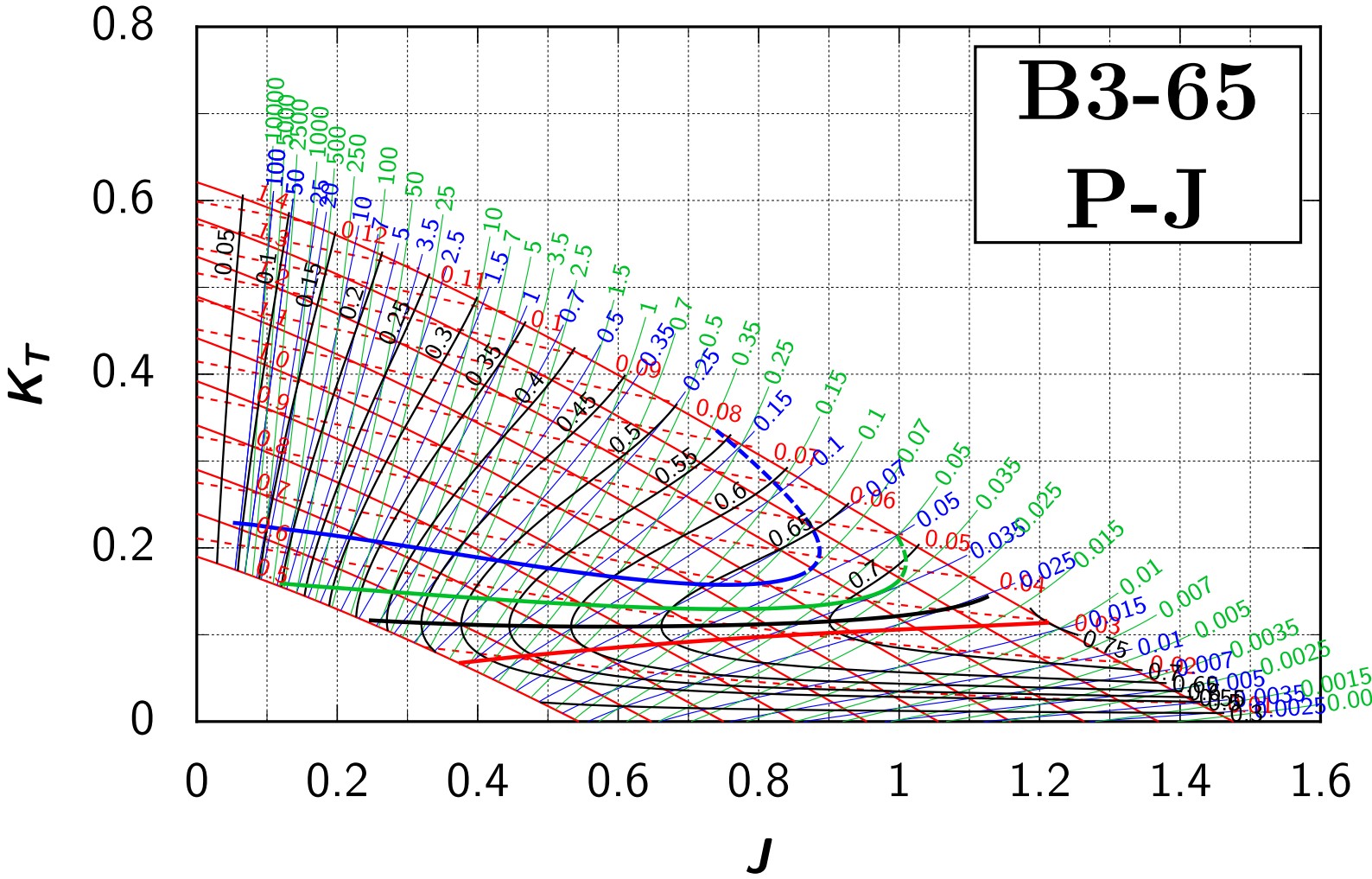

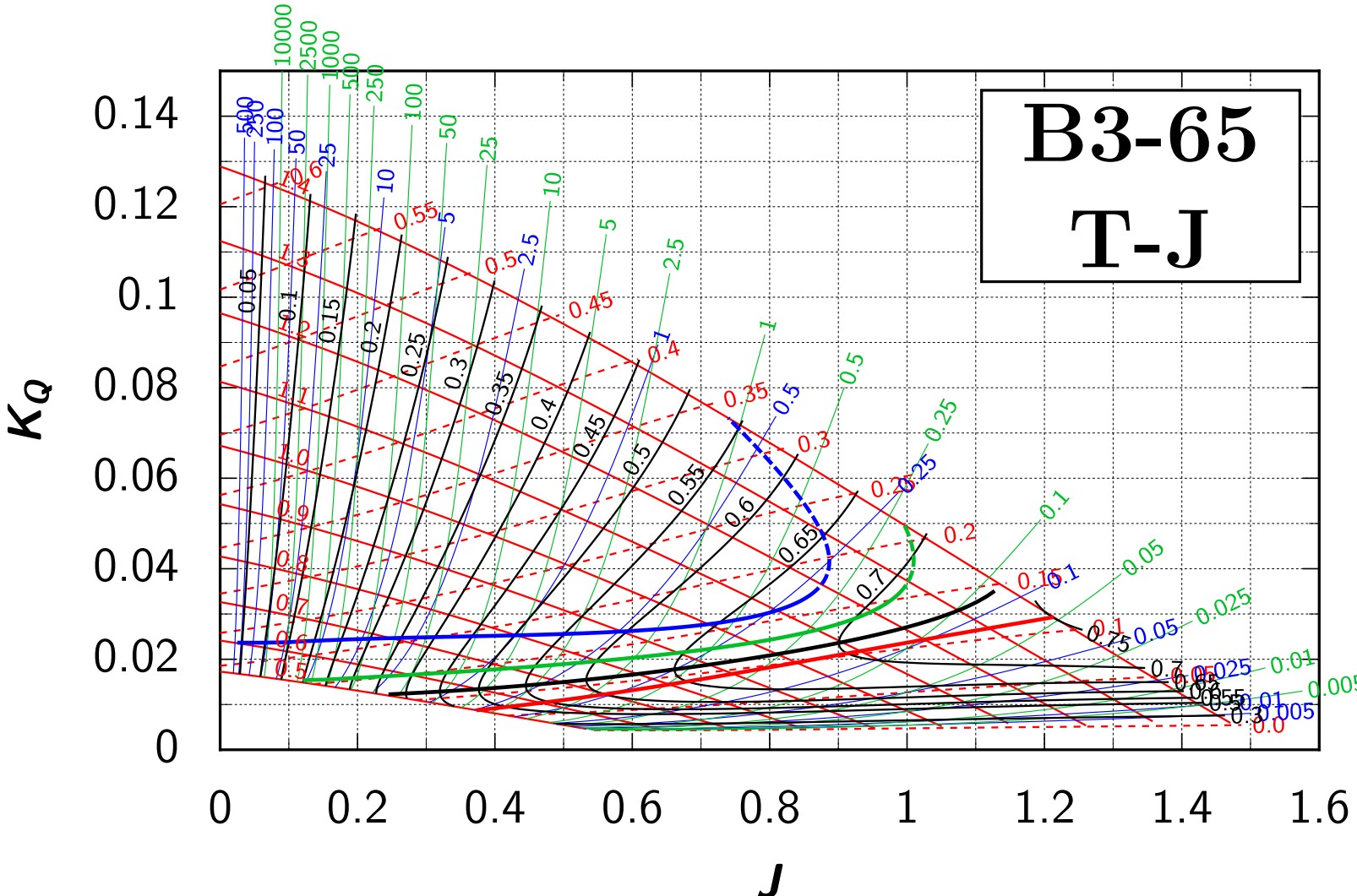

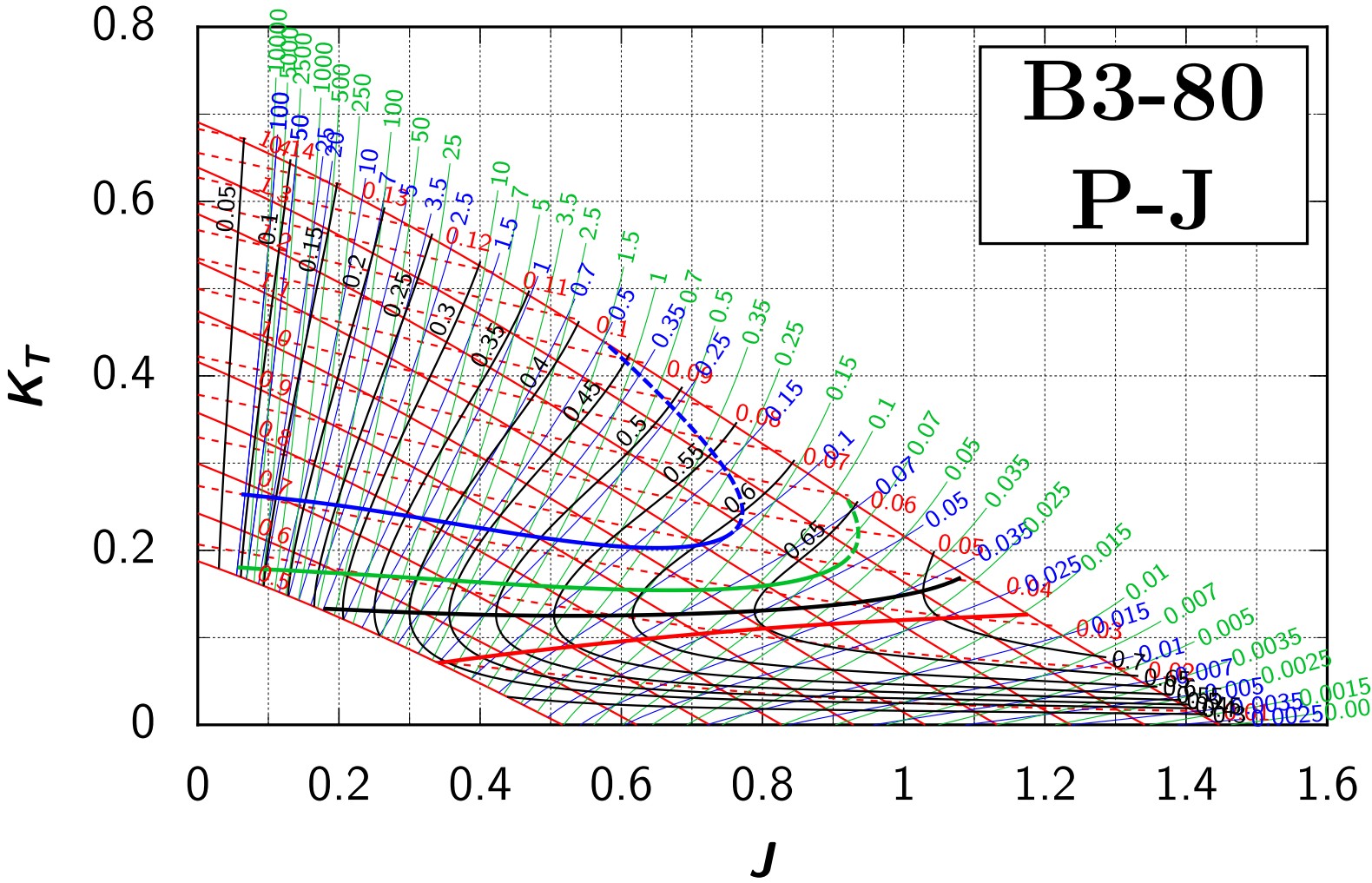

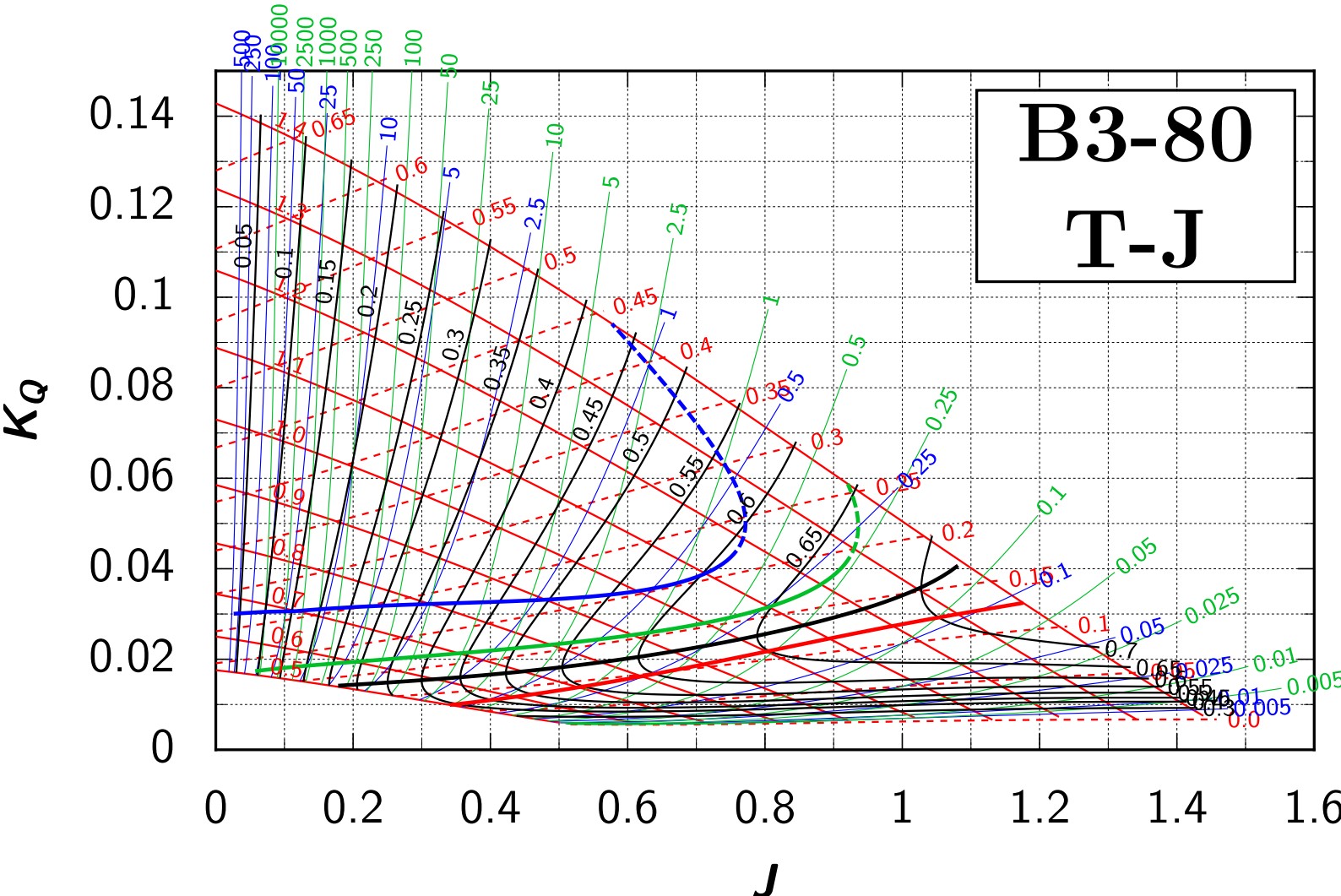

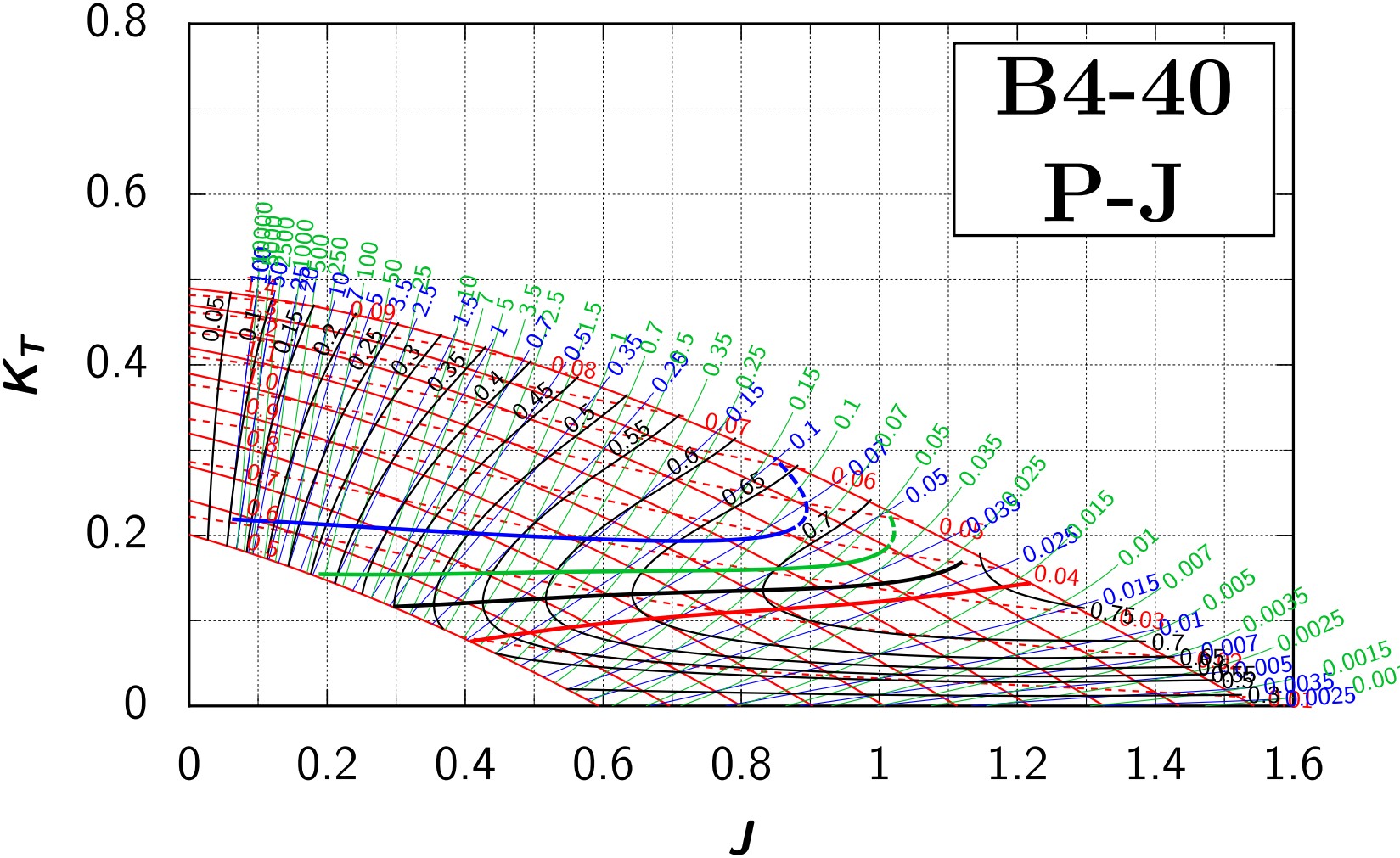

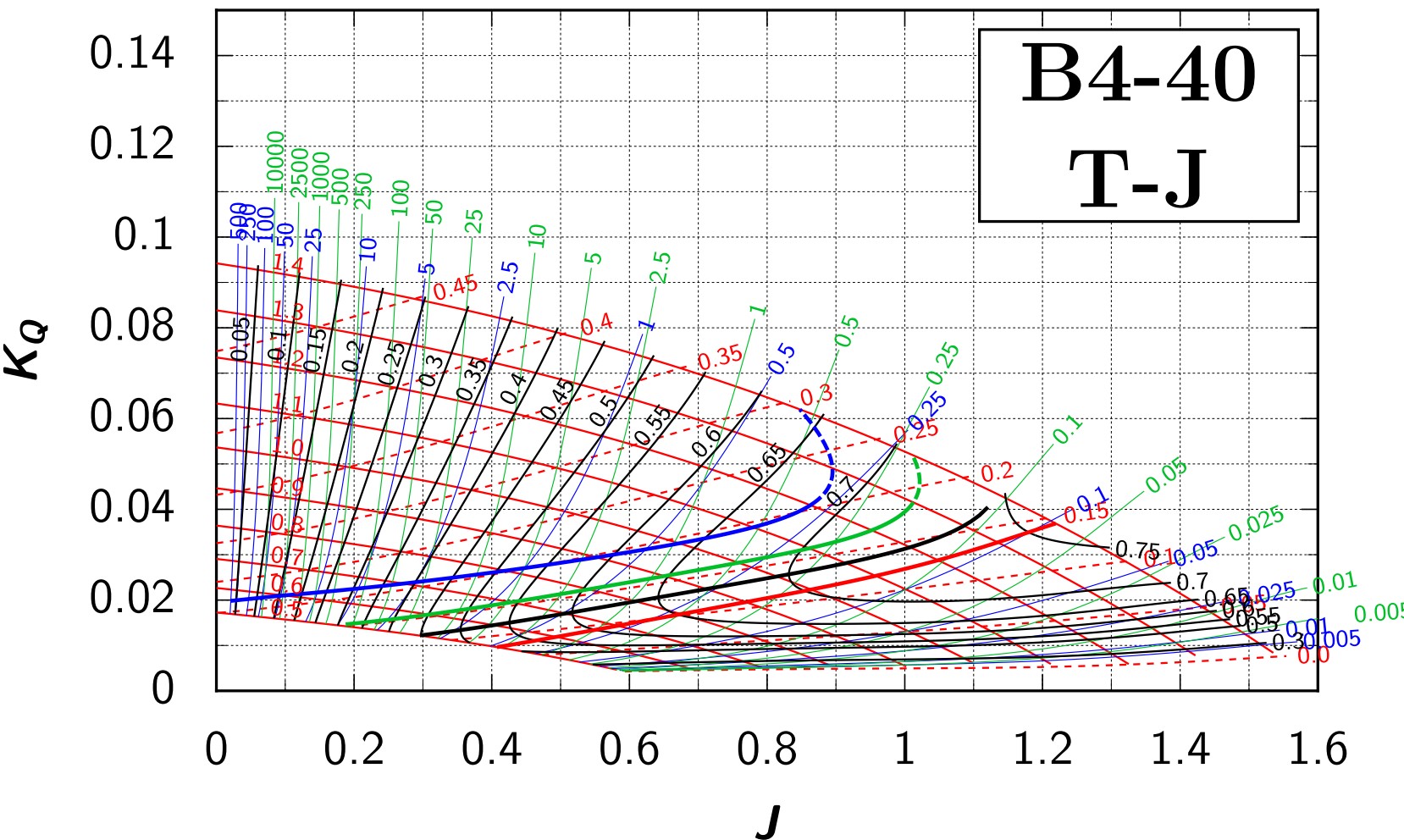

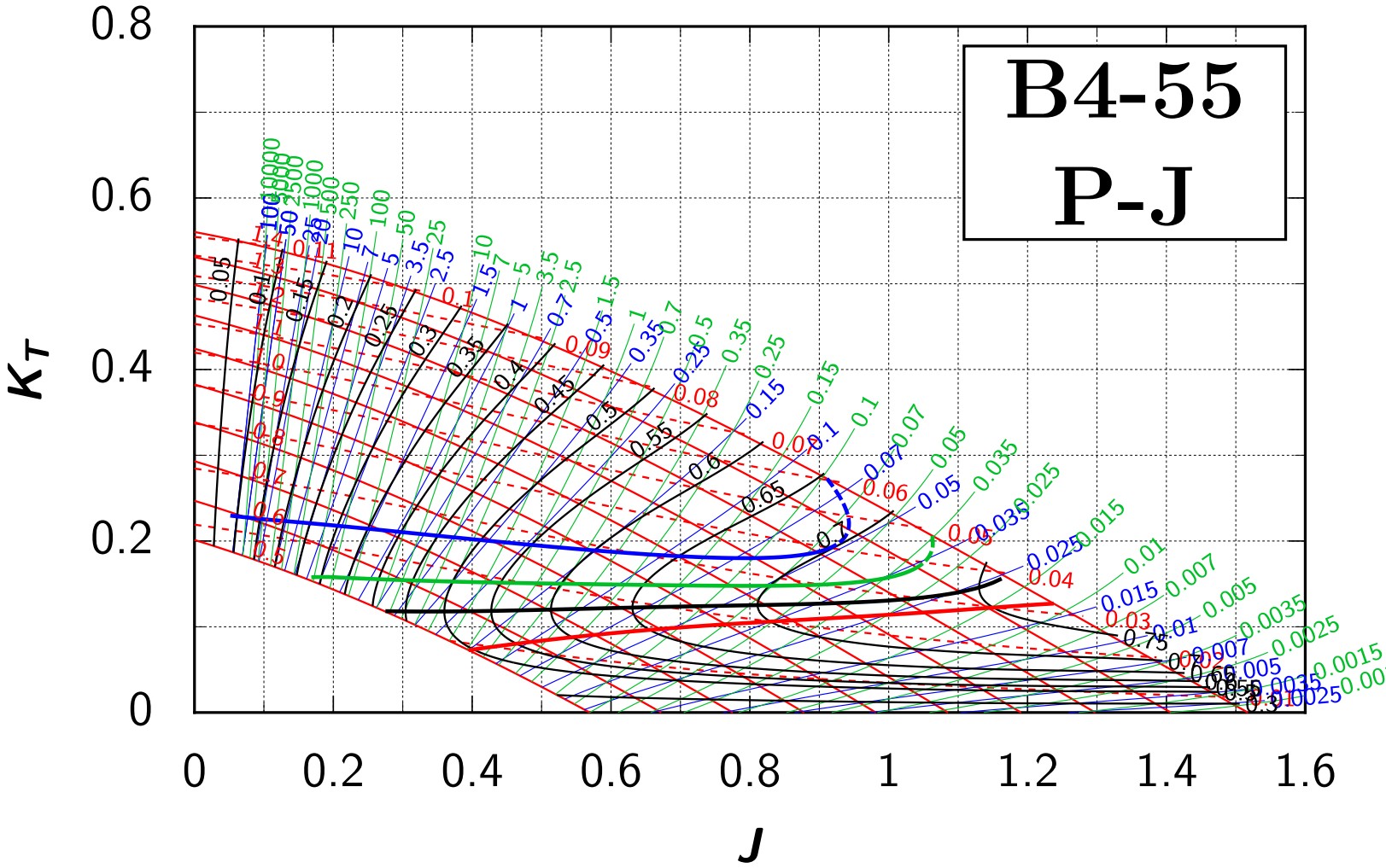

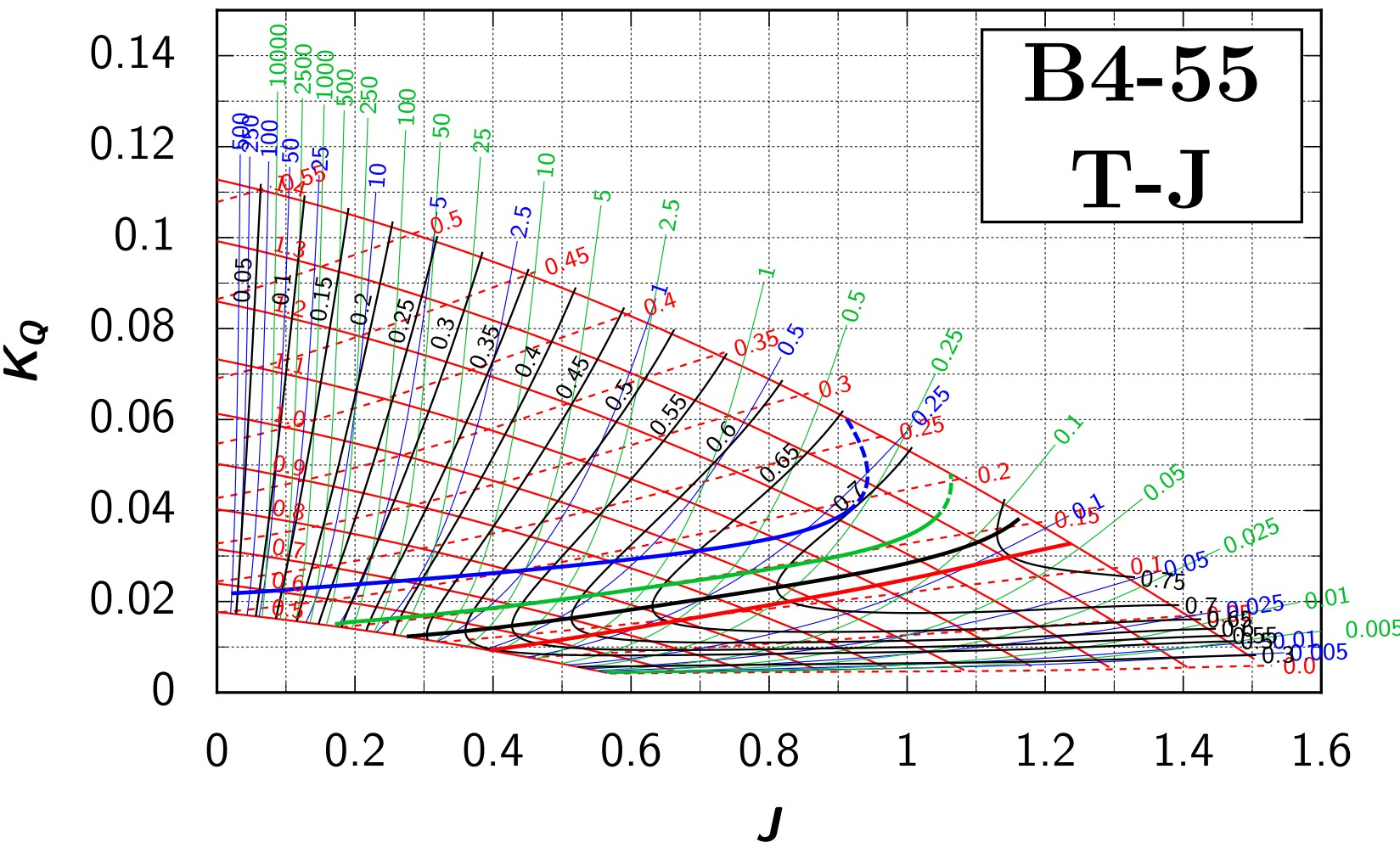

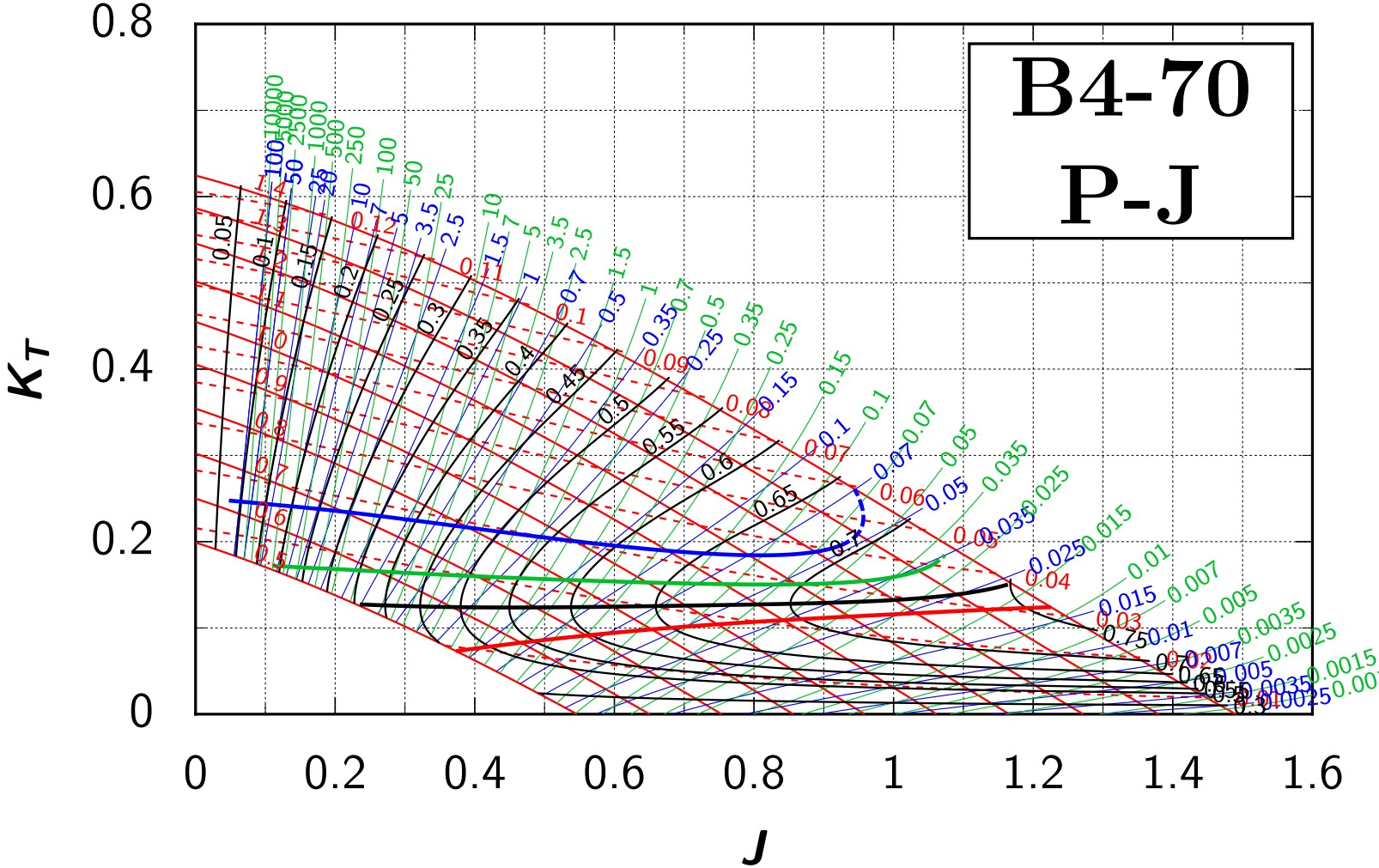

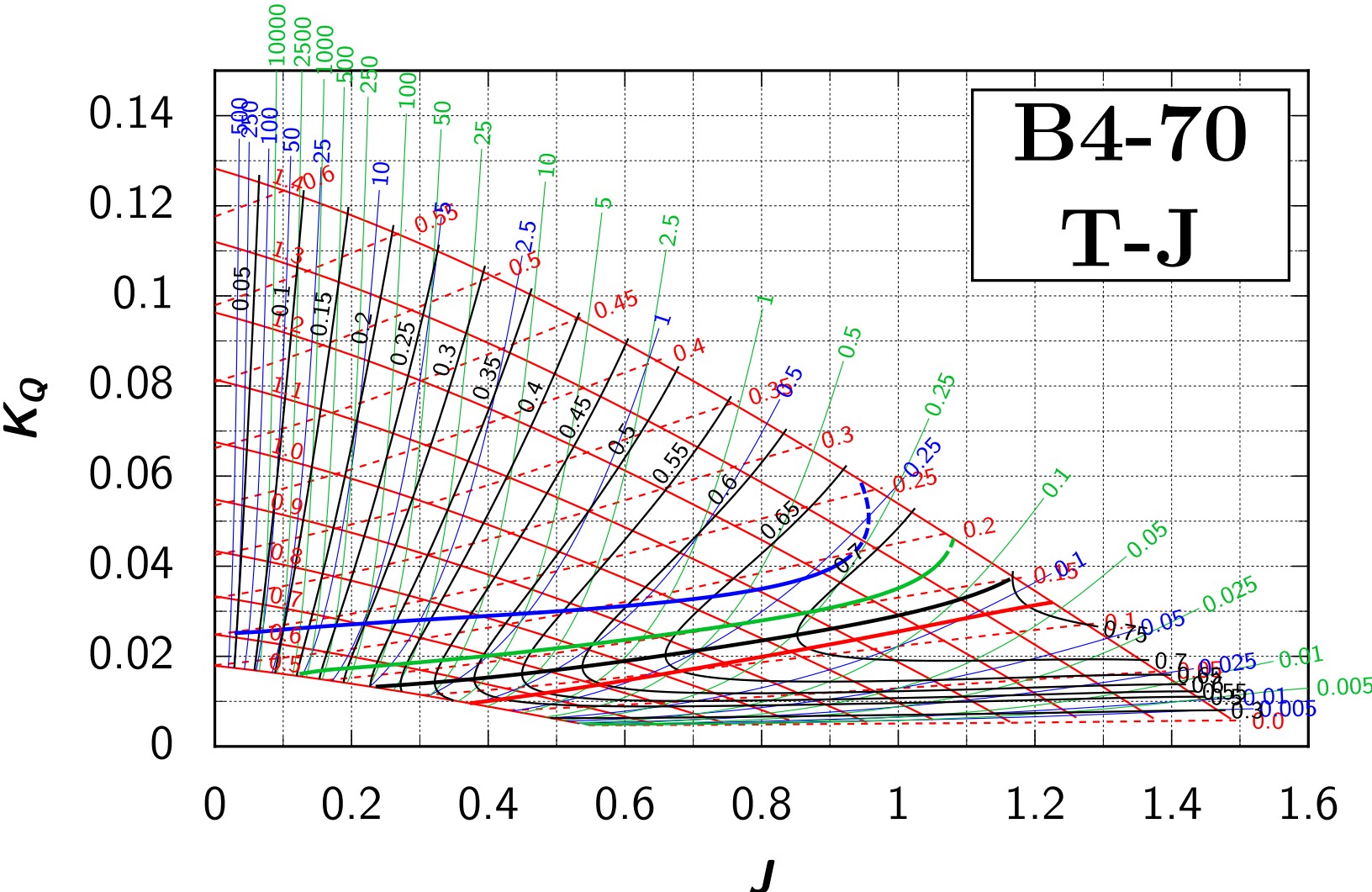

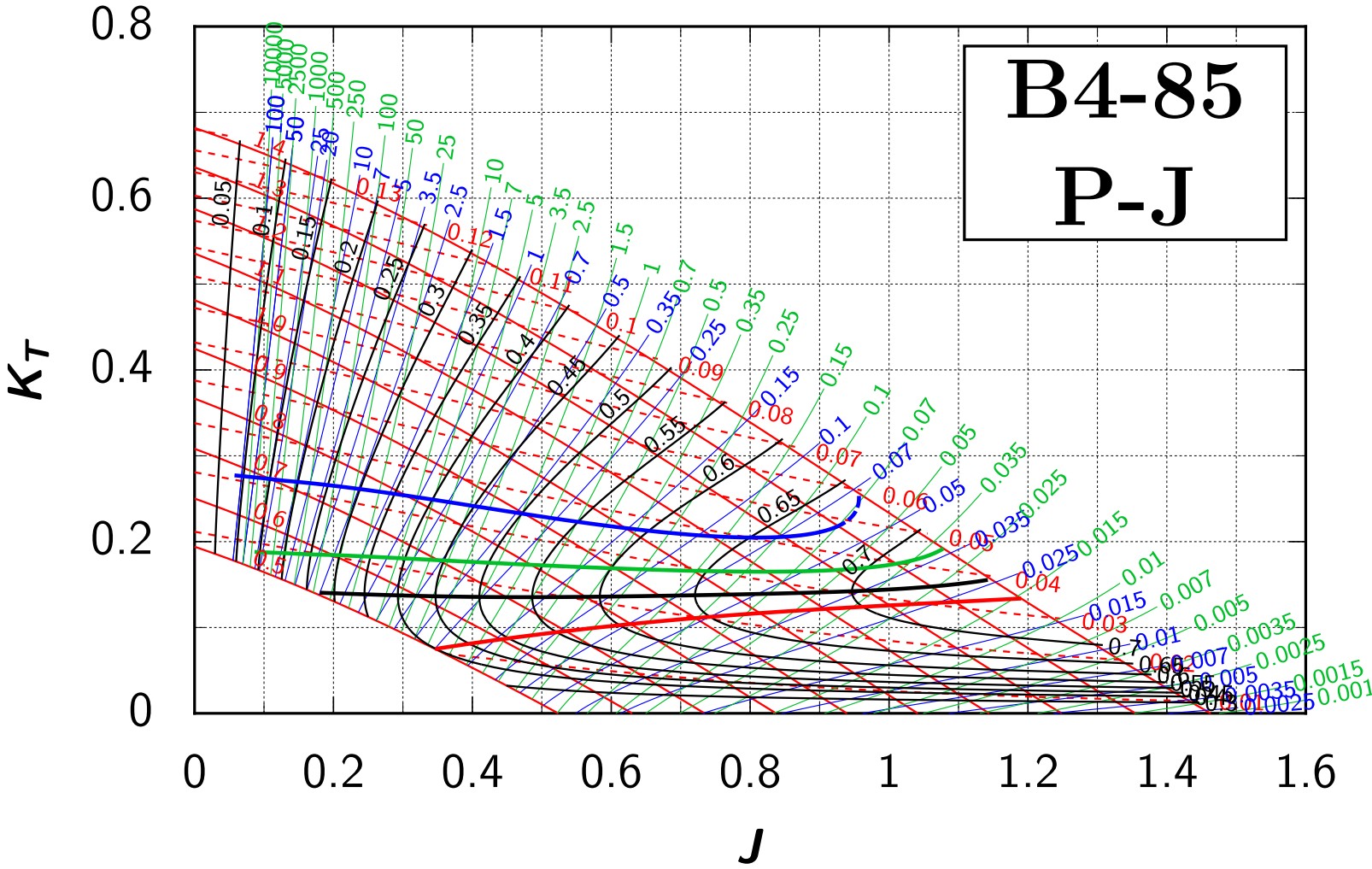

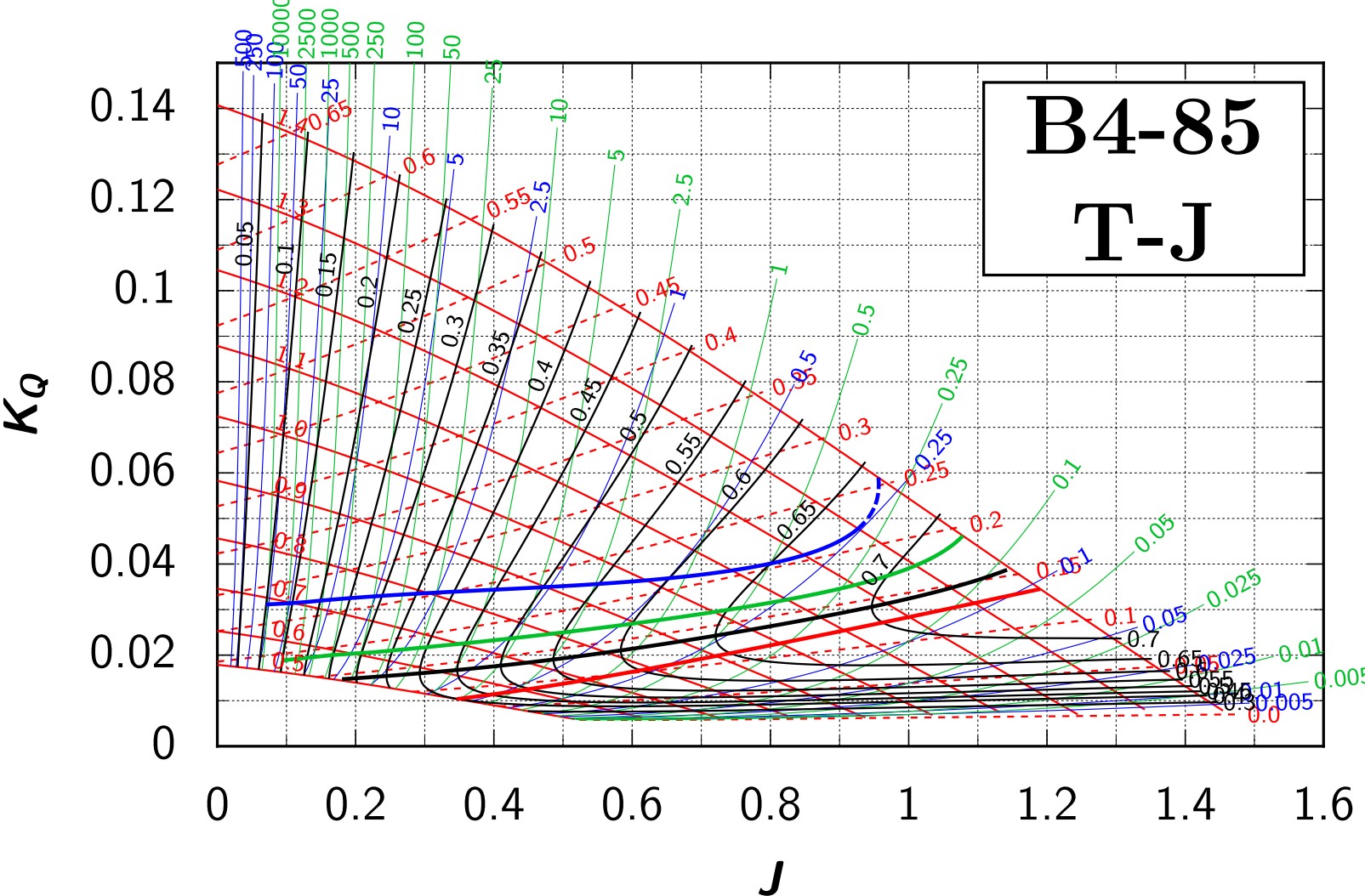

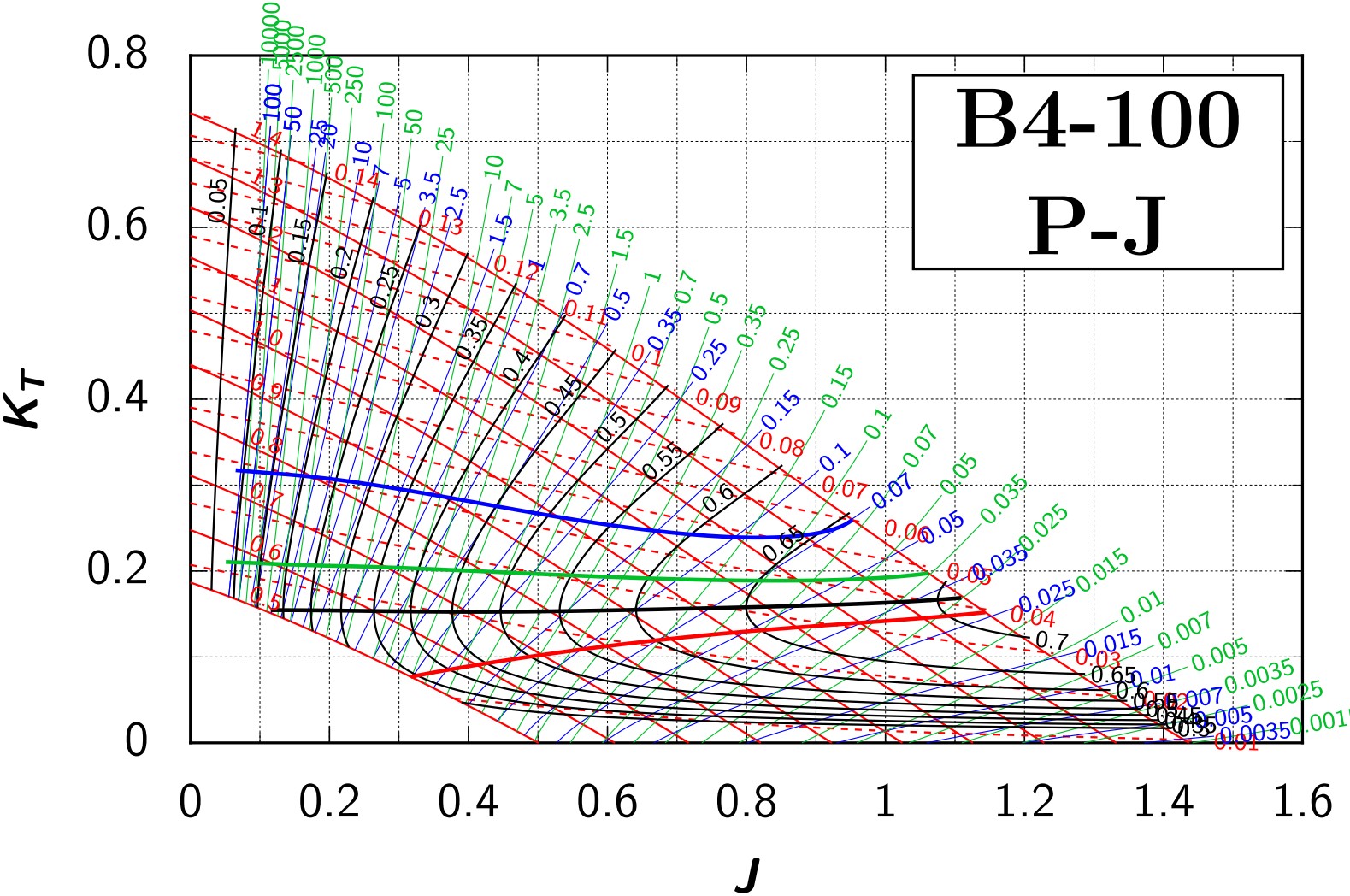

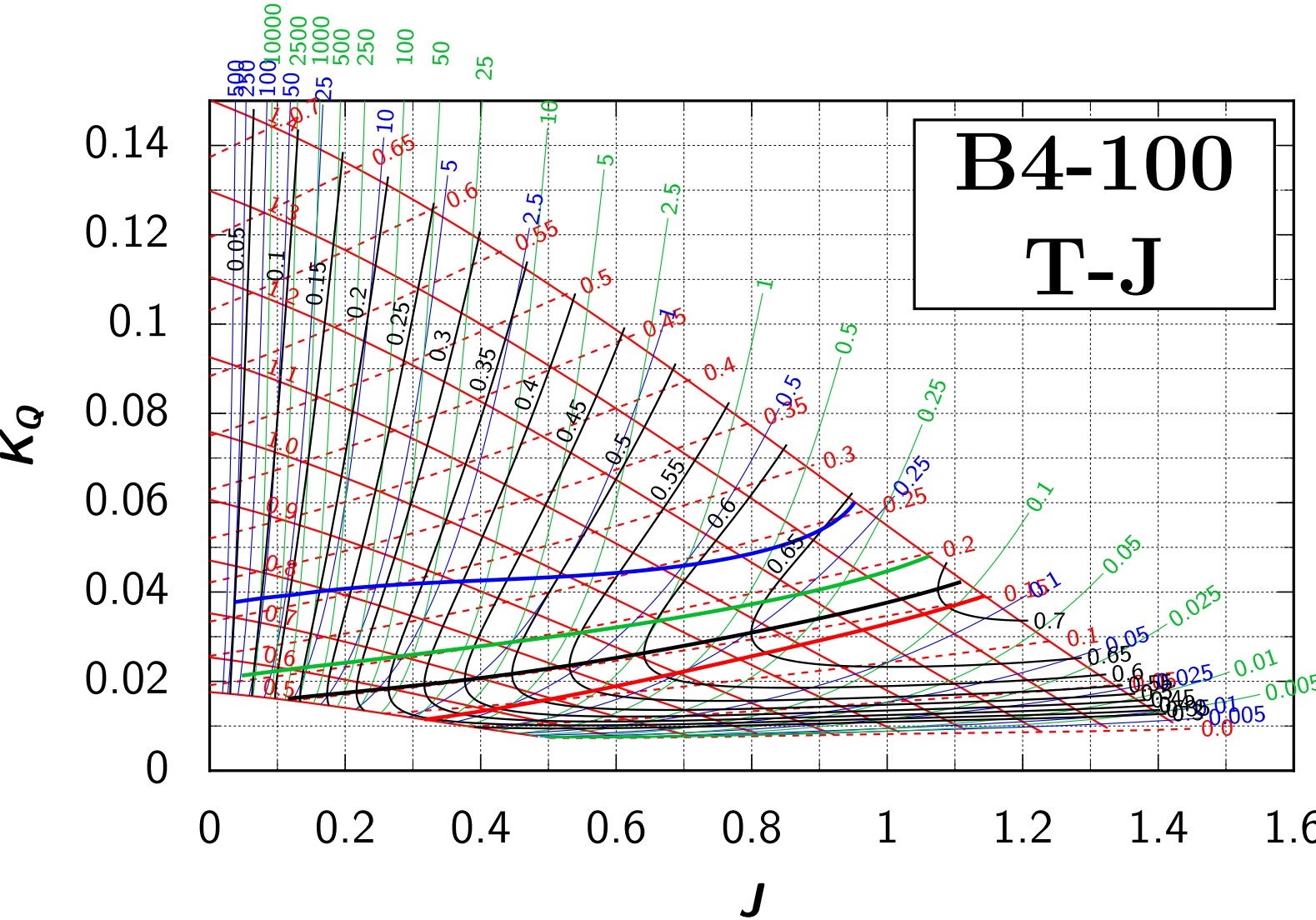

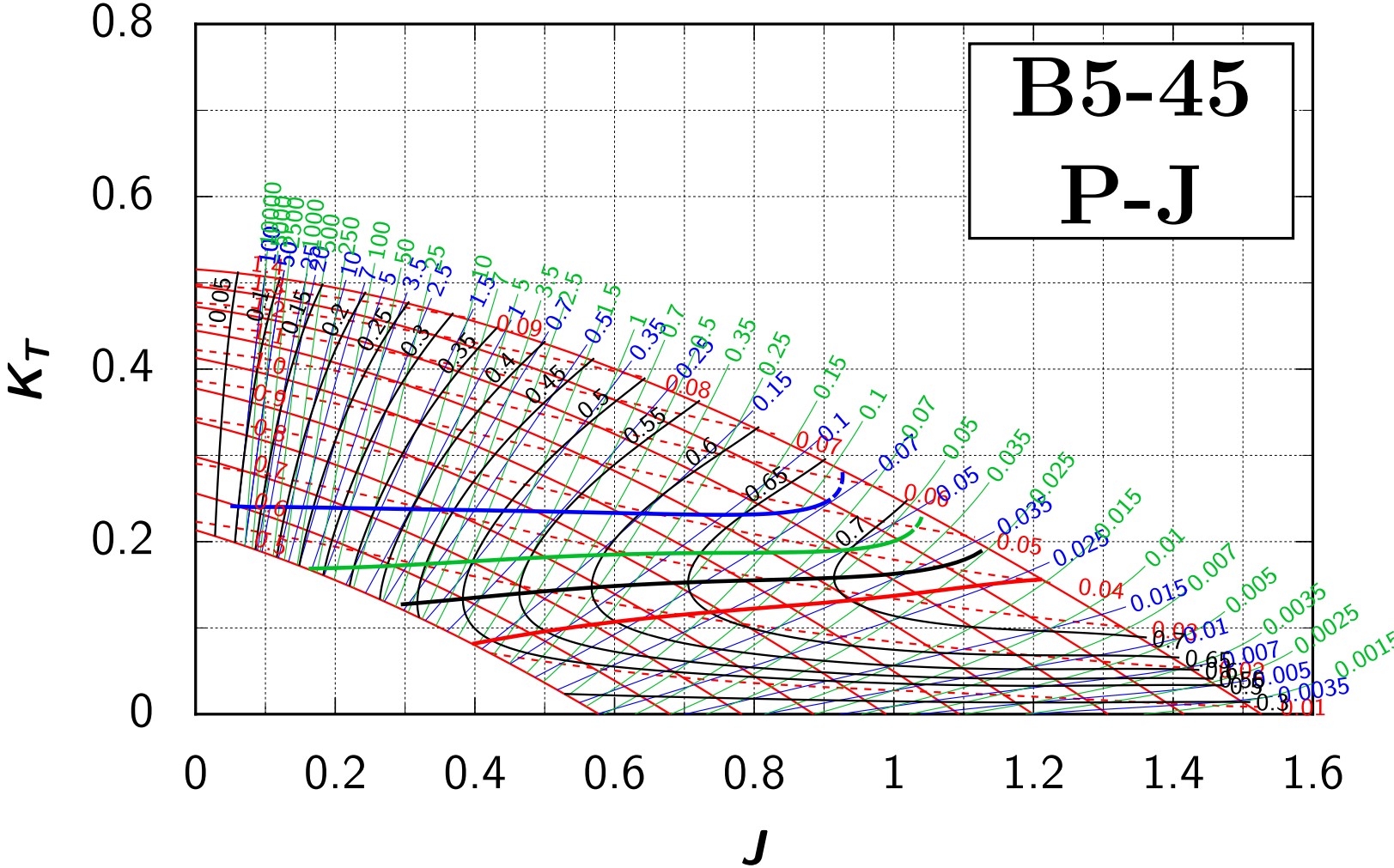

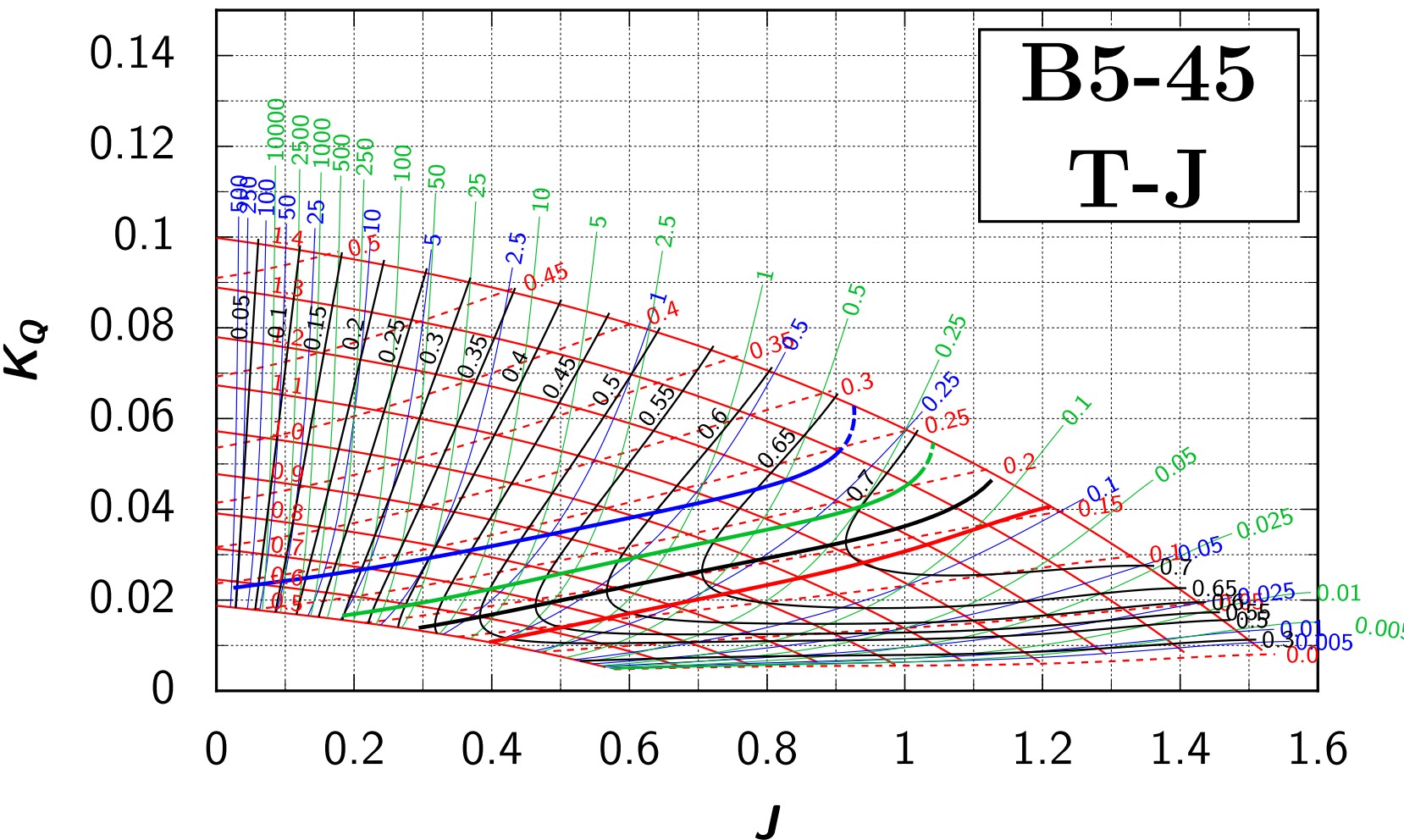

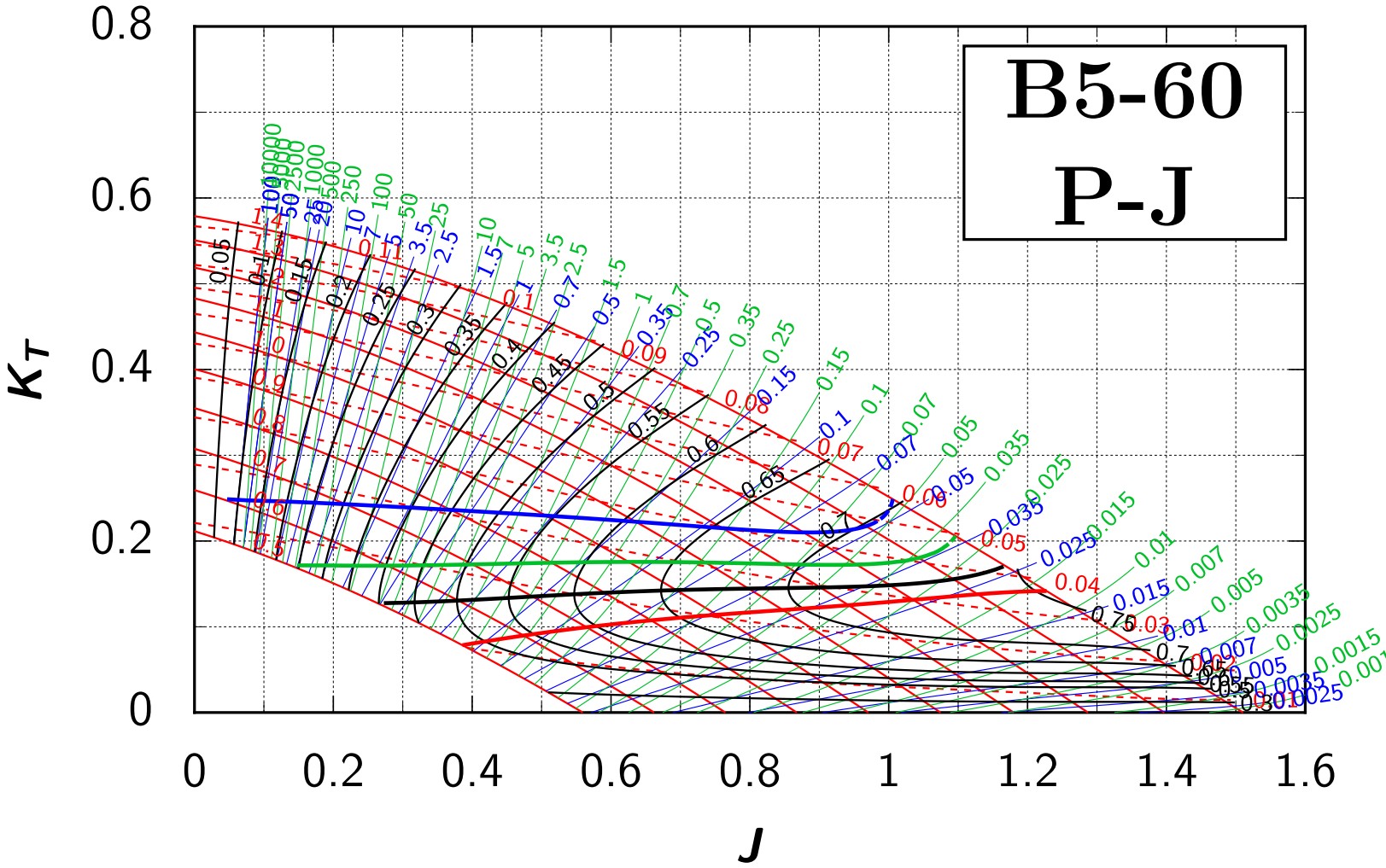

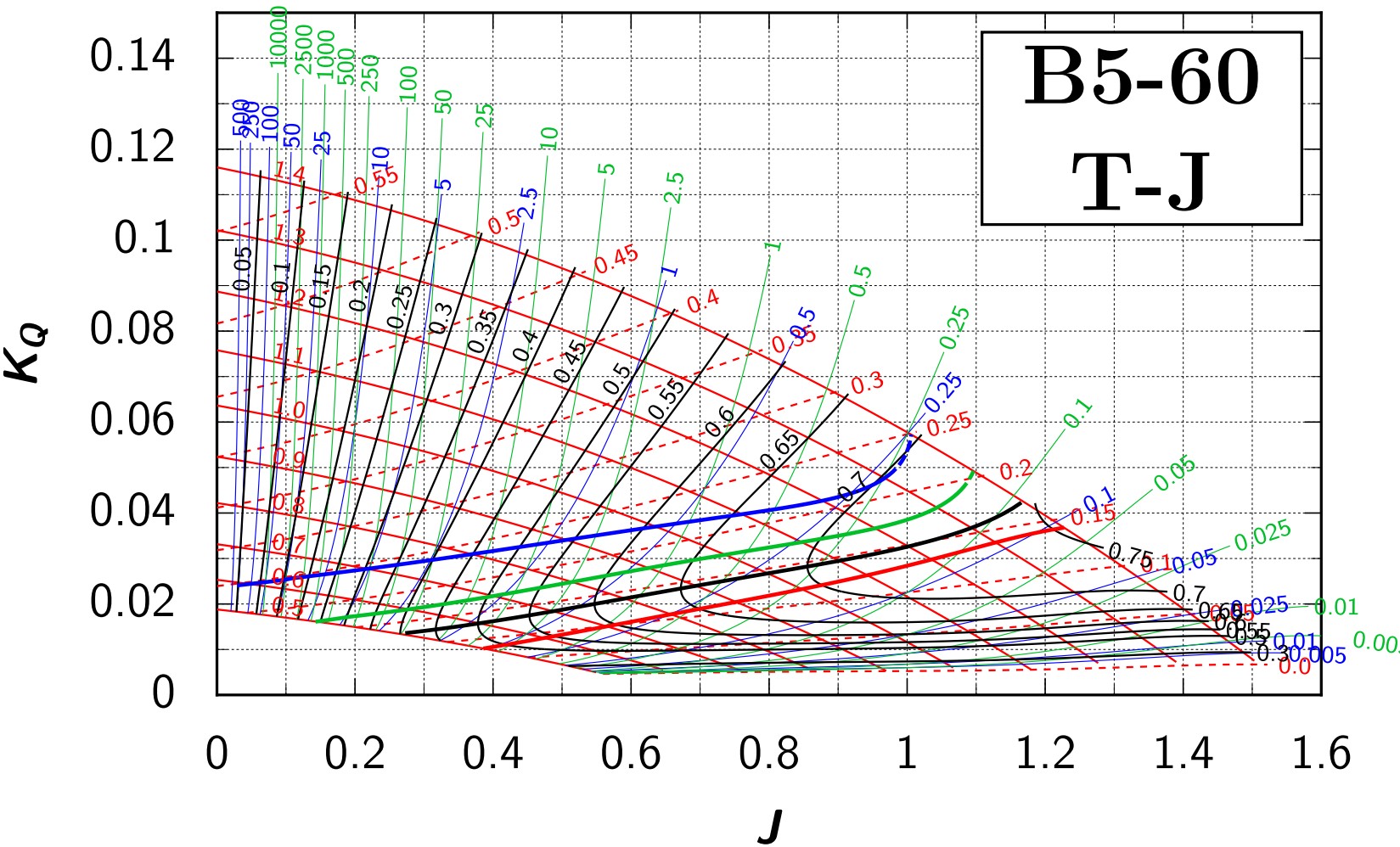

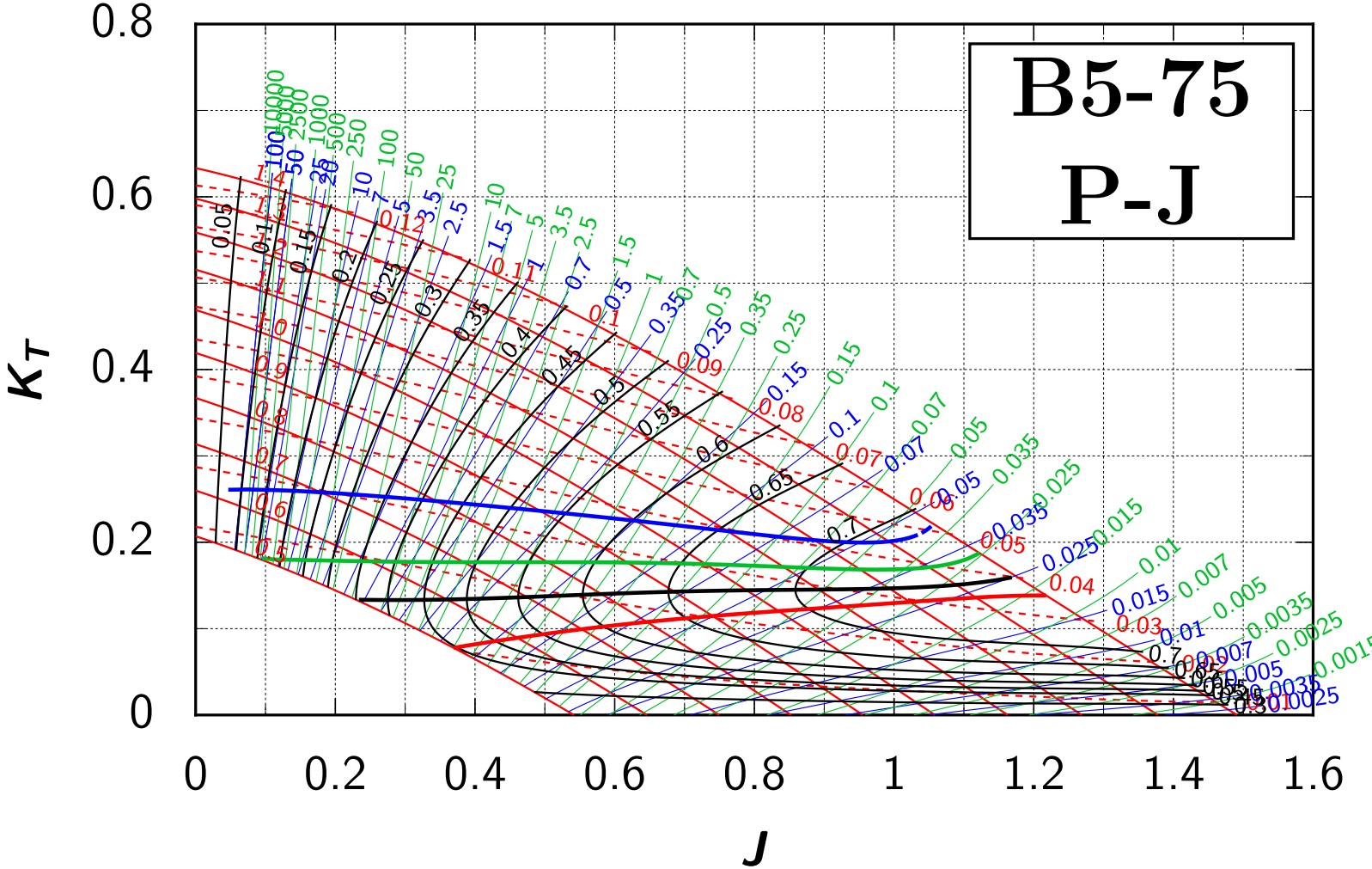

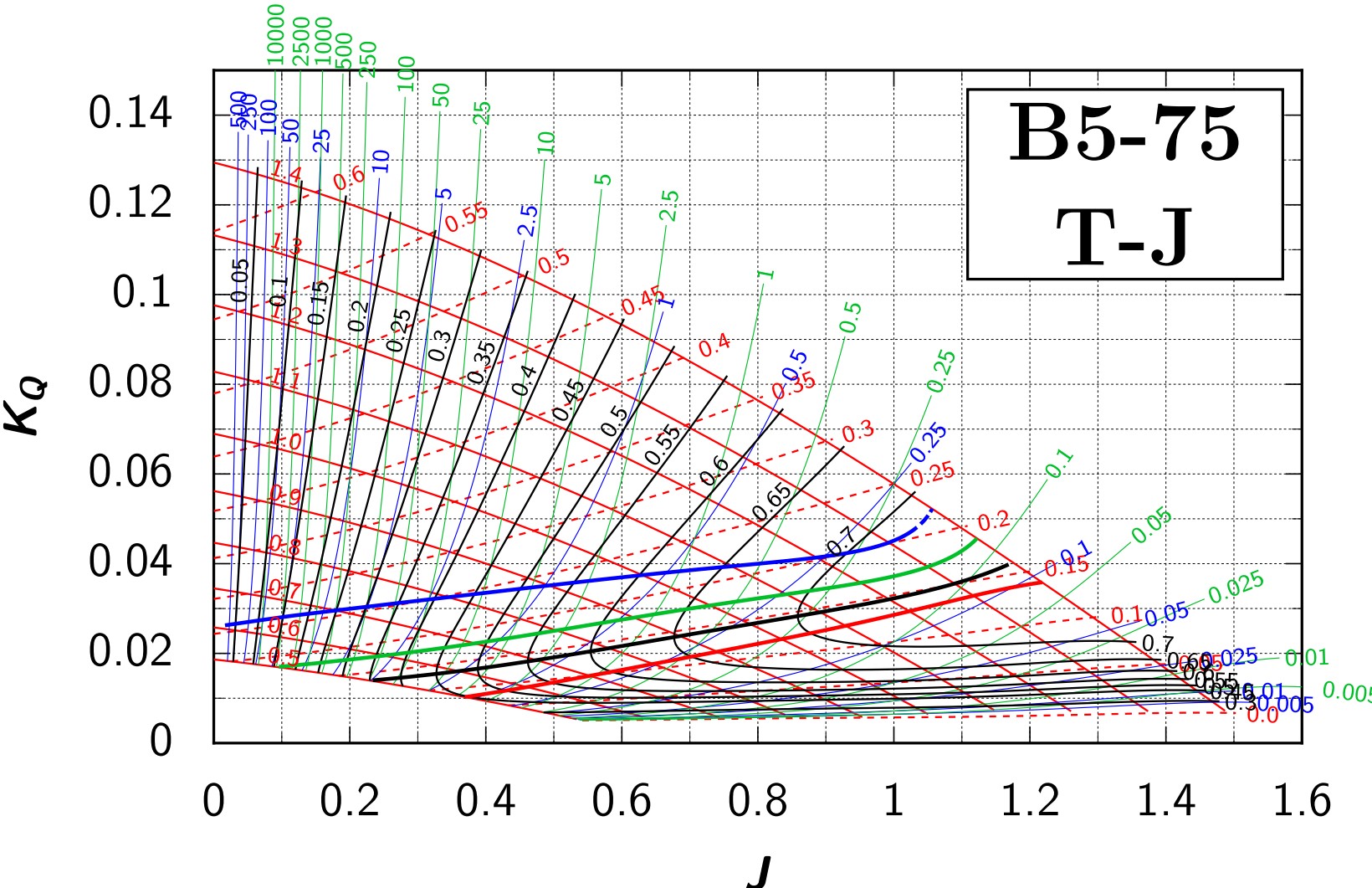

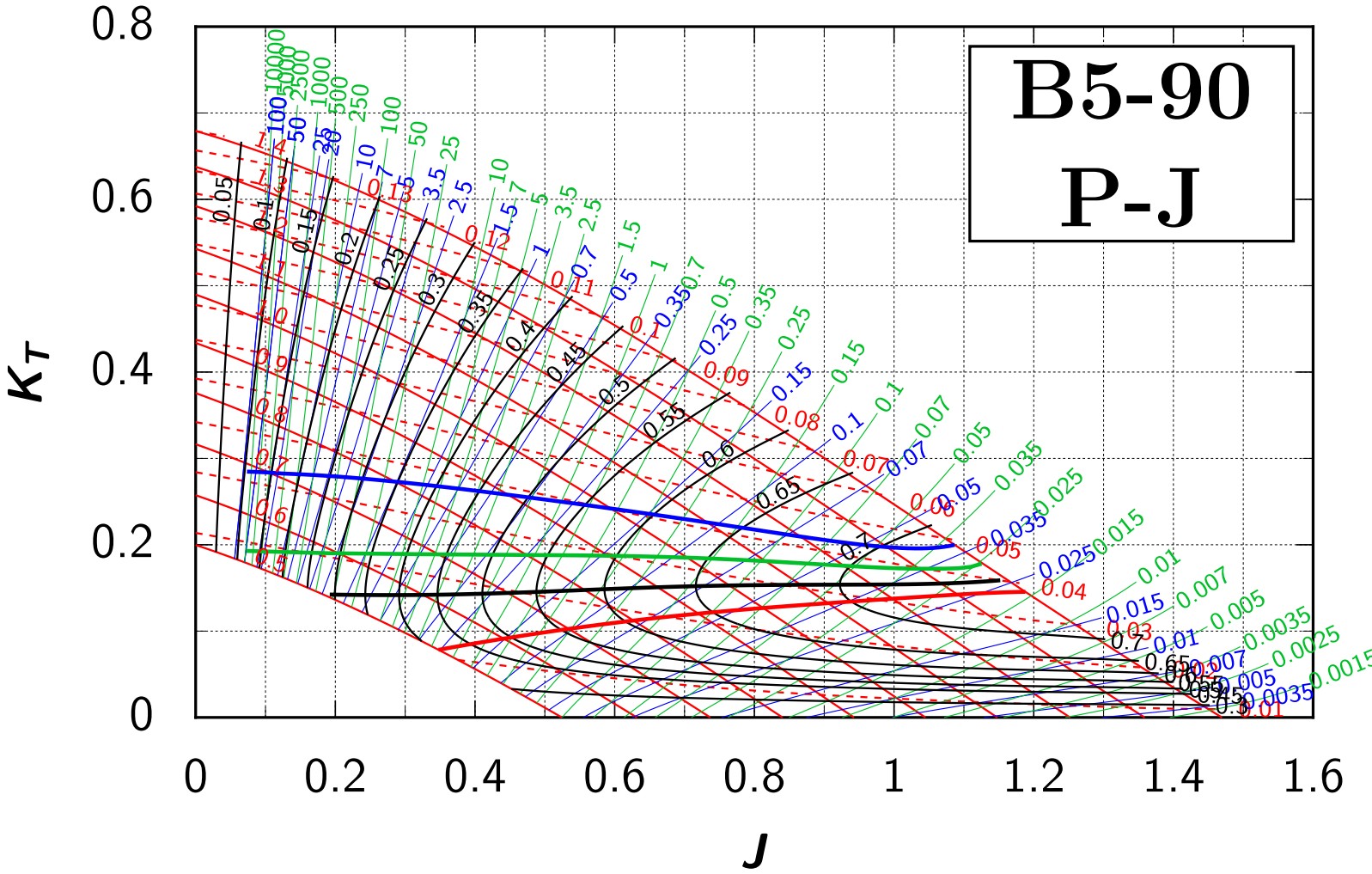

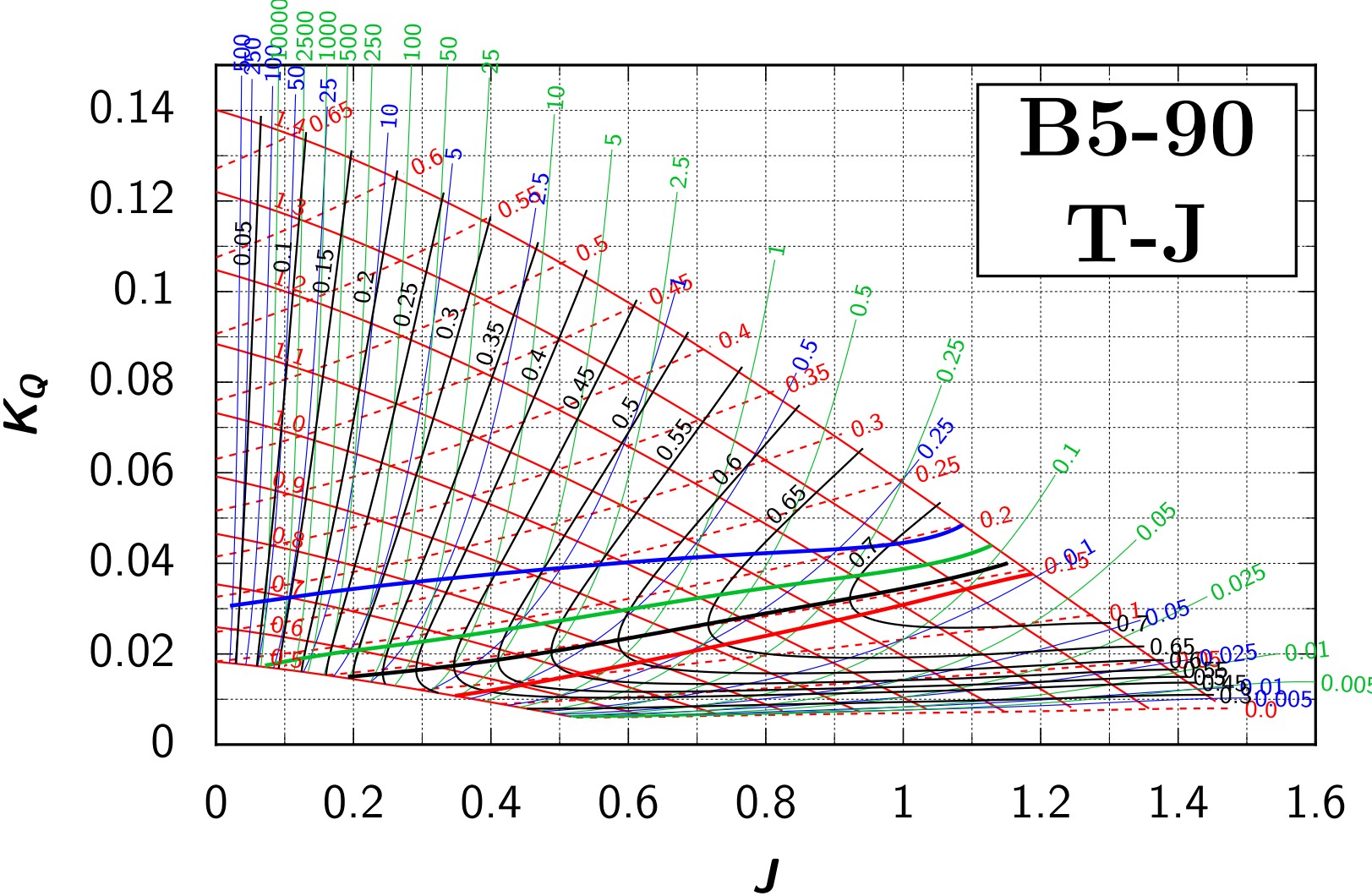

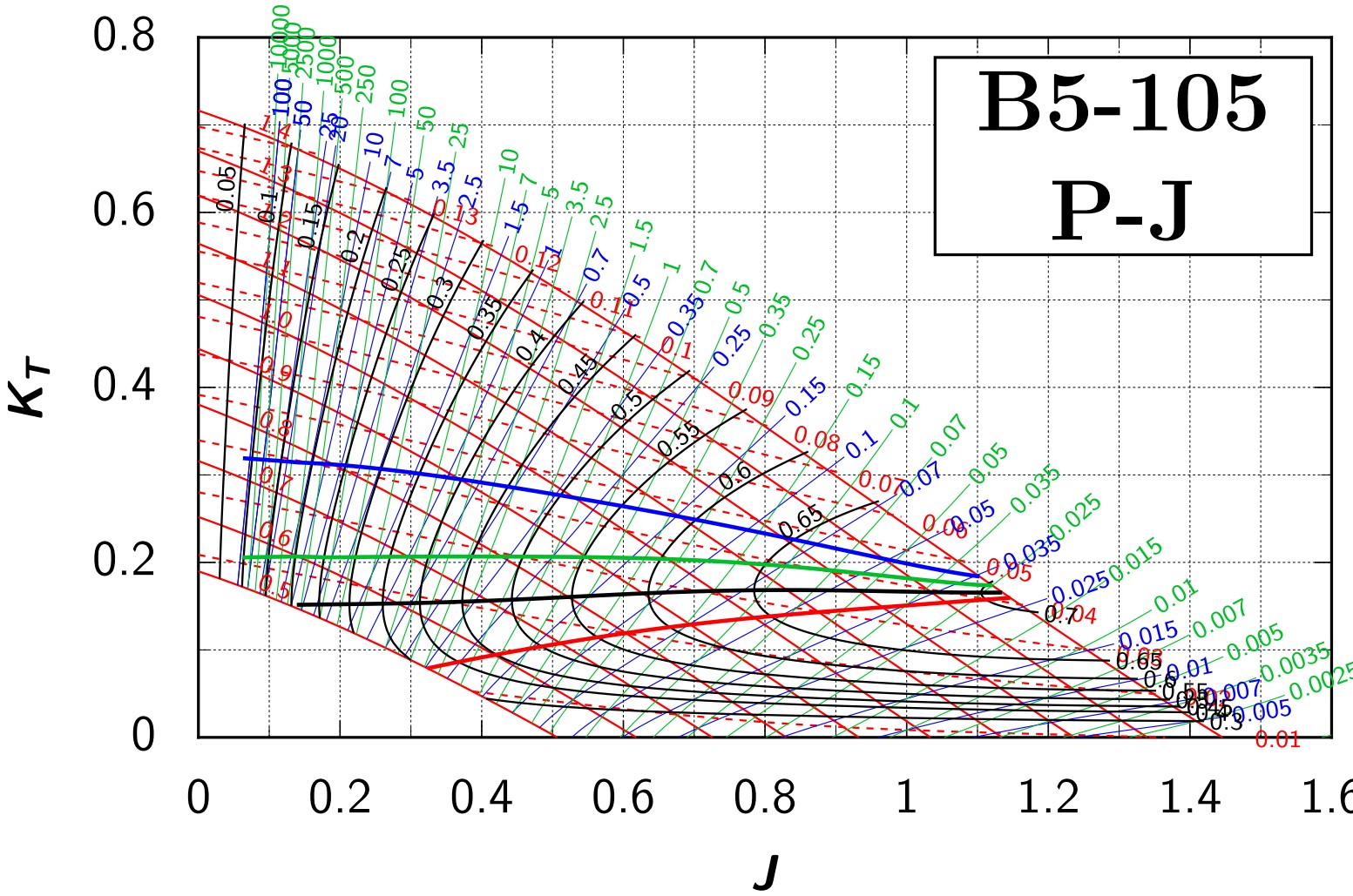

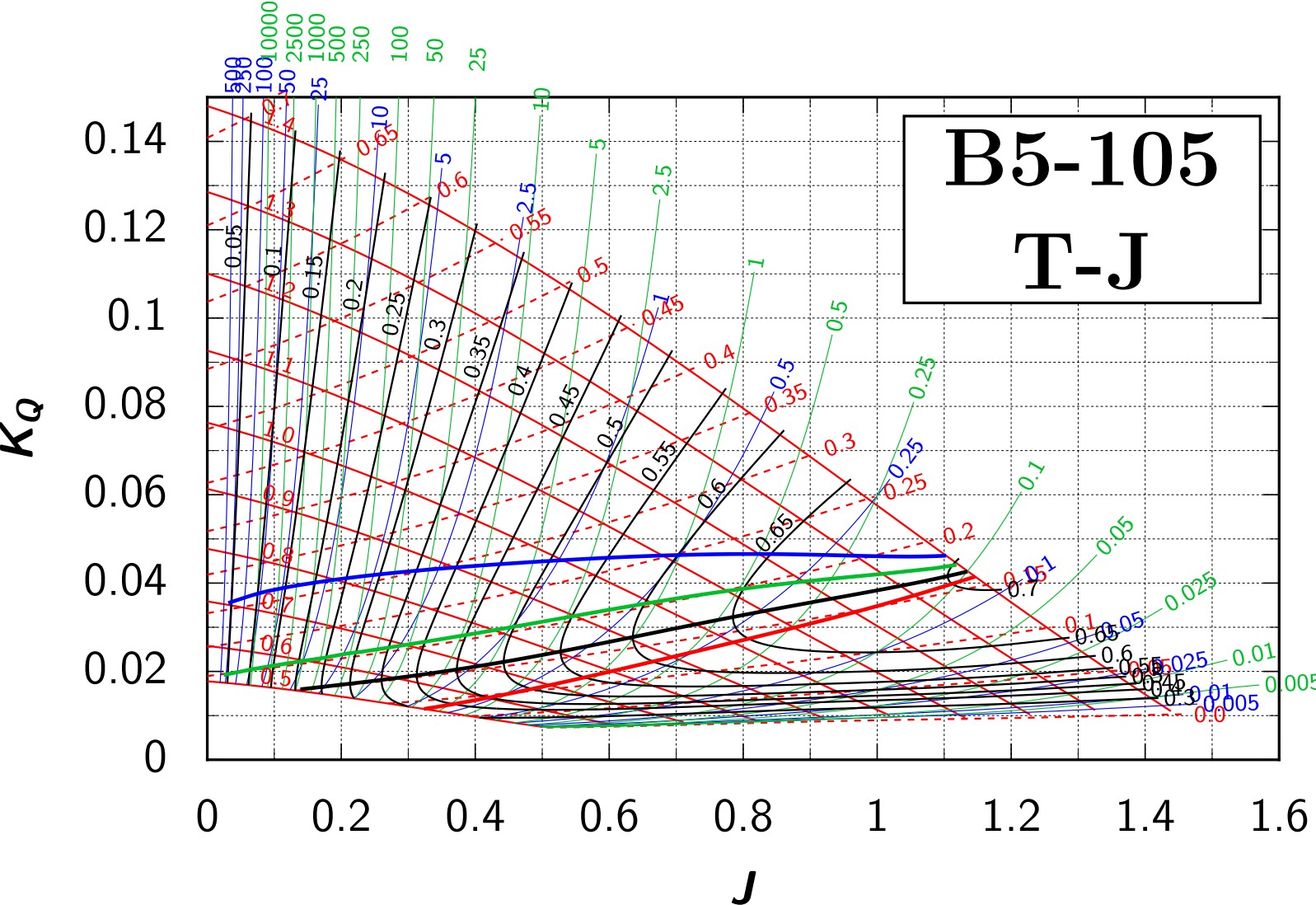

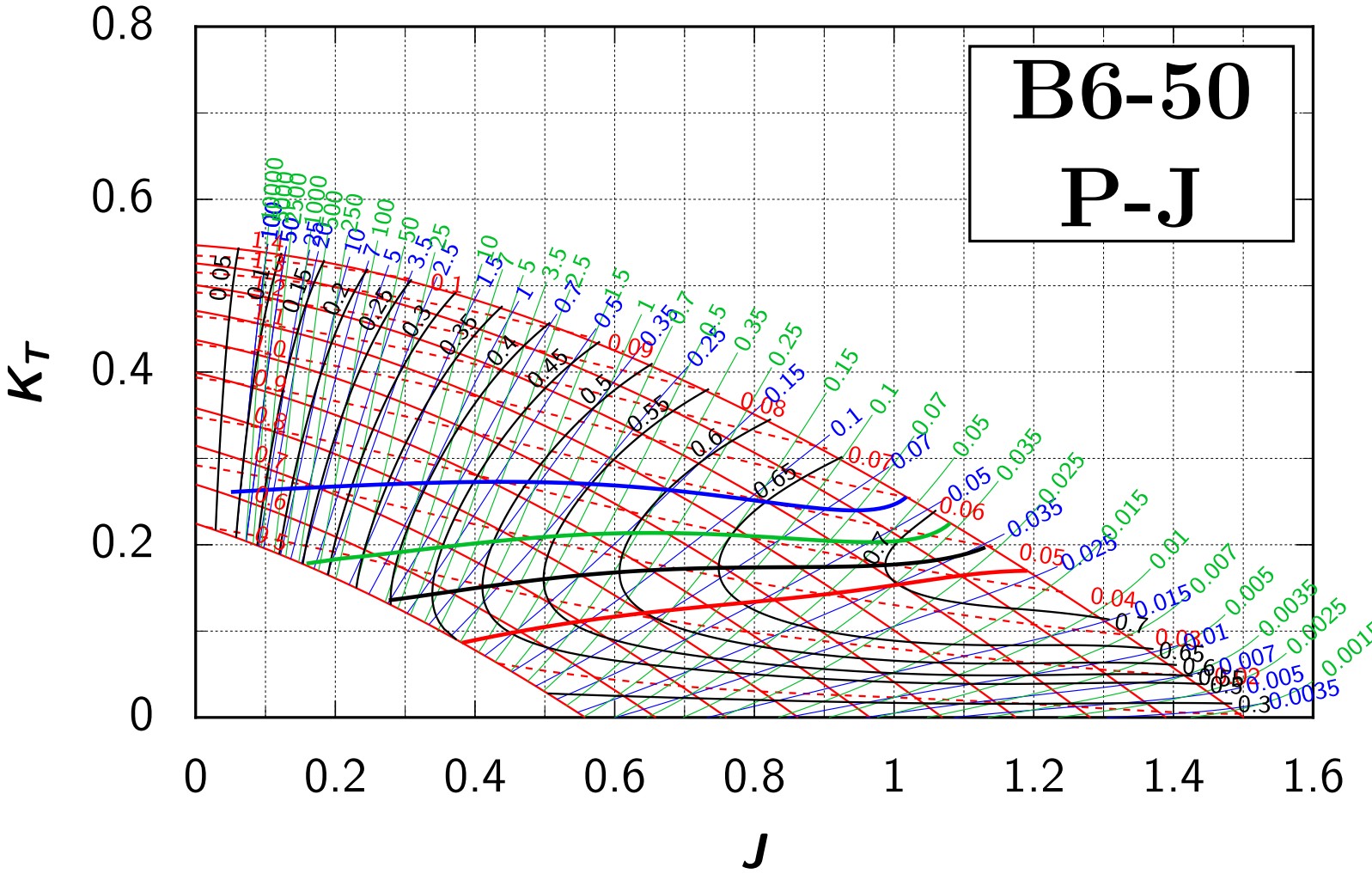

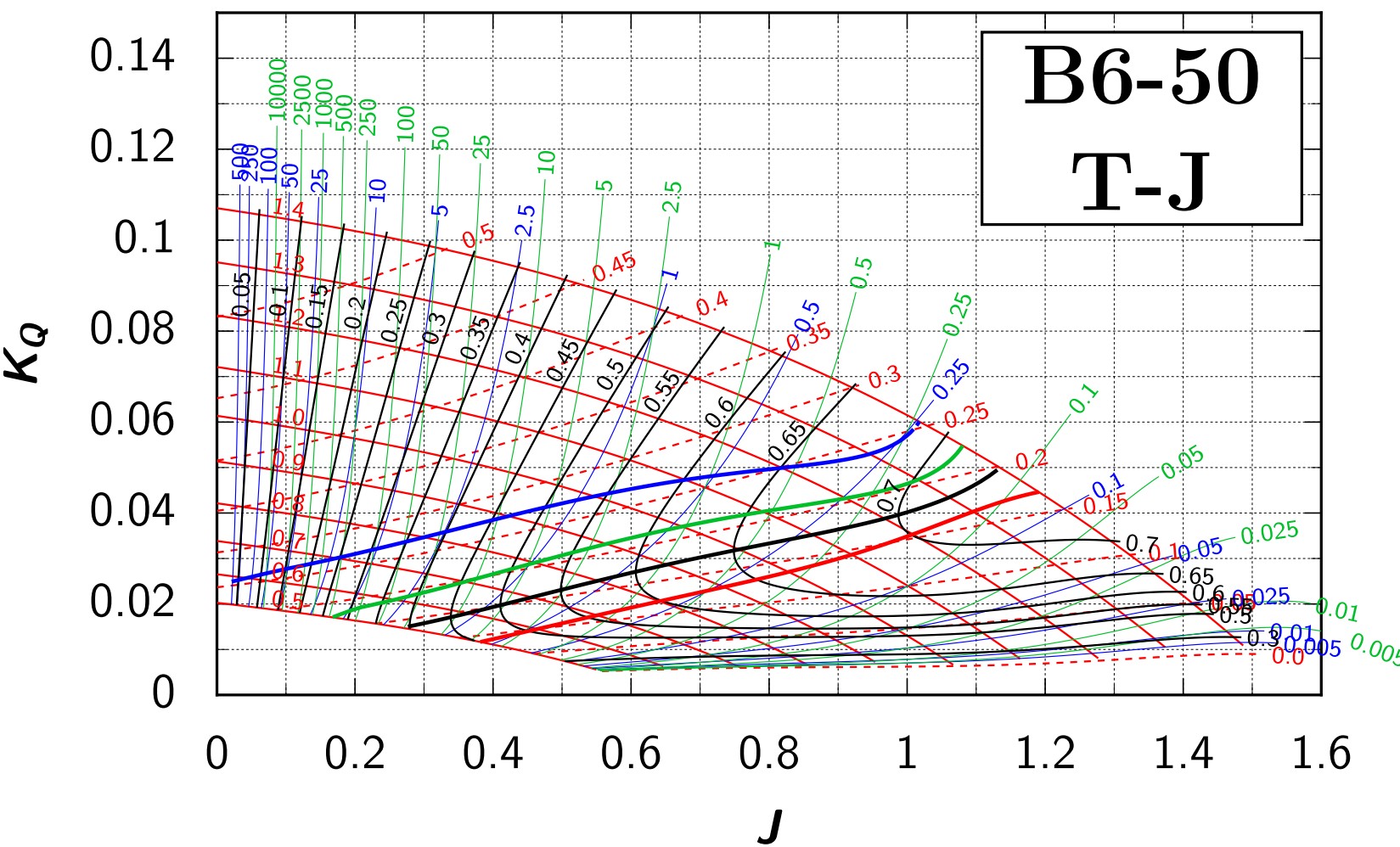

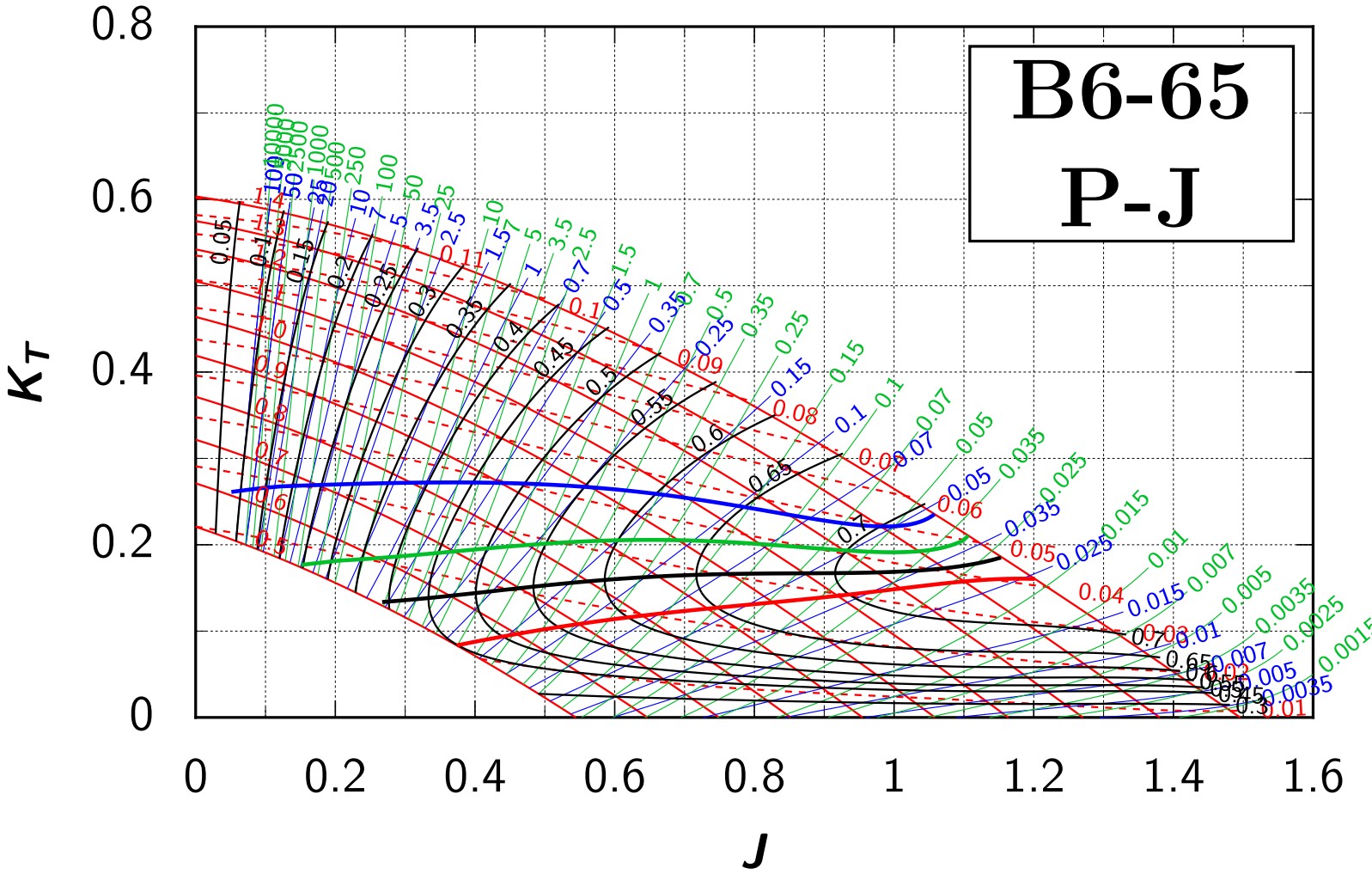

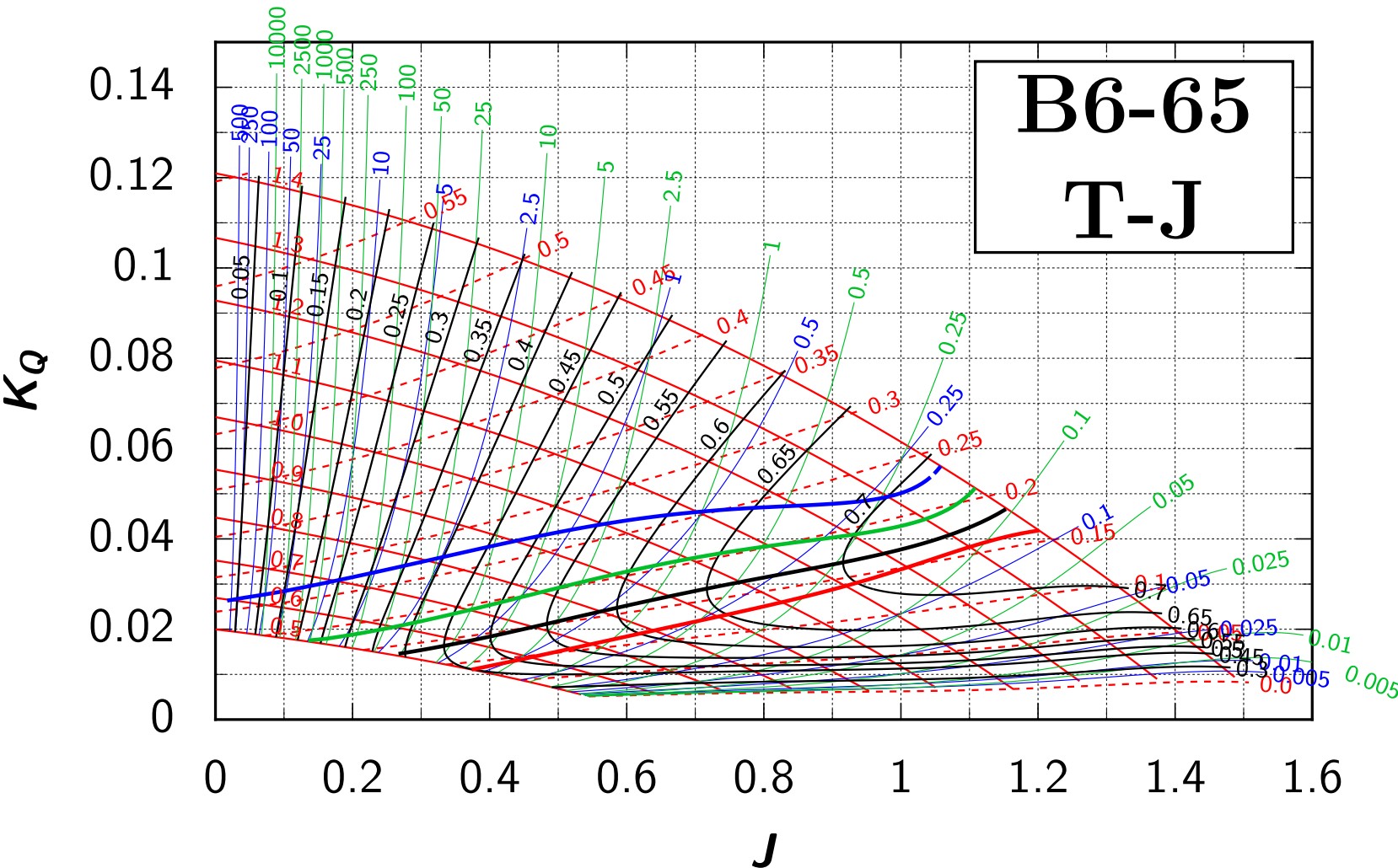

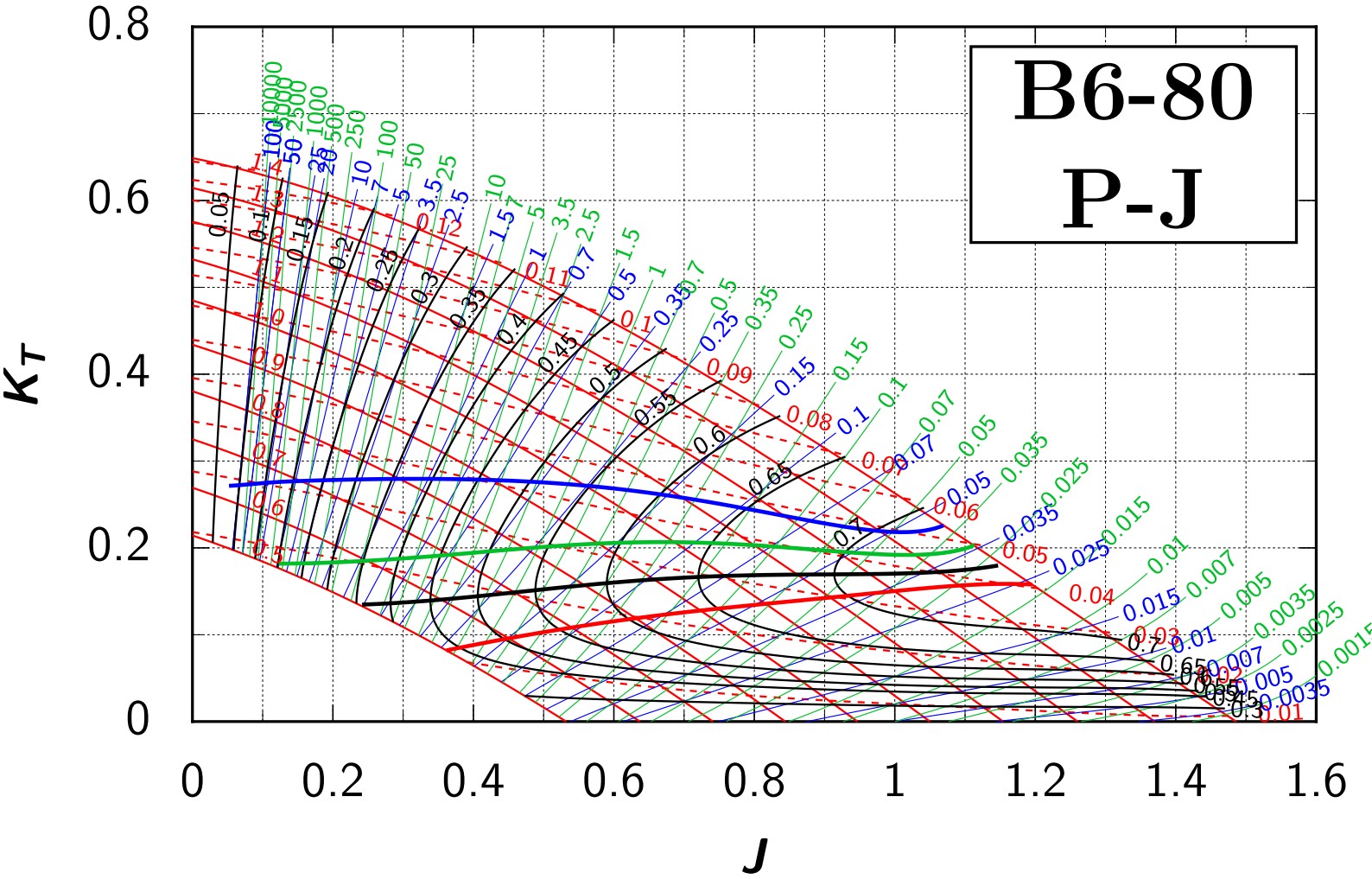

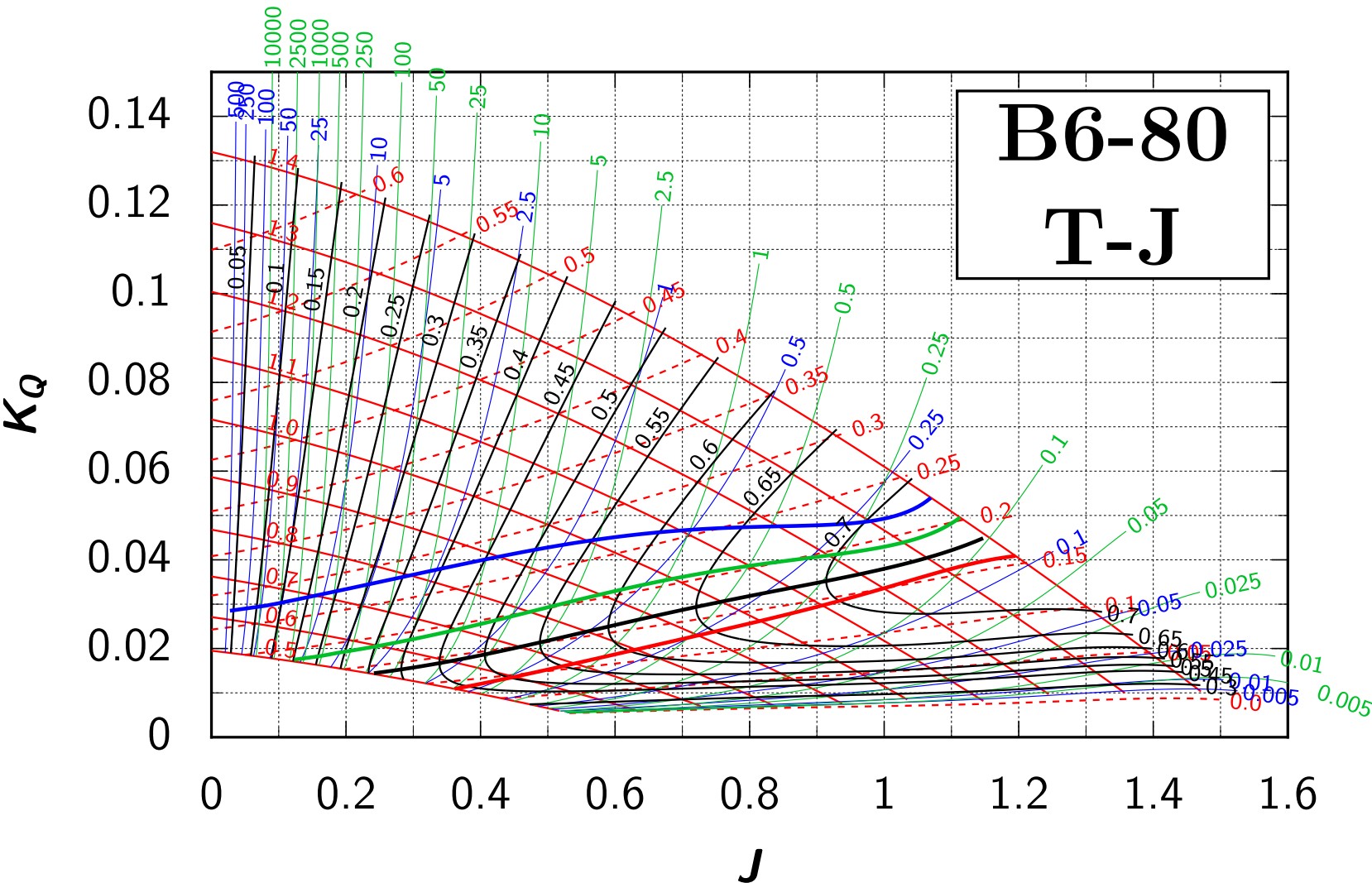

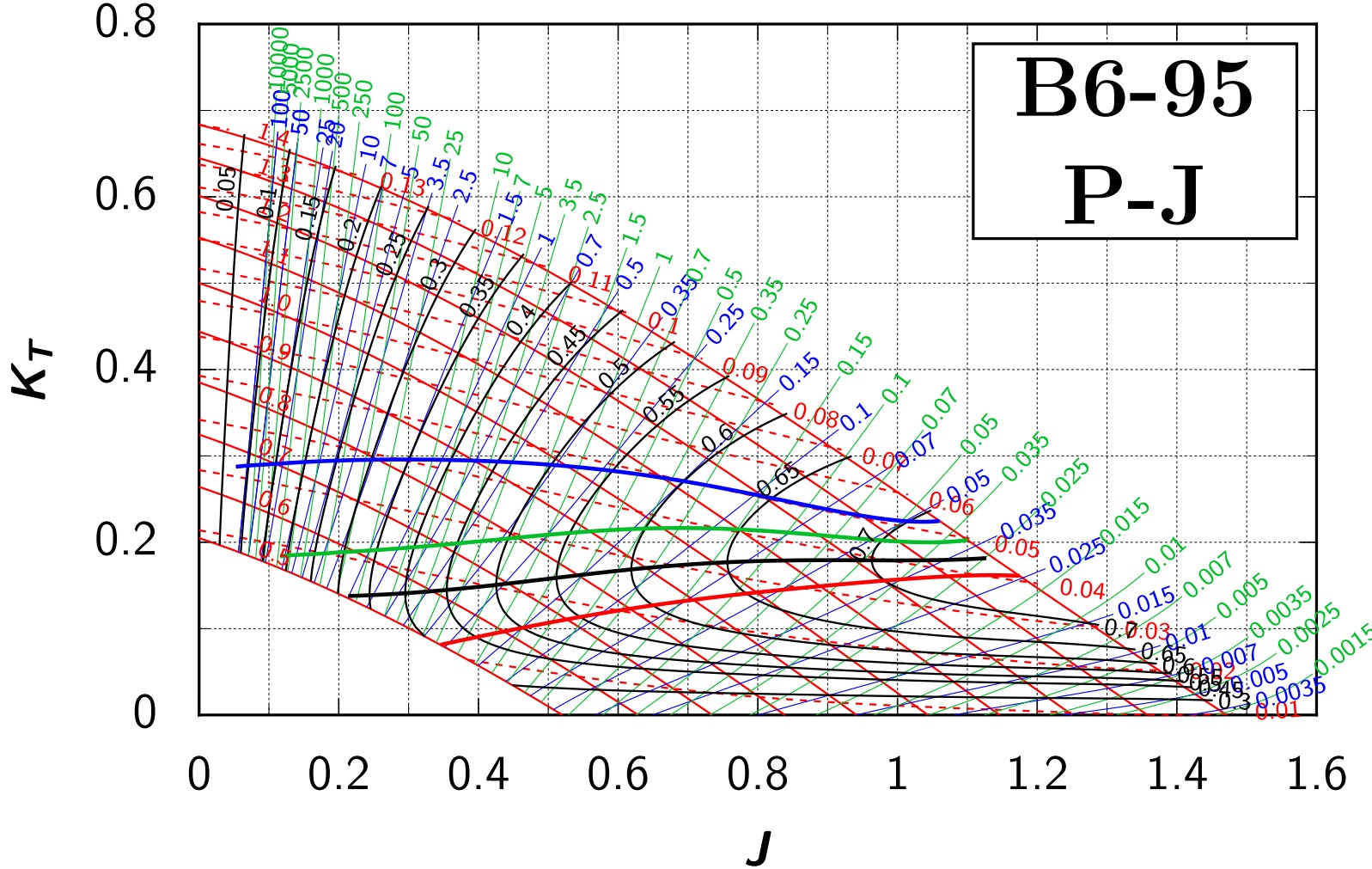

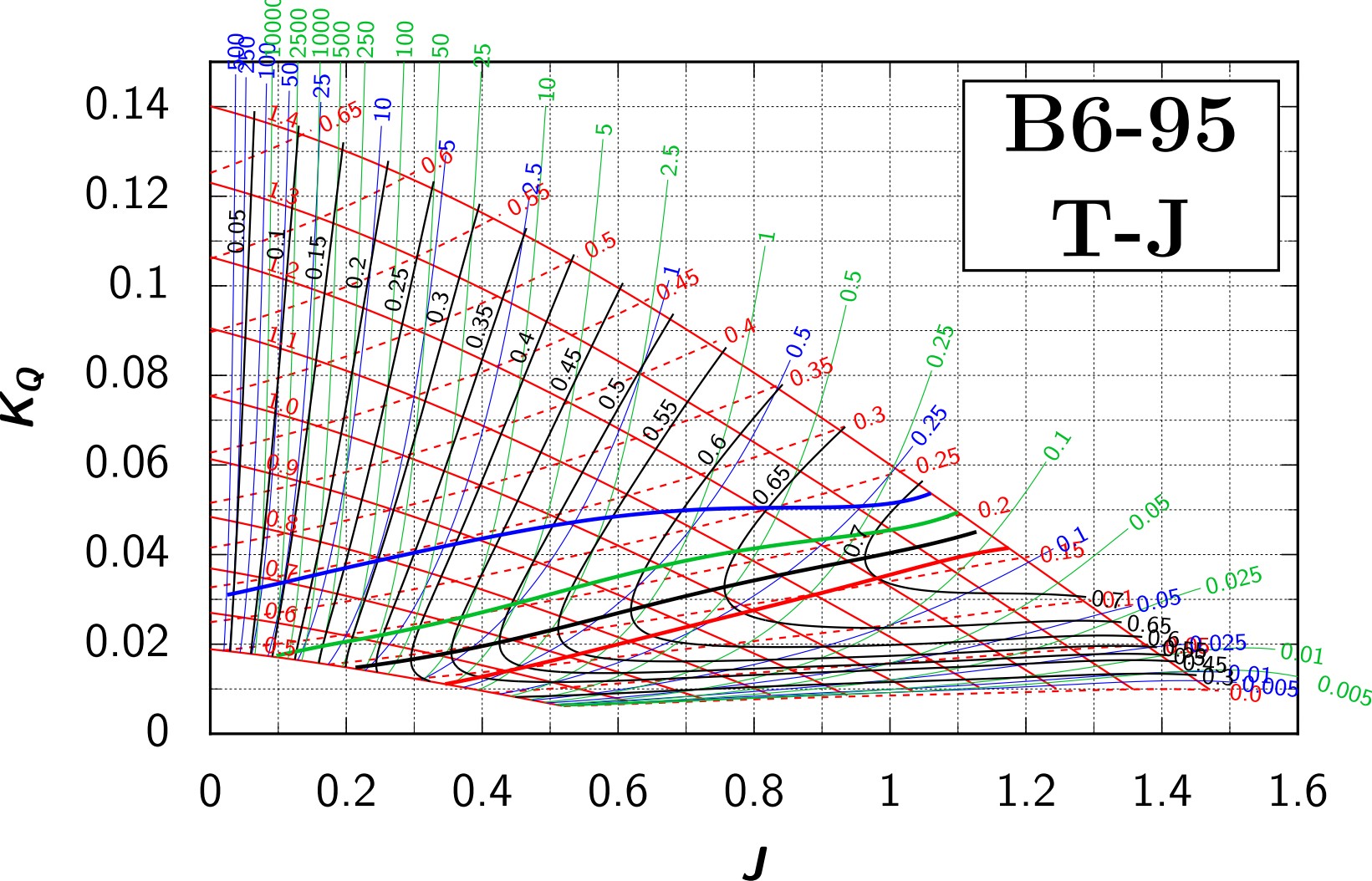

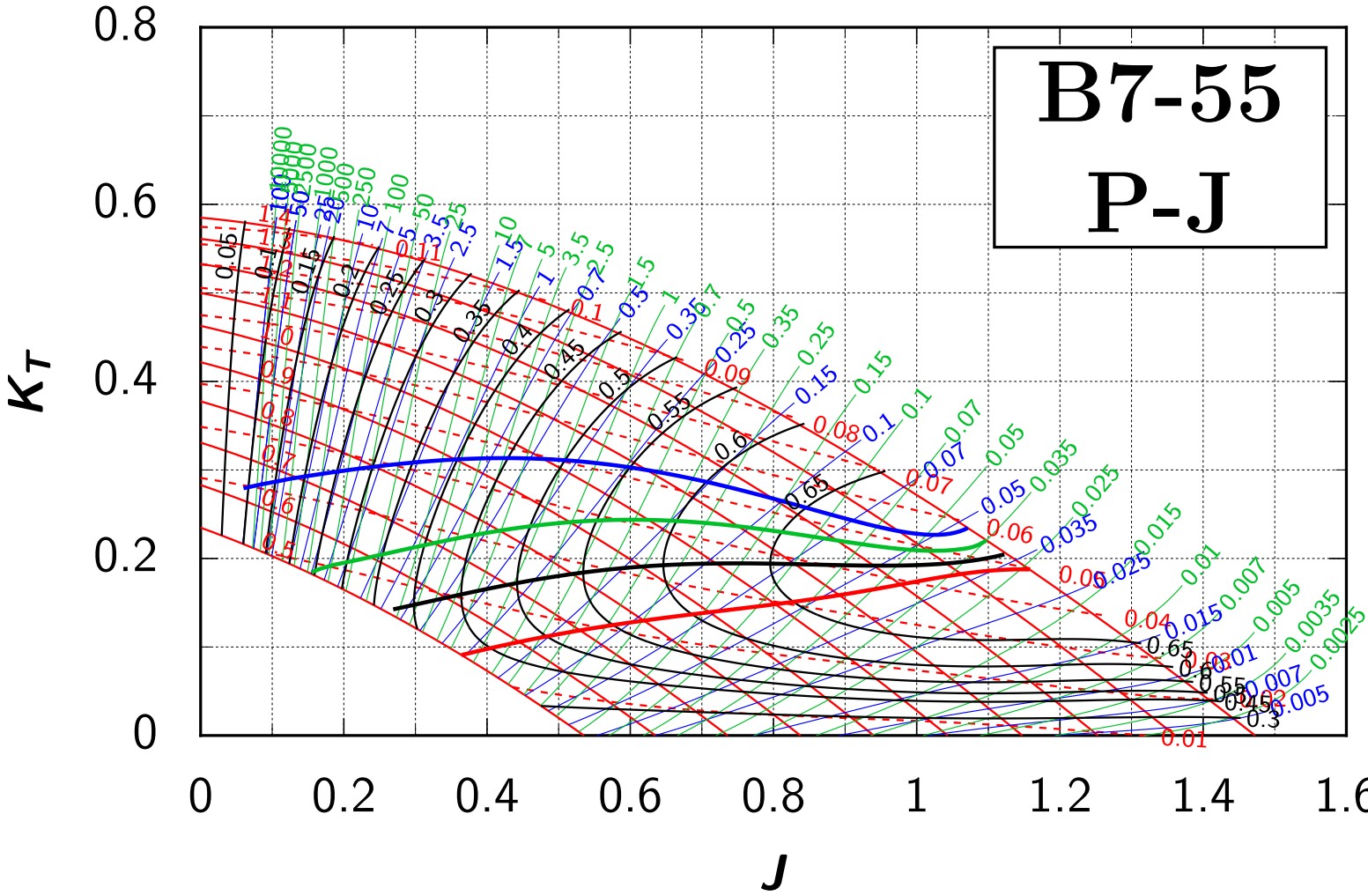

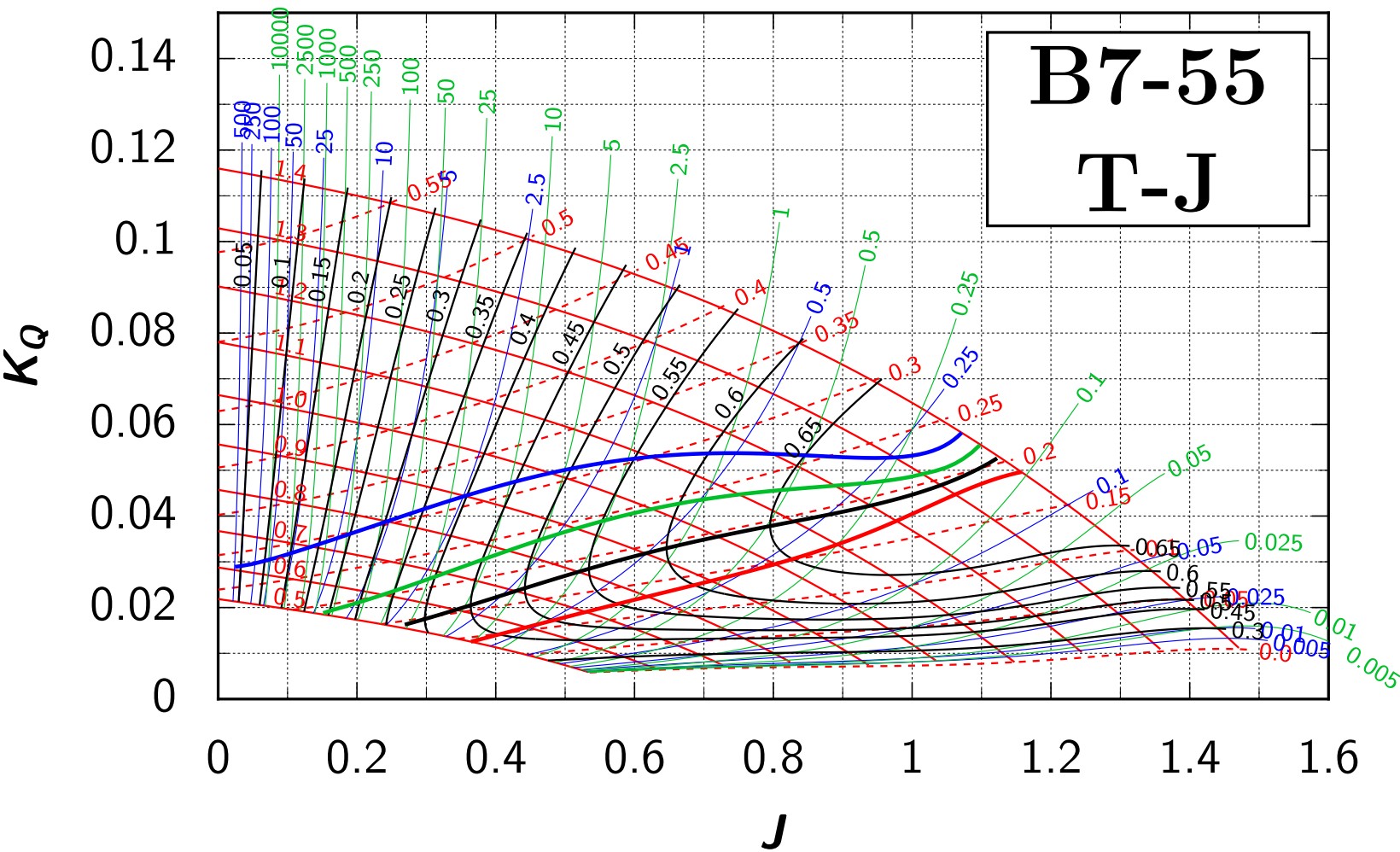

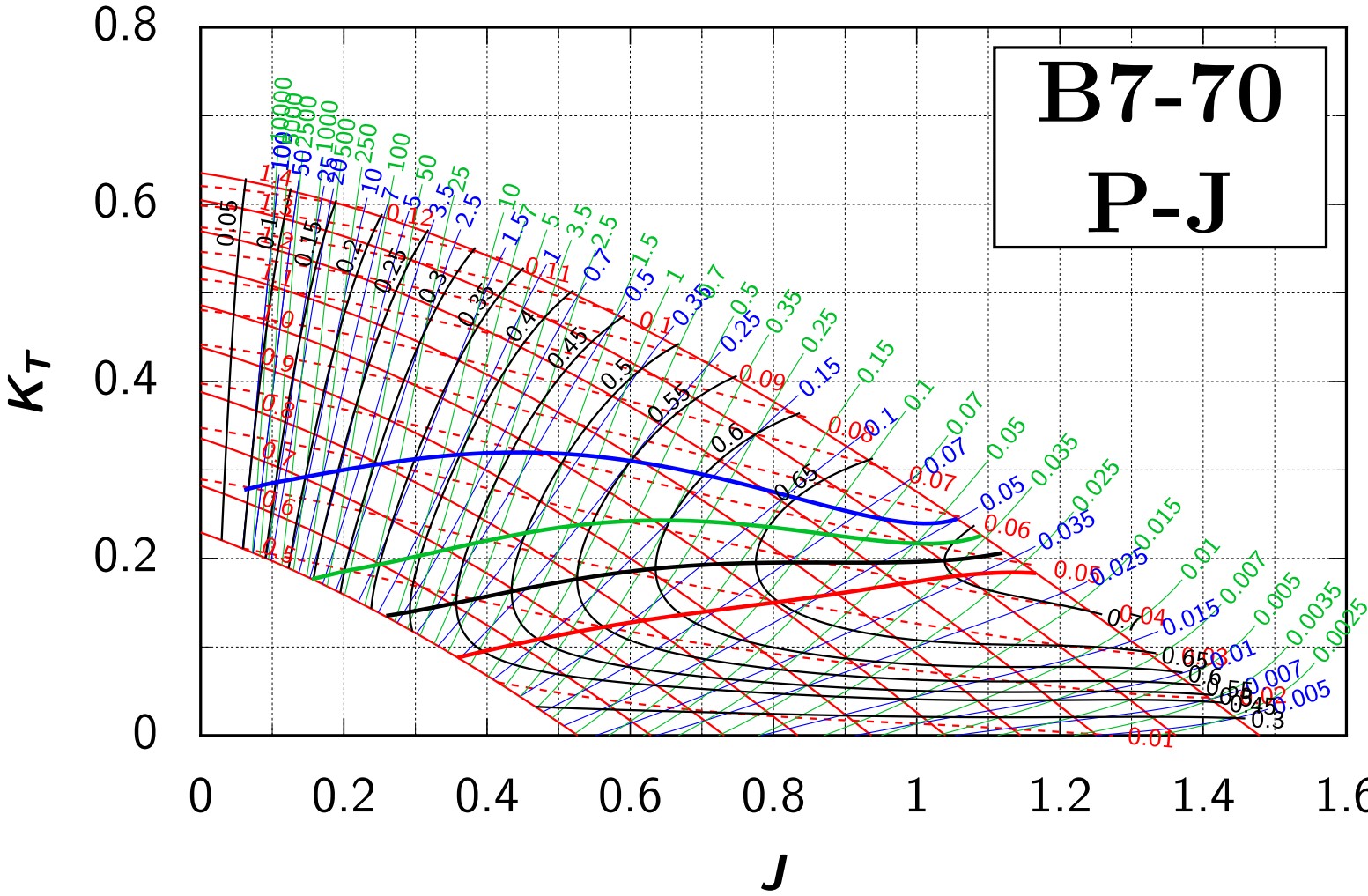

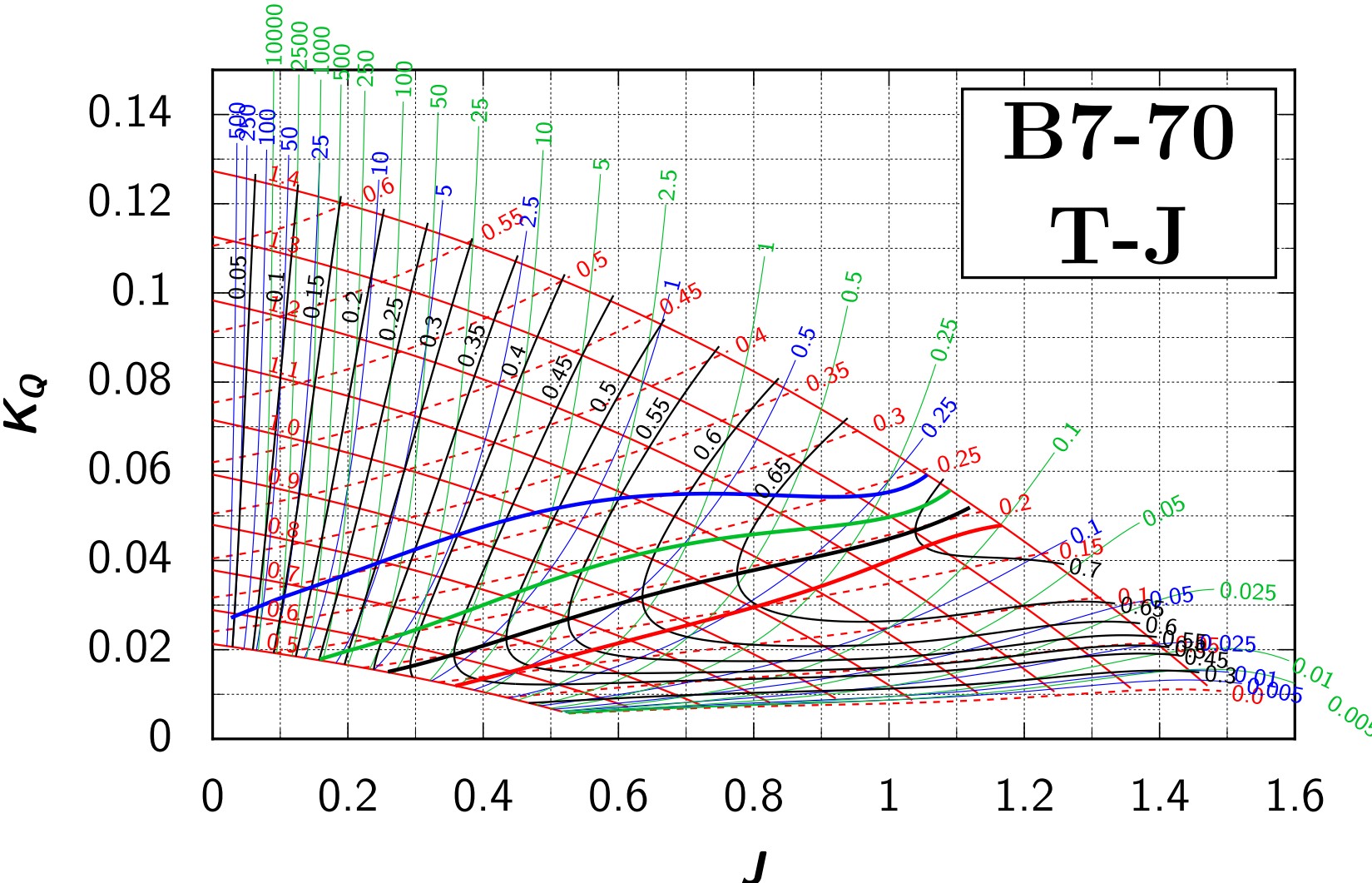

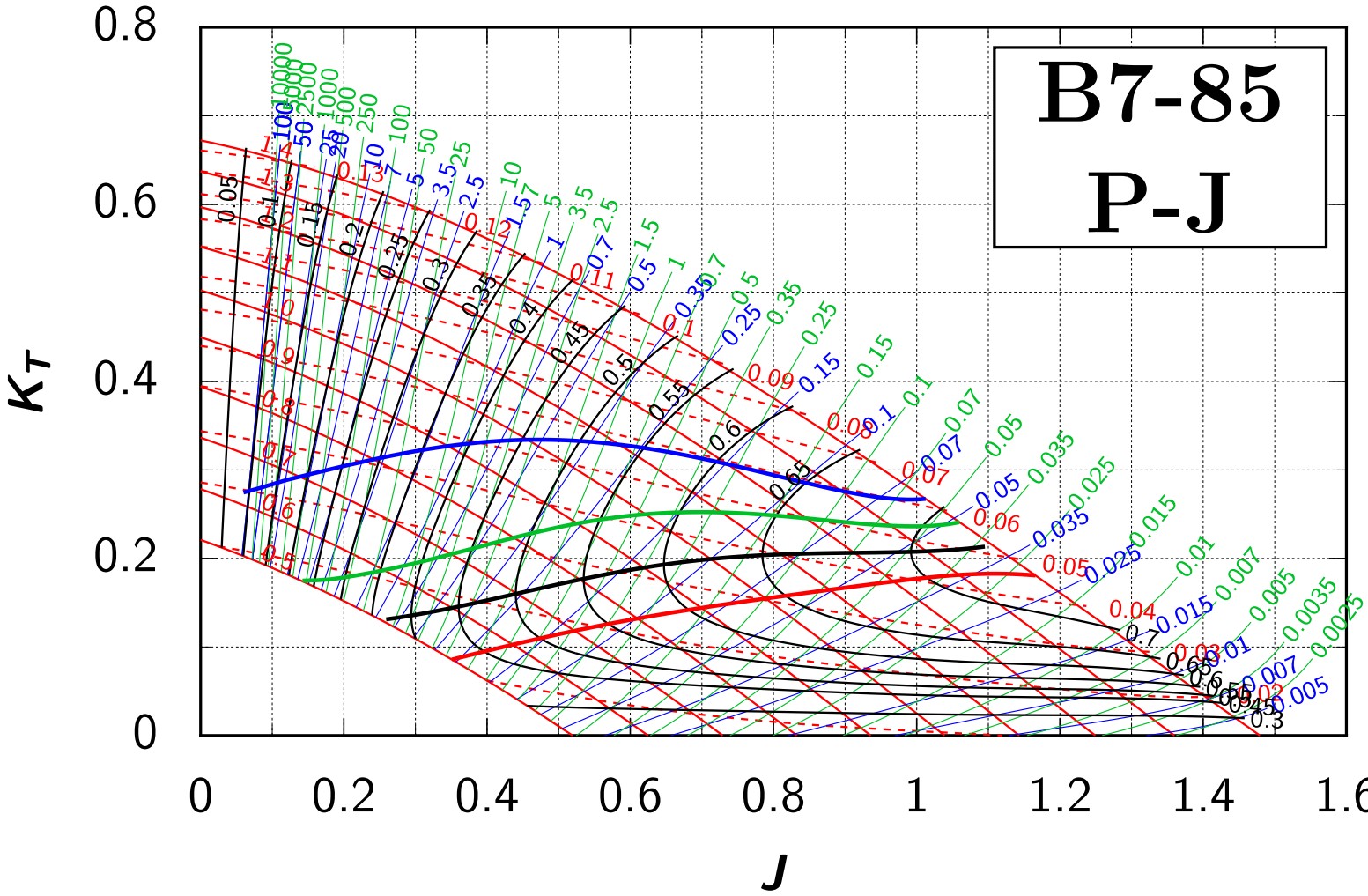

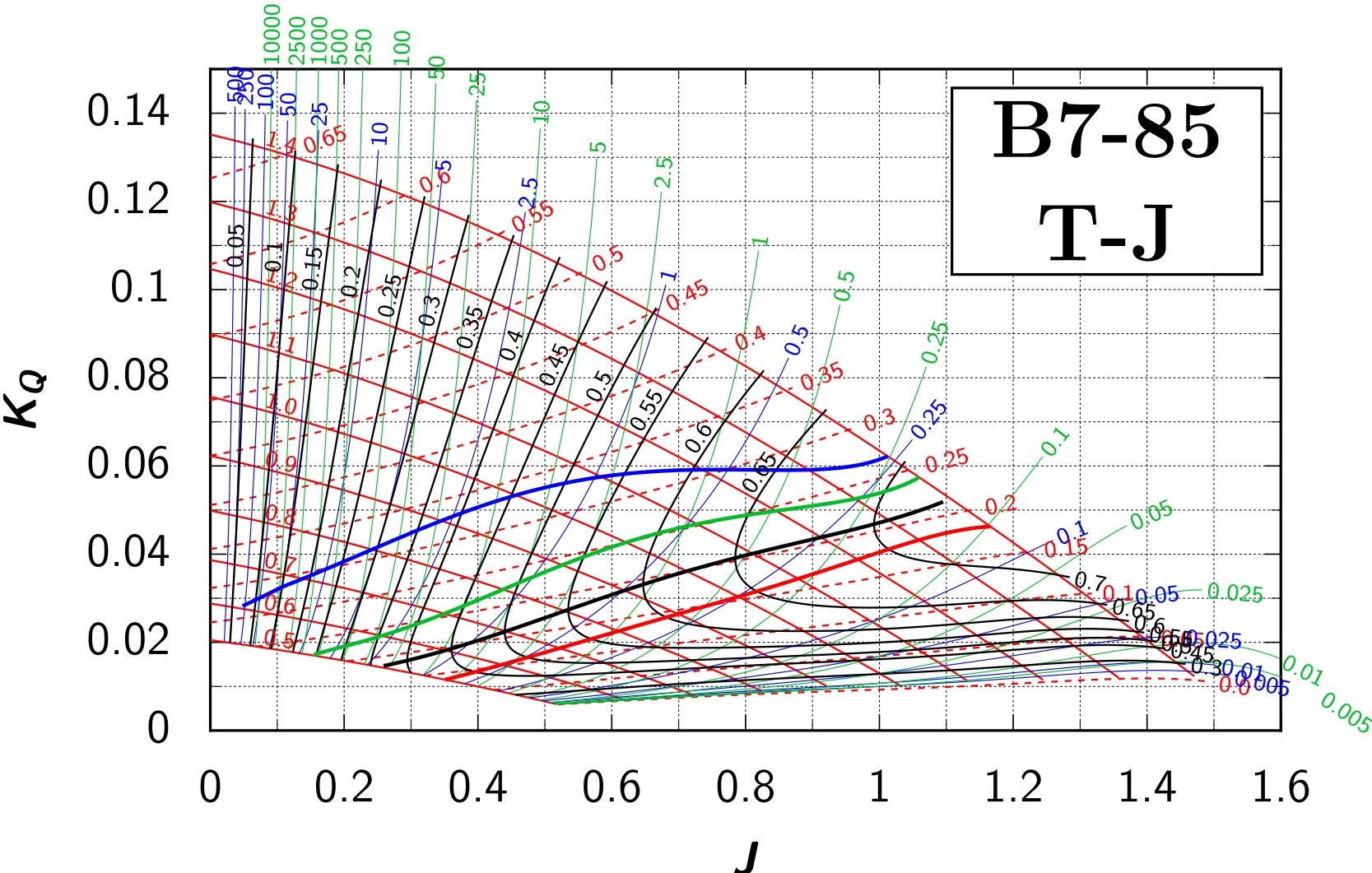

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
