# Peer review of "Surprising Behaviour of the Wageningen B-Screw Series Polynomials"

_jmse, doi:10.3390/jmse8030211_

Round 1

Reviewer 1 Report

The following improvements must be made:

Regarding the topic, it is necessary to cite and comment the book: Kuiper, G. The Wageningen propeller series, MARIN Publication 92-001, May 1992. Significance of graphical representation and specific related by-hand design methods are nowadays highly reduced. Discussion of appropriate automated design procedures (in particular, based on global optimization methods) should be added. Discussion of 4-quadrant models should be added. The available B-series regression model is uniform over all factors. It is possible to use it for screw propellers with non-integer number of blades which hardly makes any sense. Instead of a single regression model it would have been possible to develop 6 much simpler regressions for each number of blades separately and I am sure that these regressions would have been more accurate. I am asking the author to discuss this issue.

Author Response

Thank you for your kind review.

You mention that it is necessary to cite the book by Kuiper. This is a very important book indeed and it presents all the available data in a very neat and accessible way. For the open-water diagrams Kuiper uses the polynomials calculated by Oosterveld et al. Regarding the current paper, in my view he does not add anything new to the previous papers (which are all cited). May I ask, in which regard this book is relevant to this paper? I am more than happy to cite it.

The aim of the paper was not to present a new pen and paper method to design a propeller but to show the anomalies in the available polynomials. (That you can use these diagrams to quickly select an optimized propeller is indeed a nice side effect!) These anomalies are only clearly visible on the efficiency maps. It is not possible to spot them in the open-water diagrams. Unfortunately I had to expand on and explain these efficiency maps (and also their use), because they are not widely known. This might have moved emphasis the emphasis away from the anomalies of the polynomials. I might move this lengthy explanation into the appendix to put the spotlight onto the anomalies.

Regarding optimization and automated design: For more sophisticated propeller designs, the Wageningen B-series just delivers a good starting point to speed up the lengthy process of the more sophisticated optimization methods. For the smaller end of the market, the B-series is widely used in the industry to give reasonable dimensions (which then will be corrected by factors coming from experience and which are well-guarded secrets of each propeller company). Within this simplified approach – if the blade area ratio and the number of blades are fixed and influences of the propeller onto the ship speed, the wake field and the thrust deduction factor are neglected – we live in a linear world: There is just one optimum propeller for a certain combination of power (or thrust) and diameter or revs: The optimum propeller always lies on the line of ηmax. There is no need to employ sophisticated global optimization routines. But it is possible to use any optimization routine to find the point on the ηmax lines, because these lines are actually the results of solving optimization problems with different constraints. In approach outlined in the paper, I make use of the fact, that in the optimization point, the PD line (or any of the other three lines) must have the same tangent as the ηo=const line. If you are thus inclined, you can use any optimization routine you like with the right constraints. But keep in mind, that you might find a solution on the upper branch. (One of the fundamental conclusions of this paper!)

But as I pointed out before, this is not the purpose of this paper. The purpose is to show, that there are anomalies in the polynomials and that you might get unreliable results because of these.

You mention, that the 4-quadrant models should be added. Can you please expand on that? I also believe, that the four-quadrant data should be scrutinized for anomalies, but this would go far beyond the scope of this paper. I will mention it in the conclusions.

You very rightly observed, that with the existing polynomials it is theoretically possible to use it for non-integer number of blades.  During the discussion after presenting the findings presented in this paper at the smp'19 conference in Rome, I made the very same suggestion: 6 different sets of polynomials would be more accurate. It was rightly pointed out by the audience, that the reason for this form of polynomials (that is including the number of blades) is the consistency when switching the number of blades. This required consistency is only possible, if the number of blades are part of the polynomials. I will add a respective note to the paper. Thanks for bringing this up.

By the way, it is not clear at which stage of the processing the anomalies were introduced. Based alone on the open-water diagrams, it is not possible to spot them (see Fig. 1 and compare it to the diagrams for B4-70 on page 24). In the proceedings to AMT'17 I show some more snapshots of the data over the timeline.

Looking forward to your valuable comments!

Reviewer 2 Report

I do not feel that the research is very original, but I think it is interesting for students and for teaching purposes, since the Series B and BB are the only available tool for design for the average designer. It can be a starting propeller design that is optimized later.

I feel Fig 1 is very interesting and I have a question: the standard 1975 polynomials considers data from the 1969 tests?   

I recognize I used only polynomials, and I feel graphics when used with paper and pencil, present a large amount of error, but can be used for teaching purposes and the feel about the influence of some propeller characteristics into performance, can be assessed easily. How do you use the diagrams? Paper and pencil?

I fill that this paper need some practical examples. In the book “The Wageningen Propeller Series” by Kuiper, there are some practical and simple examples at Chapter 9. I wonder if you can include these examples in your paper.  

Author Response

Thank you for your kind review.

You asked, if the standard 1975 polynomials considers data from the 1969 tests. The paper by Oosterveld and Oossanen [8] states, that "these polynomials were obtained with the aid of a multiple regression analysis of the original open-water test data of the 120 propeller models comprising the B-series. All test data was corrected for Reynolds effects by means of 'equivalent profile' method developed by Lerbs." In the references it also mentions the paper "The Wageningen B-screw Series" by van Lammeren et al [3] from 1969, which includes the most recent tests. I believe it can be safely assumed, that this data was included in Oosterveld and Oossannen 1975 polynomials.

Regarding you question, how these diagrams are to be used: There are many ways to use them. One is certainly to use paper and pencil, as was the idea of the original inventors – with all the disadvantages (accuracy) and advantages (seeing actually what is going on). Yosifov et al even calculated polynomials for the lines of optimum efficiencies [13], so you find the optimum propeller without using the paper charts and – which is important – without employing optimization routines on the computer. As mentioned briefly in the Conclusion section, you can even write your own optimization routine on the computer using the published polynomials. You will notice, that sometimes the optimizer will fail, either finding no solution or alternating between two solutions or even finding the solution on the "upper" branch. But please note, this paper is NOT about optimizing propellers – either with paper and pencil or some computer routines – it is about the underlying data of these optimizations: The Polynomials. May I ask how you can trust these polynomials – or any if the derived diagrams or programs using them –, if they show such troubling behaviour? This question is the main focus of the paper and the whole thing about the diagrams is needed to show that the polynomials exhibit this behaviour. It cannot be seen in any other diagram, so I had to introduce the efficiency maps. I also noticed from all reviews, that the introduction and discussion of the efficiency maps conceals the main message of the paper. For this reason, I rearranged the paper and moved the discussion and examples how to use the diagrams into the appendix, since I believe they are still worth using. Maybe not for high-end tasks, but to get a quick overview and to check different cases in a graphical way. For this purpose I compared the solutions of the first example in Kuiper's book when using different approaches.

I once again have to stress, that the main focus of this paper was actually not to draw the diagrams to be used in design work (that's a very nice side-effect!), but to show the anomalies in the final polynomials. By the way that's why I think, the paper actually is original and very important, because nobody else has noticed these anomalies and many propeller designer (not just students) depend on the accuracy and reliability of the polynomials. I also now that many people run into problems when coding their optimization routines on the computer without knowing why. Now at least they know why...

Thank you for your valuable input!

Reviewer 3 Report

The work identifies anomalies in curve fit polynomials found in literature for empirical data that are seldom used to design marine propellers.

The paper is interesting to readers.

The contribution could be made clearer if the usefulness of designing propellers for fixed P_D and P_N, and for T_D and T_N, or even the physical meaning for doing so, was explained. This could improve the paper, as these
 design variables seem abstract and the anomalies are not encountered for more relevant design parameters, eg J, P/D, P or T.

Change:
"With the advent of the computer polynomials for the thrust and torque values were calculated from the available data sets"
to
With the advent of the computer polynomials for the thrust and torque, values were calculated from the available data sets

Change:
"Changing the presentation from open-water diagrams to efficiency maps"
to
Adding efficiency maps to open-water diagrams

Author Response

Thank you for kindly reviewing my paper.

You wrote that "The work identifies anomalies in curve fit polynomials found in literature for empirical data that are seldom used to design marine propellers." May I add two comments. Firstly, I could not identify the cause of the anomalies. They might come from the curve fit polynomials, but they might also have been introduced during previous fairing and scaling procedures. The Figure 1 shows the changes introduced over time. Secondly, the Wageningen B-series might be used less and less for selecting a propeller, but it is still widely used in industry for preliminary work or for investigating effect when changing main parameters.

You also mentioned that "The contribution could be made clearer if the usefulness of designing propellers for fixed P_D and P_N, and for T_D and T_N, or even the physical meaning for doing so, was explained." and I certainly can see your point! I will add an explanation, why these four design variables make finding the optimum propeller so much easier.

Since you wrote, that "[...] these design variables seem abstract and the anomalies are not encountered for more relevant design parameters, eg J, P/D, P or T." I might add, that in my view, these design variables are far from abstract (I will explain this in the section to be added, why we introduce these design variables and show their advantages). For example PD and Pn are used to find the optimum propeller, if the power is known (and TD and Tn, if the thrust is known). Since these anomalies occur for the design parameters Px and Tx, I might add that they also occur for P and T (albeit implicitly).

Thank you for your two proposed regarding the wording.

'Change:
"With the advent of the computer polynomials for the thrust and torque values were calculated from the available data sets"
to
With the advent of the computer polynomials for the thrust and torque, values were calculated from the available data sets'

What I mean, is:
'With the advent of the computer, the available data sets were used to calculate polynomials for the thrust and torque values.'

I hope this makes it easier to read. Thanks for pointing this out.

Change:
"Changing the presentation from open-water diagrams to efficiency maps"
to
Adding efficiency maps to open-water diagrams

Whereas you are right, that all of these diagrams show the open-water characteristics of propellers, I believe, that we should reserve the wording "open-water diagram" for the traditional presentation. This is how notion is used by most people. To avoid confusion, I believe we should use a different term (the Bp-δ diagrams also show open-water characteristics but nobody would refer to them as "open-water diagrams"). The wording "efficiency maps" is coined by the very first authors using this presentation, hence I would like to keep this notion.

Looking forward to your comments.

Reviewer 4 Report

See attached pdf

Author Response

Thank you for your very thorough review of my paper!

I noticed that all reviewers saw the emphasis of the paper in the Danckwardt diagrams and how they can be used for optimizing a propeller with paper and pencil. Since this is just a nice side-effect, the main purpose to introduce these diagrams is to show the problematic behaviour of the polynomials, which cannot be seen on any other diagram. Hence I had to rearrange the paper to put the goodie of the design part to a less prominent place in the paper - the appendix. I hope this puts the focus right.

May I comment on the issues you mentioned:

1.

By the look of it, the paper by Troost was not the first one. I added the reference [12] to the very first paper I could find to the paper (and I will try to get the author and the paper itself, so that I can fully reference it). I also have to admit, that the short introduction was a bit confusing. I hope I could put it right in the new version.

2.

I like the idea and moved the section into the introduction. Thanks.

3.

Regarding the re-ordering of the historical part to the front of the section, I do not feel, this helps in reading the paper. If this part were moved to the front of section 3, the reader would have to deal with historical facts without even knowing what efficiency maps are and why s/he should care. I find it always more interesting and easier to read, if you first get an idea what we talk about and then get a historical overview.

The main purpose of this paper is not to show, how propellers can be optimized with the help of efficiency maps, but to show the troubling behaviour of the polynomials. Hence I moved the sections dealing with the design into the appendix, because they might still be interesting to the reader. I hope the rephrased sections explains better, why we use Kt/J2, Kq/J3, Kt/J4, and Kq/J5.

Regarding the reduction gear ratio: The paper always uses the revs of the propeller shaft. I believe that every naval architect can calculate the required reduction ratio, if s/he knows the optimum shaft speed and the revs of the main engine, hence I did not deem it necessary to explain it.

Since this paper is not a text book about propeller design or optimization, explaining or even mentioning the matching of the propeller to the resistance curve would go far beyond the scope of this paper.

4.

The section mentioned by you was removed/rephrased in the new version. I also want to refer to the last paragraph of issue 3. By the way, this example does refer to a very practical design problem in the industry: In the quoting stage only the power, the shaft speed and the ship speed is known (from model tests with a stock propeller) and very accurate propeller dimensions have to be found to be able to quote to the prospective customer. Frankly speaking, the lack of the resistance curve and the relying upon power figures also surprised me when I moved academia into the industry.

5.

Thank you so much for spotting the mistake in the numbers!! You were the only reviewer to spot this! I calculated many different options to get "nice" figures and must have mixed up different cases. This should be sorted in the new paper.

Final comment

I must emphasise again, that the focus of this paper is not about solving propeller design problems, but about the existence of multiple solutions or no solutions at all for the optimizing problem. It is also not a text book about different approaches to solve the optimization problem, so I do not think it necessary to include another methodology.

Thank you for your helpful and very valuable suggestions.

Round 2

Reviewer 2 Report

Thank you for the time you employed in the examples. The paper is now more useful and at the same time, the anomalies in the use of diagrams are mentioned (yes, this is the main novelty of the work).

So, this version has improved the previous one and my deccision is Accept.

Author Response

Thank you for your positive review. I also have the feeling, that the new version is much improved thanks to your helpful input.

Reviewer 4 Report

See attached pdf

Author Response

Thank you for your valuable comments.

Re your comment about your 1st review: Is is true, that the maximum efficiency on the diagram can be found at the line P/D=1.4 for the mentioned cases. But these are at the boundary of the available data and as they say "Never trust an optimum at the boundary of your data". Common sense tells us, that there will be propellers with an higher efficiency beyond the boundary. And indeed, closer inspection reveals that the efficiency curves for PD and Pn = const are still rising near the boundary, with  ∂η/∂(P/D) > 0 at P/D=1.4. This is shown in the paper in Figure 5, orange line.

As you might have noticed, I had to rewrite a part of the paper, because nobody (and many people have seen this paper during my presentation at the smp'19 in Rome, but also none of the reviewers of this paper and the presentation) has noticed, that the two extrema are actually a maximum and a minimum. Admittedly it is hard to notice.

Re your comment about the example: As you absolutely correctly observed, "the iso-PD is nearly parallel to the iso-efficiency curve at P/D=0.57 (sic) in the range of J between 0.6-0.78." I am sure, this is a typo and means "η=0.57". These lines are nearly parallel, but they are not exactly parallel, as can be seen in Figure 5 of the revised manuscript. As explained in the revised manuscript, there is an area, where the pitch ratio can be chosen without too much effect on the efficiency. But exactly speaking, the efficiency drops, if the the pitch gets higher after the maximum to increase again after the minimum. Is this physical? I believe no. And this is the fundamental point of the whole paper! (I just needed these diagrams, to show in an easily understandable way, what is going on. The examples are just there to illustrate the points. This is not a paper about selecting the optimum propeller.)

I hope I explained in the first paragraph, why the optimum at PD=0.07, J=0.92, η=0.65 is not an optimum.

Your conclusion, that the pitch ratio can be freely selected in certain areas, would be wonderful, if it were true. Unfortunately this is a feature of the underlying polynomials and not the physical reality. And that is the point I want to make with my paper.

Thank you for your comment about the "fundamental propeller optimizing problem". I changed it to "basic propeller optimizing problem". In my view this is in accordance to the literature, e.g. the list of "Solutions to the propeller problems", published by Ulrich and Danckwardt in "Konstruktionsgrundlagen für Schiffsschrauben" (1956) ["Fundamentals for the Construction of ship screws"]. The list contains 24 propeller problems, where the first 6 problems are those mentioned by me.

Technically speaking, the ship is ignored, too, if the T,V,D triad is used. The power calculated to push the ship at that speed has to be compared to the available power from the prime mover and the ship speed (and R) adjusted, to not ignore the ship. I am sure, that all younger scientist all over the world learn all this during their lectures about ship propulsion!

Re the size of the paper: The main paper consists of 10 pages with many illustrations. I do not deem this as too long. The biggest part are the two Appendixes. The detailed explanation and the examples were included, because other reviewers requested them. The by far largest part is the second appendix with the diagrams. This is hard to reduce, because the diagrams should remain usable. We could ask the editor about her opinion?

Round 3

Author Response

Thank you for your review. May I comment on each point, but I wonder if the whole discussion is based on one simple misunderstanding: I believe that the open-water data in the form of the polynomials are flawed. I prove this by presenting the data in the form of efficiency maps and argue, that these do not represent the physical reality, because of features shown in the map, which cannot be explained neither by physics nor by experience. The reviewer on the other hand seems to believe that the open-water data is correct and draws his/her conclusions from this unquestioned data. With these two different approaches in mind, the conclusions must be different. To speed the review process up, may I ask the reviewer, if s/he believes that the underlying data in the form of the current polynomials is beyond all doubt (or if it might be possible that flaws can be found in the polynomials)?

Reviewer's comment:
From this figure it is clearly shown that:

There is always an optimum within the range of parameters of the B-series, even for smaller values of PD (i.e red line with PD=0.07 in figure) although in this case the optimum is not global (it points towards greater values of P/D), There is a region of ‘flatness’ of the results regarding efficiency. This is a very interesting design result, since it gives the freedom to the designer to select the pitch ratio (which affects corresponding optimum revolutions) with other criteria than the optimum hydrodynamic performance.

Your observations are correct, but

for low PD values – the region right of the overlap – there exists obviously a propeller with higher efficiency than that of the propeller at the boundary of P/D=1.4. If the propeller designers takes the propeller with P/D=1.4, s/he would not have a propeller of optimum design. Hence in my view, the propeller with P/D=1.4 is not an optimum propeller. The flatness is not an interesting design result, it is the result of the polynomial representation, which must be wrong: There cannot be a drop and subsequent rise in efficiency! There is absolutely no physical reason for this behaviour. Consequently there exists no freedom for the propeller designer in this region.

May I ask the reviewer, how s/he would explain the drop and raise physically.

Reviewer's comment:
Comment
: The author’s Surprising behavior... is based on considerations when approaching the P/D=1.4 line (for example his section 3.2 and table 2). The whole table 2 is devoted to this ‘Surprising behavior...’. In the above reply, the author states: "
Never trust an optimum at the boundary of your data" (obviously correct). Concluding, the author assures us that there is not at all any surprising behavior, simply boundary data can be questionable... something well known to the scientific community.

Sorry, I have to disagree. The surprising behaviour is the overlap of the ηo,max lines, which cannot be explained physically. Table 2 just classifies the overlap. By the way, TD|P/D=max does not show the optimum propeller, because it is on the branch where the efficiency has a local minimum.

Reviewer's comment:
Obviously one can expand this line of thought, asking why to trust the data in the other boundary i.e. P/D=0.5. Furthermore, are the data equally questionable in all range of J values (something which obviously can not be answered in any rational way).

You are absolutely right. And this is discussed in the manuscript in Section 4.3 "Accuracy of the polynomials".

Reviewer's comment:
Going now to greater J values and smaller PD, as shown in above figure (for example red line with PD=0.07) there is a definite optimum at P/D=1.4 within the range of B-series P/D.

The important part seems to be "within the range of [tested] B-series P/D".

Reviewer's comment:
If we assume equal uncertainty of B-series data around the P/D=1.4 line for all J, then why to trust the results of table 2 and not those of the above figure.

Both the Table 2 and Figure 5 ("the above figure") are based on the same data, the open-water data in the form of the current polynomials. Both simply describe the behaviour of the data and hence this is not a question of trust. Both are trustworthy in the sense, that the show aspects of the polynomials. (It does not say, that the polynomials are trustworthy.)

Reviewer's comment:
Since there is not any obvious reason to do so, the author’s statement in the introduction: ‘ ... or even no optimum at all for certain conditions’ is incorrect and has to be eliminated.

May I ask the reviewer to explicate, why s/he believes that "there is no obvious reason to" doubt the underlying polynomials?

The efficiency maps clearly show, that there is not propeller with a global maximum in efficiency in the region right of the overlap. But there should be one as explained in the last paragraph of Section 4.1 in the manuscript.

Reviewer's comment:
Furthermore, the author statement: Common sense tells us, that there will be propellers with a higher efficiency beyond the boundary, is obviously correct and it is in line with the observation that within the range of existing diagrams there is an optimum (maybe not global) coinciding with the P/D=1.4 line!

Indeed. But the propeller on the boundary is not the optimum propeller at all! Why should this propeller right on the boundary taken as the optimum propeller, when there is a propeller with higher efficiency available beyond this boundary?

Reviewer's comment:
Concluding, in order to accept the paper, the author has to correct the paper according to the obvious findings of the above figure (which is representative of what really happens in all cases of table 2).

Please see my remarks near the top. I added a sentence saying that there exists a (global) optimum beyond the boundary for PD=0.15 and 0.07.

Reviewer's comment:
More specifically the statement in the introduction ‘... or even no optimum at all for certain conditions’ has to be eliminated.

I adapted the sentence mentioned in the abstract.